# Bimodal centromeres in pentaploid dogroses shed light on their unique meiosis

V. Herklotz[1,10], M. Zhang[2,10], T. Nascimento[2,10], R. Kalfusová[3], J. Lunerová[3], J. Fuchs[4], D. Harpke[4], B. Huettel[5], U. Pfordt[2], V. Wissemann[6], A. Kovařík[3✉], A. Marques[2,7✉] & C. M. Ritz[1,8,9✉]

Sexual reproduction relies on meiotic chromosome pairing to form bivalents, a process that is complicated in polyploids owing to the presence of multiple subgenomes[1]. Uneven ploidy mostly results in sterility due to unbalanced chromosome pairing and segregation during meiosis. However, pentaploid dogroses (*Rosa* sect. *Caninae*; $2n = 5x = 35$) achieve stable sexual reproduction through a unique mechanism: 14 chromosomes form bivalents and are transmitted biparentally, while the remaining 21 chromosomes are maternally inherited as univalents[2,3]. Despite being studied for over a century, the role of centromeres in this process has remained unclear. Here we analyse haplotype-resolved chromosome-level genome assemblies for three pentaploid dogroses. Subgenome phasing revealed a bivalent-forming subgenome with two highly homozygous chromosome sets and three divergent subgenomes lacking homologous partners, therefore explaining their meiotic behaviour. Comparative analyses of chromosome synteny, phylogenetic relationships and centromere composition indicate that the subgenomes originated from two divergent clades of the genus *Rosa*. Pollen genome analysis shows that subgenomes from different evolutionary origins form bivalents, supporting multiple origins of dogroses and highlighting variation in subgenome contributions. We reveal that bivalent-forming centromeres are enriched with *ATHILA* retrotransposons, contrasting with larger tandem-repeat-based centromeres mainly found in univalents. This centromere structural bimodality possibly contributes to univalent drive during female meiosis. Our findings provide insights into the unique reproductive strategies of dogroses, advancing our understanding of genome evolution, centromere diversity and meiotic mechanisms in organisms with asymmetrical inheritance systems.

Whole-genome duplication or polyploidy is a frequent phenomenon across the phylogeny of land plants[4]. Meiosis is essential for sexual reproduction, ensuring the reduction in genomic content in gametes through chromosome pairing and exchanges between non-sister chromatids, that is, crossovers[5,6]. Polyploidy often results from meiotic failure, that is, the generation of unreduced gametes, which poses challenges to meiotic chromosome pairing and the maintenance of sexual reproduction[1,7]. Thus, polyploids often skip sexual reproduction by promoting vegetative propagation[8] or apomixis[9]. However, in many allopolyploids, in which distinct subgenomes come into contact through hybridization, recombination partners from homologous chromosomes (same parental subgenome) are preferred, while recombination between homoeologous chromosomes (different parental subgenomes) is suppressed[10,11].

The genus *Rosa*, which comprises approximately 150 species, is a typical example of evolution through frequent polyploidy and hybridization events[12], which is also reflected by the large variety of cultivated roses with a long breeding history that includes both processes. The genus comprises two major clades, the *Rosa* and allies clade and the *Synstylae* and allies clade with subg. *Hulthemia*, subg. *Hesperhodos* and sect. *Pimpinellifoliae* as the basalmost splits[12,13]. Available genomes from diploid roses of sect. *Synstylae*[14–16] and sect. *Rosa*[17] revealed high levels of synteny, enabling comparative studies in this taxonomically difficult genus. Studies on tetraploid cut roses (*Rosa hybrida*) undergoing regular meiosis have shown that most genomic markers were recombined freely from all four chromosome sets, but preferential recombination between chromosomes and even chromosome arms vary[18,19].

Within the *Synstylae* clade, allopolyploid dogroses (*Rosa* sect. *Caninae* (DC.) Ser.) exhibit a unique reproductive strategy known as *Canina* meiosis, in which the selective chromosome pairing results in a mixed mode of inheritance—combining biparental transmission of bivalents

[1]Senckenberg Museum for Natural History Görlitz, Senckenberg–Leibniz Institution for Biodiversity and Earth System Research, Görlitz, Germany. [2]Department of Chromosome Biology, Max Planck Institute for Plant Breeding Research, Cologne, Germany. [3]Department of Molecular Epigenetics, Institute of Biophysics, Czech Academy of Sciences, Brno, Czech Republic. [4]Leibniz Institute of Plant Genetics and Crop Plant Research (IPK) Gatersleben, Seeland, Germany. [5]Max Planck Genome Centre Cologne, Max Planck Institute for Plant Breeding Research, Cologne, Germany. [6]Institute of Botany, Systematic Botany Group, Justus-Liebig-University, Gießen, Germany. [7]Cluster of Excellence on Plant Sciences (CEPLAS), Heinrich-Heine University, Düsseldorf, Germany. [8]International Institute (IHI) Zittau, TUD Dresden University of Technology, Zittau, Germany. [9]German Centre for Integrative Biodiversity Research (iDiv) Halle-Jena-Leipzig, Leipzig, Germany. [10]These authors contributed equally: V. Herklotz, M. Zhang, T. Nascimento. ✉e-mail: kovarik@ibp.cz; amarques@mpipz.mpg.de; christiane.ritz@senckenberg.de

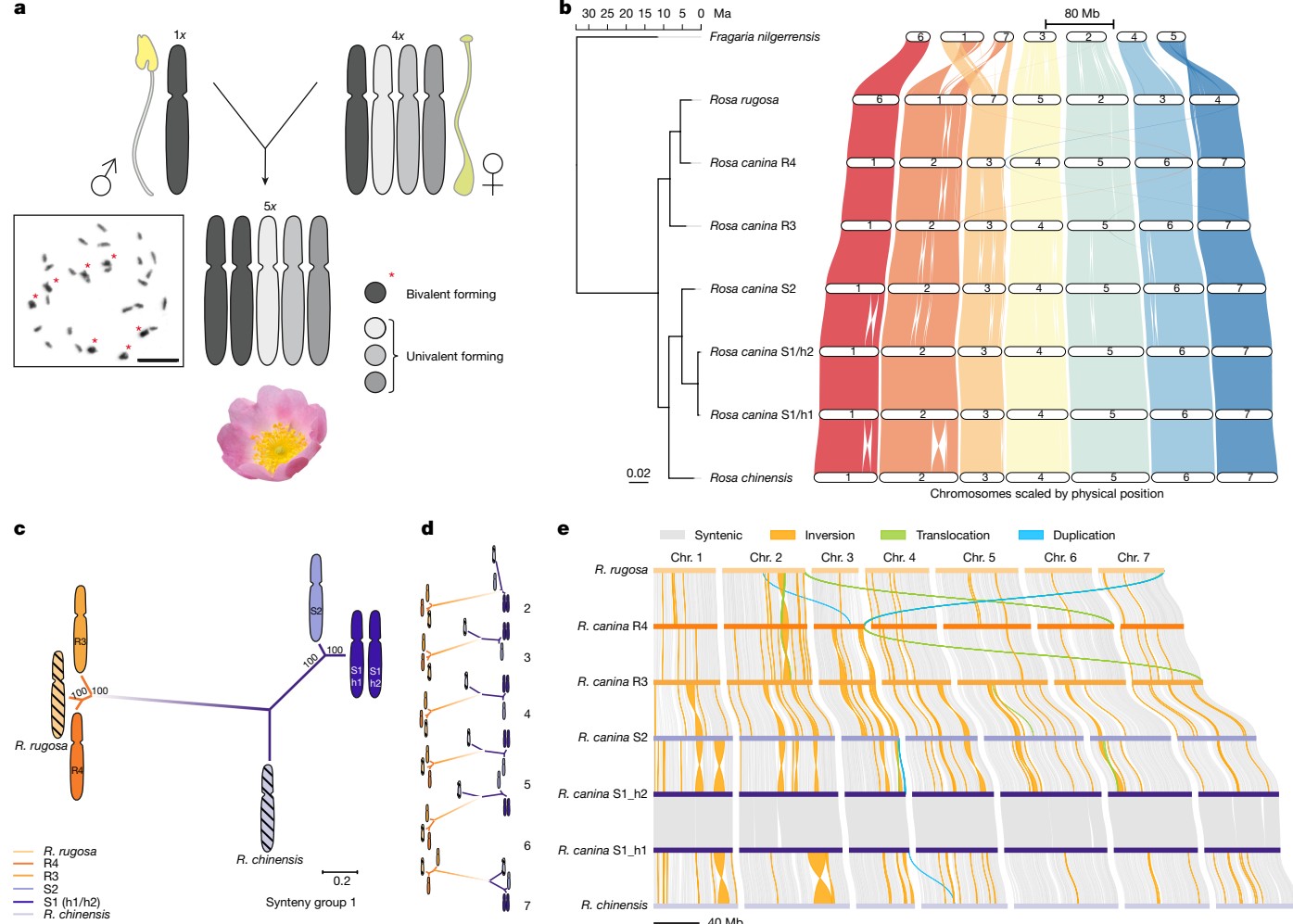

**Fig. 1 | Synteny-based classification and phylogenetic relationships of *R. canina* subgenomes. a**, The known sexual reproduction of the pentaploid *R. canina* (2*n* = 5*x* = 35). Different chromosome sets are represented by one chromosome each. During *Canina* meiosis[2,3,22,23], two chromosome sets form bivalents (dark grey and red asterisks) during meiosis and are transmitted through both pollen and egg cells[24–26]. The remaining three sets form univalents (lighter grey) and are transmitted through the egg cell only[24–26]. Diakinesis of male meiosis I (*n* = 15) of *R. canina* is shown on the left. Scale bar, 10 μm. **b**, GENESPACE synteny and phylogenetic relationships of the five chromosome sets of *R. canina* and their close diploid relatives *R. chinensis* (sect. *Synstylae*) and *R. rugosa* (sect. *Rosa*) with a dated phylogenetic tree constructed using 16,372 orthologous genes on the left. Each colour indicates synteny to each *R. chinensis* chromosome, which was used as reference to name the *R. canina*

chromosomes. **c**,**d**, Unrooted maximum-likelihood phylogenies of the homoeologous *R. canina* chromosomes and chromosomes from the respective linkage groups of the diploid species *R. chinensis* (sect. *Synstylae*) and *R. rugosa* (sect. *Rosa*) based on alignments of whole-chromosome sequences. The phylogeny of synteny group 1 chromosomes is shown in **c**. Synteny groups 2–7 are shown in **d**. Filled chromosomes refer to subgenomes of *R. canina* belonging to the *Synstylae* clade (violet/light blue) and the *Rosa* clade (dark/light orange). Chromosomes from the diploid roses are indicated by hatching. **e**, Synteny and rearrangement analyses (SyRI) of the *R. canina* genome assembly. Pairwise comparisons of the synteny of all *R. canina* subgenomes (S1_h1/S1_h2, S2, R3 and R4) are juxtaposed against the corresponding chromosomes (chr.) of *R. chinensis* and *R. rugosa*. Only synteny blocks and rearrangement blocks greater than 50 kb in length are shown.

and uniparental transmission of univalents within the same nucleus[20,21]. First observed in the early twentieth century, this mechanism is most common in pentaploid dogroses (2*n* = 5*x* = 35), in which the male and female parents contribute 7 and 28 chromosomes, respectively, to the zygote[2,3,22,23]. During meiosis, 14 chromosomes form 7 bivalents, while the other 21 chromosomes remain as univalents. Bivalent-forming chromosomes from two highly homozygous sets are transmitted to both sperm and egg cells, whereas univalents are inherited exclusively through the egg cell and excluded from pollen grains, restoring pentaploidy in the offspring through the fusion of haploid male and tetraploid female gametes[24–27] (Fig. 1a). Despite extensive study, the precise cellular mechanisms underlying this asymmetric inheritance remain poorly understood.

Bivalent-forming and univalent chromosomes are thought to have originated from multiple ancient hybridization events[28–30]. Phylogenetic

studies based on maternally inherited plastids suggest that dogroses are polyphyletic within the *Synstylae* clade, with subsects. *Rubigineae* and *Caninae* are separated by species exhibiting regular meiosis[13,31–33]. Cytogenetic evidence shows that bivalent-forming chromosomes in subsect. *Caninae* become univalents in subsect. *Rubigineae* and vice versa[34,35]. The rose-specific (peri)centromeric satellite repeat (*CANR4*) is notably enriched in dogrose univalents, possibly linking centromere expansion to their drive during female meiosis[36,37]. Although other uneven polyploid systems with hemisexual reproduction exist[38–40], the meiosis observed in dogroses is unique among eukaryotes. However, the lack of genomic studies has hindered understanding of how centromere properties contribute to this enigmatic reproductive behaviour.

Here we present a high-quality haplotype-phased chromosome-scale assembly of the pentaploid genome of *Rosa canina* (subsect. *Caninae*) and compare it with another *R. canina* individual and *R. agrestis*

(subsect. *Rubigineae*), both from the Darwin Tree of Life (DToL; https://www.darwintreeoflife.org/). Subgenome-aware analyses revealed that dogroses are composed of four subgenomes with one subgenome being present in two highly homozygous haplotypes and the other three in only one. Targeted sequencing of rose-specific single-copy orthologues (SCOs) from pollen DNA, together with the analysis of synthetic $F_1$ hybrids, confirmed that the two-copy subgenome is biparentally inherited and therefore forming bivalents, while 21 non-recombining univalents from three distinct subgenomes are exclusively inherited through the female germline. Our SCO-based phylogenetic analysis supports the multiple-origin nature of dogroses, as subgenomes from different evolutionary origins were found in pollen, that is, forming bivalents. We also identified a bimodal centromere architecture with small (retrotransposon *ATHILA*-based) and large (tandem repeat *CANR4*-based) centromeres. Notably, *CANR4*-based centromeres were prevalent in univalents, possibly contributing to their drive in asymmetric female meiosis. Our results therefore provide a valuable basis for studying the trade-offs between sexual and asexual reproduction within a single genome.

## Unlocking dogrose pentaploid genomes

Although *R. canina* has been recognized as a pentaploid species ($2n = 5x = 35$) for decades, its genome has remained unresolved owing to its complex polyploid structure and hybrid origin. To address this, we assembled a de novo haplotype-resolved, chromosome-level genome using PacBio HiFi sequencing (23× coverage) and chromatin conformation capture (Hi-C) data (Supplementary Table 1). The total size of the assembled 35 pseudochromosomes is about 2.4 Gb, achieving 99.2% completeness in terms of gene content (Extended Data Fig. 1). This high-quality reference genome provides a critical resource for understanding *R. canina*'s genetic features, asymmetric meiosis and hybridization.

All-to-all chromosome alignments revealed seven syntenic groups in *R. canina*, each consisting of five chromosomes (Supplementary Fig. 1). In each group, two chromosome sets consistently exhibited 99–100% similarity, indicating two haplotypes of the same subgenome, while the remaining three chromosome sets showed lower similarities (95–98%; Supplementary Fig. 1 and Supplementary Data 1), which may derive from three different subgenomes. Phylogenetic analyses based on gene and chromosome data, using the diploid rose genomes of *Rosa chinensis* (sect. *Synstylae*)[16] and *Rosa rugosa* (sect. *Rosa*; https://www.darwintreeoflife.org/) as references, revealed that two subgenomes are closely related to *R. chinensis* and were therefore designated 'S', while the other two are more similar to *R. rugosa* (Fig. 1b–e) and were therefore designated 'R'. The two highly similar *Synstylae*-like haplotypes were named S1_h1 and S1_h2, while the more divergent *Synstylae*-like chromosome set was named S2. The *Rosa*-like subgenomes were named R3 and R4, respectively. Moreover, using subgenome-specific $k$-mers[41], we observed that chromosomes assigned to the same subgenome clustered together both in the $k$-mer heat map and the principal component analysis plot, confirming the correct assignment of the four primary subgenomes (S1, S2, R3 and R4; Extended Data Fig. 2a–c). These findings resolve the long-standing question regarding the identity of the five homoeologous chromosomes within each syntenic group.

To further validate the evolutionary relationships of the subgenomes in pentaploid *R. canina*, we conducted orthologous cluster analysis and genome-wide comparisons of synonymous substitution rates ($K_s$) between *R. canina*, *R. chinensis* and *R. rugosa*. These results were consistent with previous phasing assignments and confirmed the allopolyploid origin of *R. canina* (Supplementary Figs. 2–4). Structure-based pairwise chromosomal analysis across the subgenomes of *R. canina* and genomes of *R. chinensis* and *R. rugosa* revealed a strong conservation of synteny between the two haplotypes of the homozygous S1 subgenome (S1_h1/h2). By contrast, the S1 subgenome showed much lower synteny

with the other three subgenomes (S2, R3, R4), which were characterized by large inversions, duplicated regions and translocations (Fig. 1e). Notably, the R3 and R4 subgenomes of *R. canina* exhibit greater synteny to the *R. rugosa* genome than to the other subgenomes of *R. canina* (Fig. 1e), supporting their origination from sect. *Rosa*[35]. Despite the distinct divergence and origins of four subgenomes, a comparison of *R. canina* chromosomes of all subgenomes against *R. rugosa* syntenic chromosomes revealed no evidence of differential fractionation (loss of one or the other copy of a duplicated gene; Supplementary Fig. 5). To detect differential evolutionary rates between subgenomes, we analysed the ratio of nonsynonymous versus synonymous substitution rates and revealed, besides a few outliers, strong purifying selection across orthologous genes in all subgenomes (Supplementary Fig. 6). Together, these results suggest an absence of large-scale subgenome dominance.

We next took advantage of the recent HiFi and Hi-C sequencing datasets from the DToL (https://www.darwintreeoflife.org/) for another *R. canina* accession (European Nucleotide Archive (ENA): PRJEB79802) and from *Rosa agrestis* (subsect. *Rubigineae*; ENA: PRJEB79880) to generate de novo pseudochromosomes (Supplementary Table 2). Comparative analysis revealed a high degree of synteny between our *R. canina* S27 genome and the DToL *R. canina*, both sharing the same subgenome composition (Extended Data Fig. 3a and Supplementary Fig. 7a,b). The *R. agrestis* genome, also pentaploid with 35 pseudochromosomes, displayed a different subgenome composition, with two highly similar haplotypes for the R4 subgenome (R4_h1/h2) and only one copy of the S1 subgenome (Extended Data Fig. 3b–e and Supplementary Fig. 7c,d). Comparative subgenome phasing revealed a gradient of differentiation between the subgenomes of *R. canina* and *R. agrestis*. The S1 subgenomes of both species were the least differentiated and clustered together (Supplementary Fig. 8a). This was followed by the R4 subgenomes, which exhibited a slightly higher degree of differentiation compared with S1 (Supplementary Fig. 8b). The most pronounced differentiation was seen in the S2 and R3 subgenomes, which were distinctly separated in both species (Supplementary Fig. 8c,d). This pattern suggests that the subgenomes S2 and R3 are accumulating more divergence over time.

## Tracing the hybridizations of dogroses

To trace the hybridization history of dogroses, we identified subgenome-specific long terminal repeat retrotransposons (LTR-RTs) and estimated their insertion times in *R. canina* and *R. agrestis* to determine the timing of subgenome differentiation before hybridization. Notably, we observed a distinction in the median insertion times of LTR-RTs: the S1, S2 and R3 subgenomes in *R. canina* were estimated to have diverged around 0.7 million years ago (Ma), while the R4 subgenome was older, at approximately 1.2 Ma (median values, 95% confidence intervals; Extended Data Fig. 4a–c). In *R. agrestis*, the median insertion times for S1 and S2 (~0.7 Ma) and for R4 (~1.2 Ma) were the same as for *R. canina*, whereas a slightly older median insertion time was detected for the R3 subgenome (around 0.9 Ma; Extended Data Fig. 4c). These results suggest that the combination of S1, S2 and R3 subgenomes arose at different timepoints in *R. canina* and *R. agrestis*. This is further supported by the $K_s$-based divergence time estimation obtained from SCOs (Supplementary Fig. 6) and comparable findings from the high differentiation between their R3 subgenomes (Supplementary Fig. 8). Together, these findings suggest that modern dogroses originated through independent, stepwise hybridization events.

## Unlocking dogrose reproduction mode

Only bivalent-forming chromosomes are able to segregate properly and produce viable haploid ($1x$) pollen in dogroses. We therefore used flow sorting to isolate pollen nuclei as a proxy to confirm which subgenomes are exclusively pollen-inherited and form bivalents in dogroses. We successfully collected around 200,000 generative nuclei from

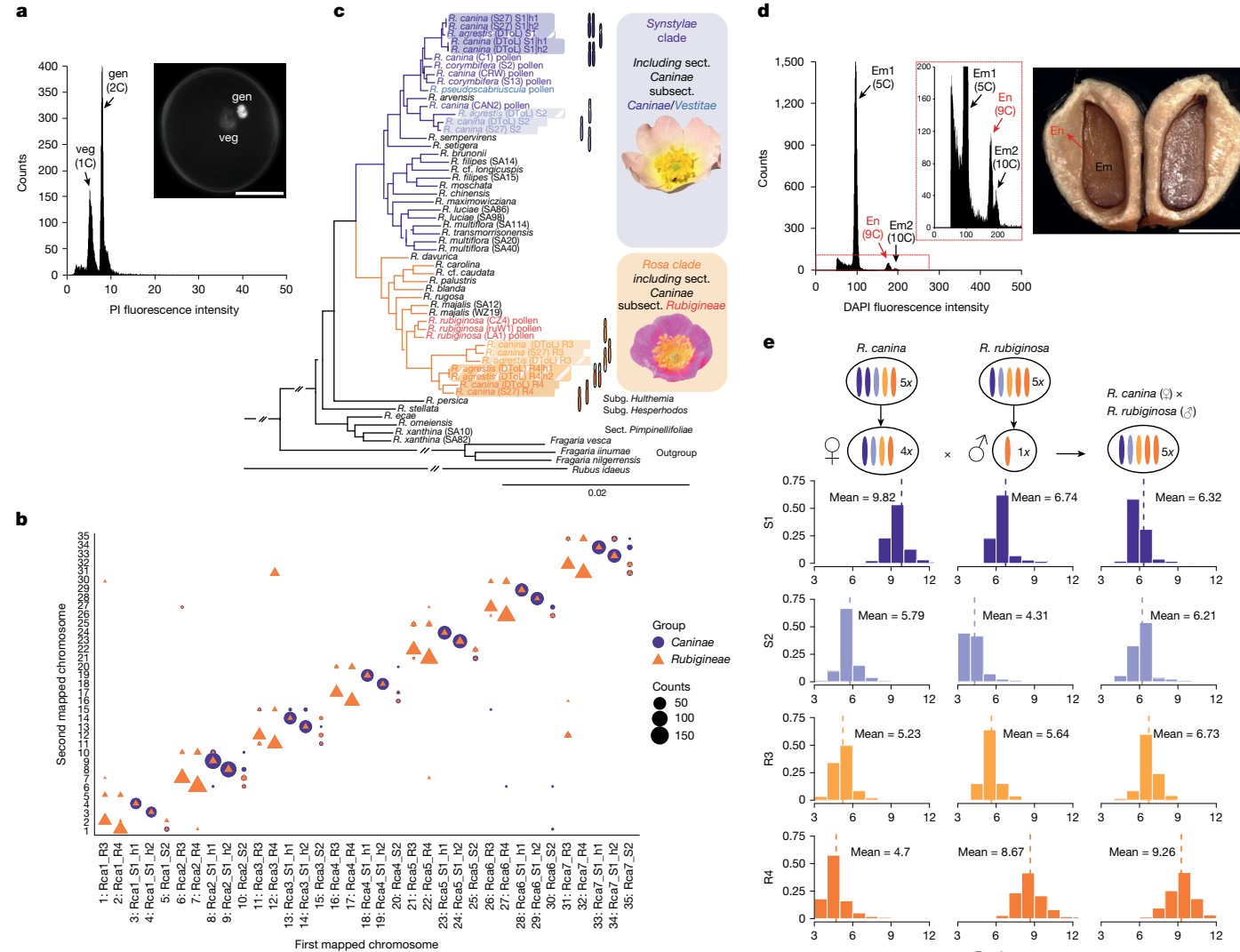

**Fig. 2 | Experimental validation of the reproduction mode of pentaploid dogroses. a**, Flow cytometry analysis of *R. canina* isolated pollen nuclei. Vegetative (veg) and generative (gen) nuclei differ in their DNA content in the binucleate pollen grains. Inset: an intact pollen grain after DAPI staining of both nucleus types. Scale bar, 10 μm. **b**, Genome-wide pollen SCO mapping of eight dogrose species (subsect. *Caninae*: three samples of *R. canina* and two samples of *R. corymbifera*; subsect. *Rubigineae*: three samples of *R. rubiginosa*) to the *R. canina* S27 genome. The bubble map represents chromosomal hits, which were selectively filtered to display loci with a single alternative hit. The size of the symbols corresponds to the mean counts of pollen SCOs mapped to each chromosomal pair, identifying seven pollen-inherited chromosomes from the S1 subgenome within the *R. canina* (subsect. *Caninae*) genome assembly. In *R. rubiginosa* (subsect. *Rubigineae*), pollen SCO mapped preferentially to the R3 and R4 subgenomes. **c**, Maximum-likelihood phylogeny of the genus *Rosa* based on SCO loci including those retrieved from chromosome-scale

assemblies and all pollen samples. Nodes with less than 100% bootstrap are indicated by dashed lines. **d**, Flow cytometry analysis of nuclei isolated from nutlets of *R. canina*, showing an endosperm/embryo ratio of 1.8, corresponding to the expected 9C/9x endosperm and 5C/5x embryo ratio (in which C denotes unreplicated haploid DNA content; x is the basic chromosome number), confirming sexual reproduction and endosperm fertilization. Em1, embryo G0/G1; Em2, embryo G2; En, endosperm G0/G1. Scale bar, 1.5 mm. **e**, Assessment of the parental genomes contribution of a synthetic hybrid between *R. canina* (female donor) and *R. rubiginosa* (male donor). The x axis shows the coverage histogram of the short reads from this species mapped to each *R. canina* subgenome. The y axis shows the probability densities. The hybrid revealed doubled coverage for R4, indicating the presence of two sets of R4 copies, biparentally inherited, while only one set of maternally inherited S1 was detected, confirming sexual reproduction and the subgenome's inheritance through male and female meiosis.

pollen samples (Fig. 2a) of 9 pentaploid dogrose accessions (including 3 accessions of *R. canina*, 2 of *R. corymbifera*, both subsect. *Caninae*; and 3 of *R. rubiginosa*, subsect. *Rubigineae*) and analysed them using single-copy orthologous nuclear locus target enrichment (Methods). This enabled us to create sample-specific reference sequences for each SCO locus. We mapped a total of 5,794 SCO sequences to the *R. canina* S27 genome and identified 7 major chromosome pairs, with most hits in *R. canina* pollen located on the S1_h1/h2 chromosomes (Fig. 2b and Supplementary Data 2). By contrast, *R. rubiginosa* pollen showed hits primarily on the R3 and R4 chromosomes (Fig. 2b and Supplementary Data 2). By leveraging dogrose pollen SCO mappings on the *R. canina*

genome, we unambiguously identified seven pairs of bivalent-forming chromosomes across several dogrose species. Moreover, mapping SCO loci to the *R. agrestis* genome revealed that the R4_h1/h2 subgenome forms bivalents in this species (Extended Data Fig. 5a). These results confirm that bivalent-forming chromosomes in subsect. *Caninae* form univalents in subsect. *Rubigineae* and vice versa[34,35].

Next, we aligned SCO loci obtained from the *R. canina* S27, *R. canina* DToL and *R. agrestis* DToL subgenomes, along with pollen DNA from section *Caninae* and different diploid rose samples and outgroups. This resulted in 58 sequences, totalling 642,158 positions derived from 1,904 concatenated SCO loci. Subgenome-wise as well as chromosome-wise

phylogenetic analysis delineated two large clades within the genus *Rosa*: the *Synstylae* clade and the *Rosa* clade, as well as the earlier splits of subg. *Hulthemia* (*Rosa persica*), subg. *Hesperhodos* (*Rosa stellata*) and sect. *Pimpinellifoliae* (Fig. 2c and Supplementary Data 3). The tree corroborates the allopolyploid origin of dogroses[13,28–30]. In *R. canina* (subsect. *Caninae*), the bivalent-forming subgenome (S1) and the univalent-forming subgenome S2 clustered in the *Synstylae* clade, while the univalent-forming subgenomes R3 and R4 were part of the *Rosa* clade, sister to the European species *R. majalis*. All of the subgenomes of *R. agrestis* grouped with the corresponding *R. canina* subgenome samples, supporting a common origin of individual subgenomes despite the high differentiation observed in S2 and R3 subgenomes (Fig. 2c and Supplementary Fig. 8). However, the pollen SCO data from all *R. rubiginosa* samples grouped as sister to both R3 and R4 subgenomes from *R. canina* and *R. agrestis*, respectively, implying multiple origins of the bivalent-forming chromosomes in subsect. *Rubigineae*. Notably, the pollen SCO data from *R. canina* 'CAN2' was sister to the univalent-forming subgenome S2, suggesting some intraspecific variation in the bivalent-forming subgenomes within *R. canina* (Fig. 2c and Supplementary Data 3), as indicated by variation in microsatellite alleles from bivalent-forming chromosomes in *R. canina*[26]. Supported by the respective clustering of bivalent-bearing pollen data from subsect. *Caninae* and subsect. *Rubigineae*, our data demonstrate an independent origin for the bivalent-forming subgenomes of dogroses and, consequently, the independent origins of asymmetric meiosis[34,35]. Furthermore, the finding that representatives of subsect. *Caninae* and subsect. *Rubigineae* (including the newly generated plastome assemblies for *R. canina* and *R. agrestis*) contain phylogenetically distant plastids from the *Synstylae* clade[13,31–33] (Extended Data Fig. 5b) supports the hypothesis that two progenitors from the *Synstylae* clade formed reciprocal hybrids, which subsequently incorporated *R* genomes through pollen donors.

While haploid (1*x*) pollen nuclei are clearly a product of *Canina* meiosis (Fig. 2a,b), tetraploid (4*x*) sexually derived egg cells were inferred by the respected embryo/endosperm ratio in seeds (Fig. 2d). In sexually reproducing diploids with double fertilization of the egg cell and the polar nuclei, the endosperm/embryo ratio is 1.5 (3*x* endosperm/2*x* embryo; Extended Data Fig. 5c). However, in sexually reproducing 5*x* dogroses, the ratio was found to be 1.8, indicating a 9*x* endosperm and a 5*x* embryo (Fig. 2d), similar to previous findings[42]. To further check the reproduction mode of dogroses, we have investigated the genome composition of two synthetic hybrids obtained from controlled crossing experiments[43]. In the first cross, the female gamete came from *R. canina* (subsect. *Caninae*) and the male donor was *R. rubiginosa* (subsect. *Rubigineae*). As anticipated from the result of the *Canina* meiosis, the subgenome contribution in the hybrid was consistent with the expected 4*x* egg cell containing one copy of each S1/S2/R3/R4 subgenome from *R. canina* and a 1*x* pollen nucleus with the R4 subgenome from *R. rubiginosa* (Fig. 2e and Extended Data Fig. 5d). In the second case, the female gamete came from *R. rubiginosa* and the male donor was *R. corymbifera*—a very close relative of *R. canina*[44]. Again, the hybrid showed the expected subgenome composition, consisting of a male haploid S1 subgenome and a female tetraploid S1/S2/R3/R4 subgenome (Extended Data Fig. 5e,f). These results are in agreement with S1 and R4 being bivalent-forming subgenomes and confirm the 1*x* male versus 4*x* female gamete composition. Our findings further suggest that different subgenomes are potentially interexchangeable in hybridization events; however, hybrids in extant populations originated mostly from unreduced eggs suggesting some subsection-specific differentiation subgenomes, which might impact bivalent formation[45].

## The bimodal centromeres of *R. canina*

To gain further insights into the subgenome differentiation of *R. canina*, we aimed to characterize its global repeat composition, both genome-wide and specifically at centromeres. The *R. canina* genome exhibited a very high content of LTR *Ty1/Copia* elements, which made up 40% of the total repeat content, compared with 23% of *Ty3/Gypsy* elements. Among the *Ty1/Copia* elements, the *BIANCA* family accounted for more than 45% of all annotated full-length LTR-RTs while, among the *Ty3/Gypsy* elements, *RETAND* and *ATHILA* were the largest classes found, comprising 10% and 8% of all annotated full-length LTR-RTs, respectively. Tandem repeats, that is, satellite DNA, were mainly composed of the (peri)centromeric *CANR4* repeats[15,20] and rDNA sequences (Fig. 3a, Supplementary Fig. 9 and Supplementary Data 4). The repeat profile across the 35 chromosomes revealed prominent 2–3 Mb peaks of highly dense repeats probably corresponding to the centromeres (Fig. 3a).

To validate the DNA sequences associated with functional centromeres, we developed an *R. canina* centromeric histone H3 (CENH3)-specific antibody and performed chromatin immunoprecipitation followed by sequencing (ChIP–seq). Notably, our results revealed two main types of centromere composition—*Ty3/Gypsy ATHILA* and *CANR4* satellite-based centromeres (Fig. 3a,b). Analysis of the centromere-wide repeat structures revealed that *ATHILA*-based centromeres were most frequent in the chromosomes of the S1 and R4 subgenomes, while larger *CANR4*-based centromeres were found across all chromosomes in the S2 and R3 subgenomes (Fig. 3a–c and Supplementary Fig. 10). *CANR4* centromeric arrays were also found in 3 out of 7 syntenic groups (2, 4 and 5) of the S1_h1/h2 bivalent-forming chromosomes and in two R4 chromosomes, in which these arrays were frequently interrupted by *ATHILA* elements (Fig. 3a–c, Supplementary Fig. 10 and Supplementary Table 3). A similar centromeric sequence composition was observed in both *R. canina* and *R. agrestis* from DToL, despite considerable variation in sequence length (Supplementary Figs. 11 and 12 and Supplementary Data 5–8). Further structural sequence analysis of the diploid relatives *R. chinensis* and *R. rugosa* revealed that *CANR4* repeats are present in only four and three centromeric regions, respectively, while centrophilic *ATHILA* elements were found in all centromeres (Supplementary Data 9–11). Together, our results confirm the expansion and predominance of *CANR4*-based centromeres in exclusively maternally inherited univalent chromosomes in dogroses[20].

Moreover, we identified two centromeres of *R. canina* S27 from the R4 subgenome (Rca1_R4 and Rca4_R4) that lack *CANR4* repeats but exhibited high affinity for CENH3 in regions other than *ATHILA* elements. These two centromeres were characterized by the presence of several tandem-repeat sequences with very long monomers ranging from 1,425 to 2,596 bp. Detailed characterization of these tandem-repeat arrays has revealed that all of these sequences identified are probably derived from different centrophilic *ATHILA* elements, as they all share over 75% similarity with their LTR sequences and are therefore referred to *cenLTR1–4* (Extended Data Fig. 6 and Supplementary Fig. 13). Notably, the *cenLTR* arrays showed significantly higher CENH3 enrichment compared with neighbouring *ATHILA* elements, with the most pronounced enrichment observed in Rca1_R4, which contained a large array of *cenLTR1* (235 kb; Extended Data Fig. 6). Although *cenLTR* arrays found in Rca4_R4 were shorter and characterized by less CENH3 enrichment compared with *cenLTR1* in Rca1_R4, we found two different arrays of *cenLTR2* and *cenLTR3* with higher enrichment than neighbouring *ATHILAs*. Furthermore, the *cenLTR1* monomer sequence showed over 85% similarity to the LTR sequences of *ATHILAs* in Rca4_R4; however, it was not found in tandem arrays in this chromosome (Supplementary Fig. 13). Notably, these *cenLTR* arrays were not detected in either of the DToL genome assemblies of *R. canina* or *R. agrestis* (Supplementary Figs. 11 and 12), suggesting that the formation of these centromeric tandem repeat arrays is a very recent evolutionary event.

To investigate the epigenetic organization within centromeres, we analysed DNA methylation patterns across chromosome arms, scaling from the telomeres to the centromere midpoints. Methylation levels

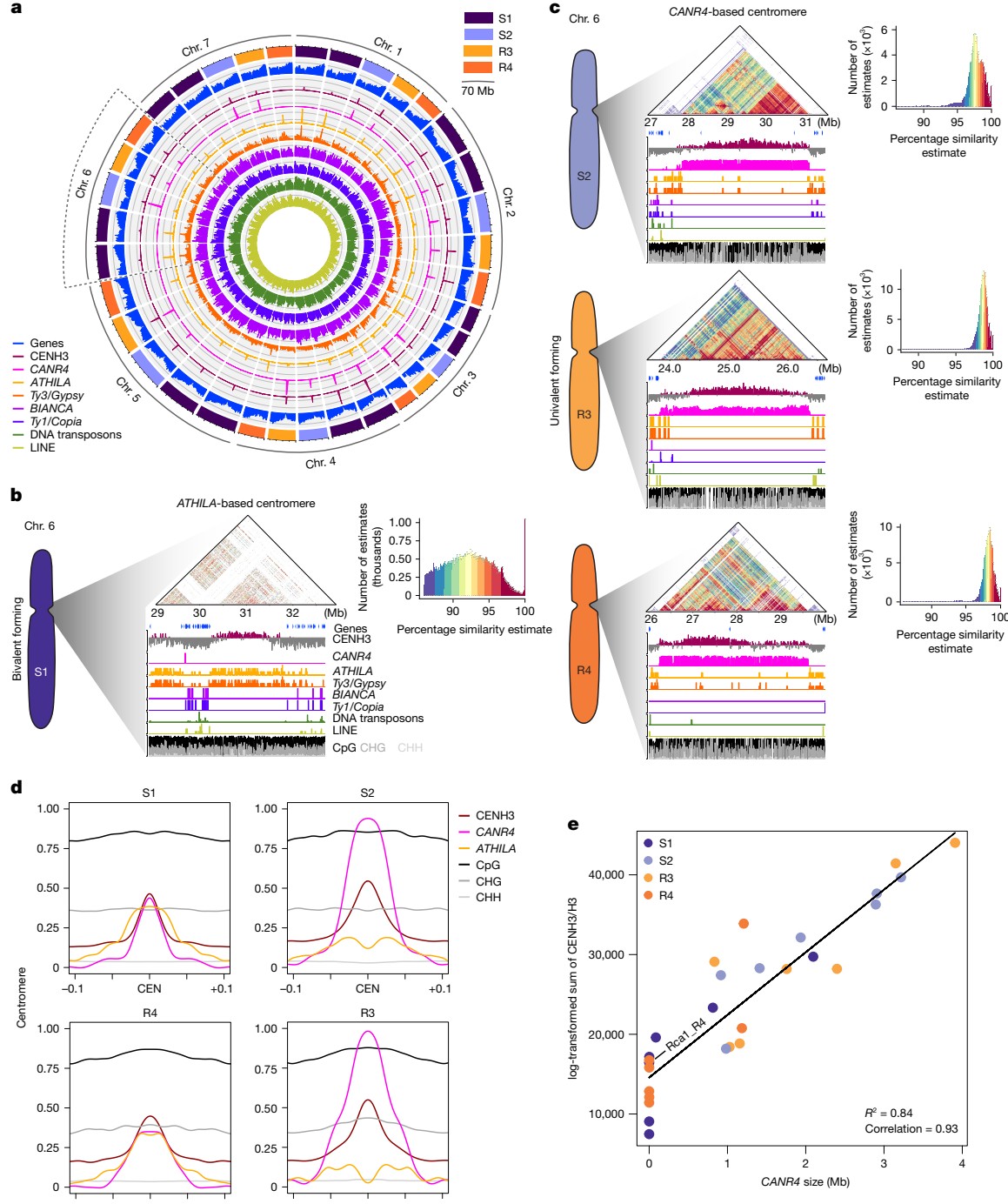

**Fig. 3 | The bimodal sequence composition of *R. canina* centromeres.**
**a**, The global distribution of the main types of repeats identified across all
chromosomes. Subgenomes (top right) and sequence tracks (bottom left) are
assigned by coloured names. Window size, 100 kb. **b**,**c**, Centromere analysis of
tandem-repeat structures in the chromosomes of synteny group 6 (additional
detailed plots for the other chromosomes are provided in Supplementary
Fig. 10). The sequence structure of bivalent (**b**) and univalent (**c**) centromeres
was visualized using ModDotPlot (top triangles). The colour-intensity histograms
(top right) show the number of alignments versus pairwise sequence similarity.
The sequence tracks plotted below highlight the main classes of repeats
identified and the respective association with CENH3 and DNA methylation.
The window size in **b** and **c** is 10 kb. For the *y* axes in **a**–**c**, all features were scaled
[0, 1]; the original values are provided in Supplementary Fig. 10. **d**, CENH3
ChIP–seq enrichment (log$_2$[CENH3/H3]) compared with the densities of the

centromeric elements *ATHILA* and *CANR4* in 50 kb windows, and DNA
methylation in the CpG, CHG and CHH sequence contexts, for each subgenome.
Only the centromere proximity regions are shown—10% of the centromere-to-
telomere distance. Centromeres (CEN) were defined by the maximum CENH3
enrichment. All signal values (*y* axis) were scaled from 0 to 1 based on the global
minimum to global maximum, except for DNA methylation, for which the
original percentage values were retained. **e**, Linear regression of *CANR4* size
and CENH3 abundance on the centromere across all chromosomes. Each dot
coloured by its subgenome presents a centromere. The abundance of CENH3
was calculated by the sum of CENH3 ChIP–seq (log$_2$[CENH3/H3]) signals on
centromeres normalized to coverage. The *R*$^2$ of the linear regression model is
0.84. The Spearman's rank correlation is 0.93. Note the high CENH3 enrichment
for the *cenLTR1*-based centromere in Rca1_R4, which lacks *CANR4* repeats.
Source data are provided in Supplementary Data 15.

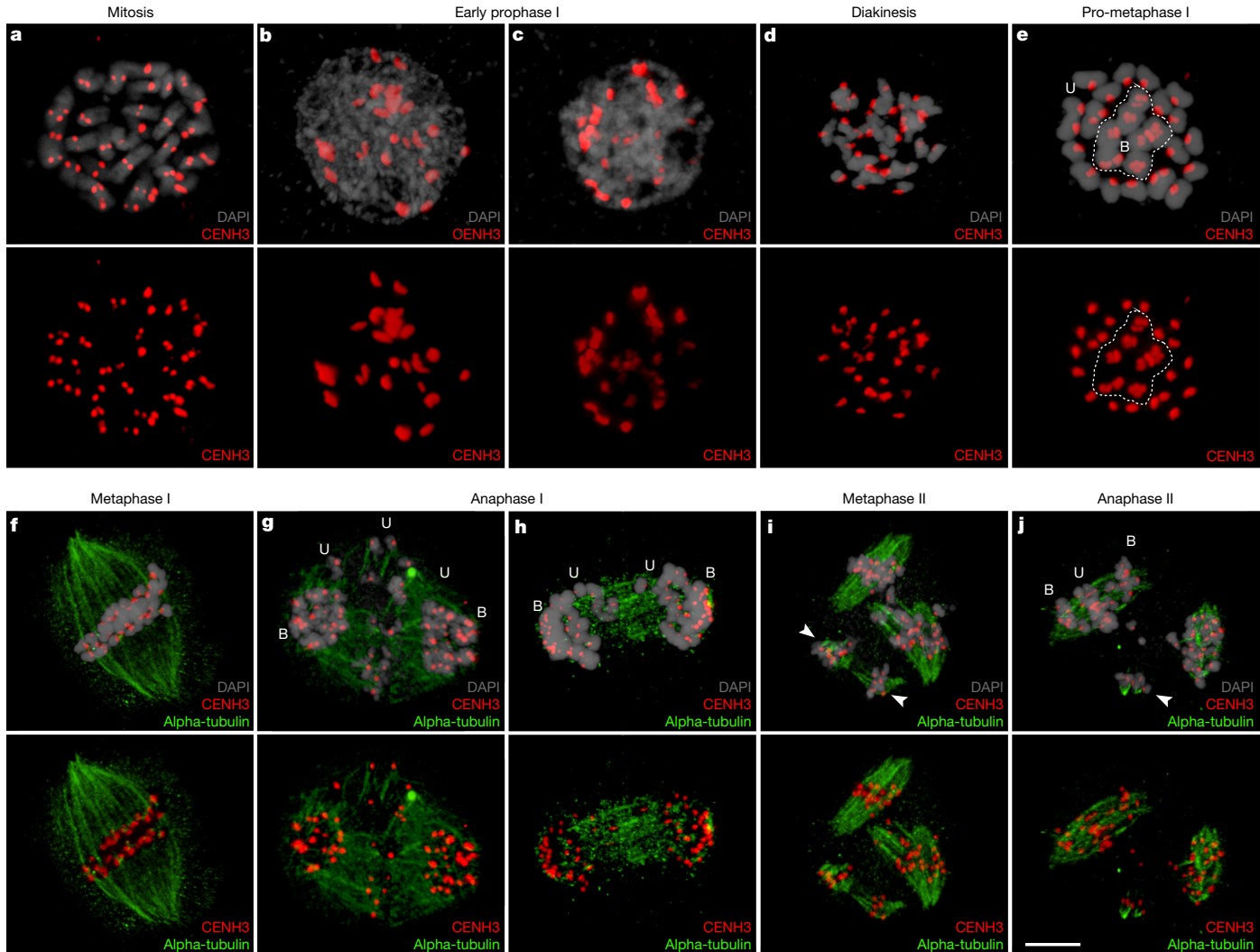

**Fig. 4 | Immunodetection of CENH3 and alpha-tubulin in mitotic and meiotic cells of *R. canina*. a**, Centromeres on all chromosomes were detected by CENH3 protein in mitotic metaphase. Note the size difference in CENH3 signals among different chromosomes. **b,c**, Centromeric organization during early prophase I. **d**, The orientation of centromeres during the diakinesis stage. **e**, Metaphase I with univalents (U) surrounding the bivalents (B); the dashed line highlights the typical clustering of bivalents in the middle. **f**, Metaphase I, with all chromosomes organized in the equatorial plate initiating the segregation of homologous pairs (in the case of bivalents) and separation of sister chromatids (in the case of univalents). **g,h**, Early (**g**) and late (**h**) anaphase I shows early separation of bivalents, while sister chromatids of univalents usually lagging behind. **i,j**, During metaphase II (**i**) and anaphase II (**j**), two main spindles are formed, while lagged chromosomes are still attached to additional abnormal spindles (arrowheads). At anaphase II (**j**), sister centromeres of the bivalent-forming chromosomes finally segregate to form haploid gametes, while single chromatids from univalents lag behind and are eliminated. Experiments were independently repeated at least ten times with similar results to track all meiotic stages represented here. Scale bar, 5 μm (**j**).

were generally elevated at centromeres across CpG, CHG and CHH contexts. However, DNA methylation was slightly reduced, particularly in *CANR4*-based centromeres (Fig. 3d, Extended Data Fig. 7 and Supplementary Fig. 14). These findings suggest distinct methylation patterns in *CANR4* centromeres compared with *ATHILA*-based centromeres. Indeed, *ATHILA* accumulation was less pronounced in the centromeres of univalents of S2 and R3 subgenomes, despite a few insertions being found within *CANR4* centromeric arrays (Fig. 3c and Supplementary Figs. 10 and 14). Notably, *ATHILA* insertions in *CANR4* arrays disrupted CENH3 binding, while *ATHILA*-based centromeres were smaller and showed a lower level of CENH3 association compared with *CANR4*-based centromeres (Fig. 3b–d, Extended Data Fig. 7 and Supplementary Figs. 10 and 14). Frequent insertion of *ATHILA* elements into *CANR4* arrays was also observed in *R. agrestis* and the diploid roses *R. chinensis* and *R. rugosa* (Supplementary Data 7–11). A similar disruption of centromere activity by centrophilic *ATHILA* has been recently found within *Arabidopsis* centromeres[46,47].

We further observed a positive correlation between the amount of *CANR4* repeats and CENH3 abundance along *R. canina* S27 centromeres (Fig. 3e). In fact, the total centromere length, as the measurement of the CENH3-binding regions per subgenome, confirmed that the S2 and R3 centromeres were larger compared with those of S1 and R4 (Fig. 3e, Supplementary Fig. 15 and Supplementary Data 5). Most *CANR4*-based centromeres showed increased levels of CENH3 accumulation compared with the *CANR4*-less ones. Notably, the centromere on Rca1_R4, which is mainly based on a *cenLTR1* array (Extended Data Fig. 6), showed one of the highest enrichments for CENH3 among *CANR4*-less centromeres (Fig. 3e). Thus, tandem repeats bearing different monomer composition seem to accumulate high CENH3 levels in the *R. canina* centromeres.

CENH3 immunostaining of *R. canina* chromosomes revealed differences in the size of individual centromeres (Fig. 4a and Supplementary Video 1; *n* = 12). The size difference was further confirmed by immunostaining analysis of the kinetochore component KNL1[48] (Extended

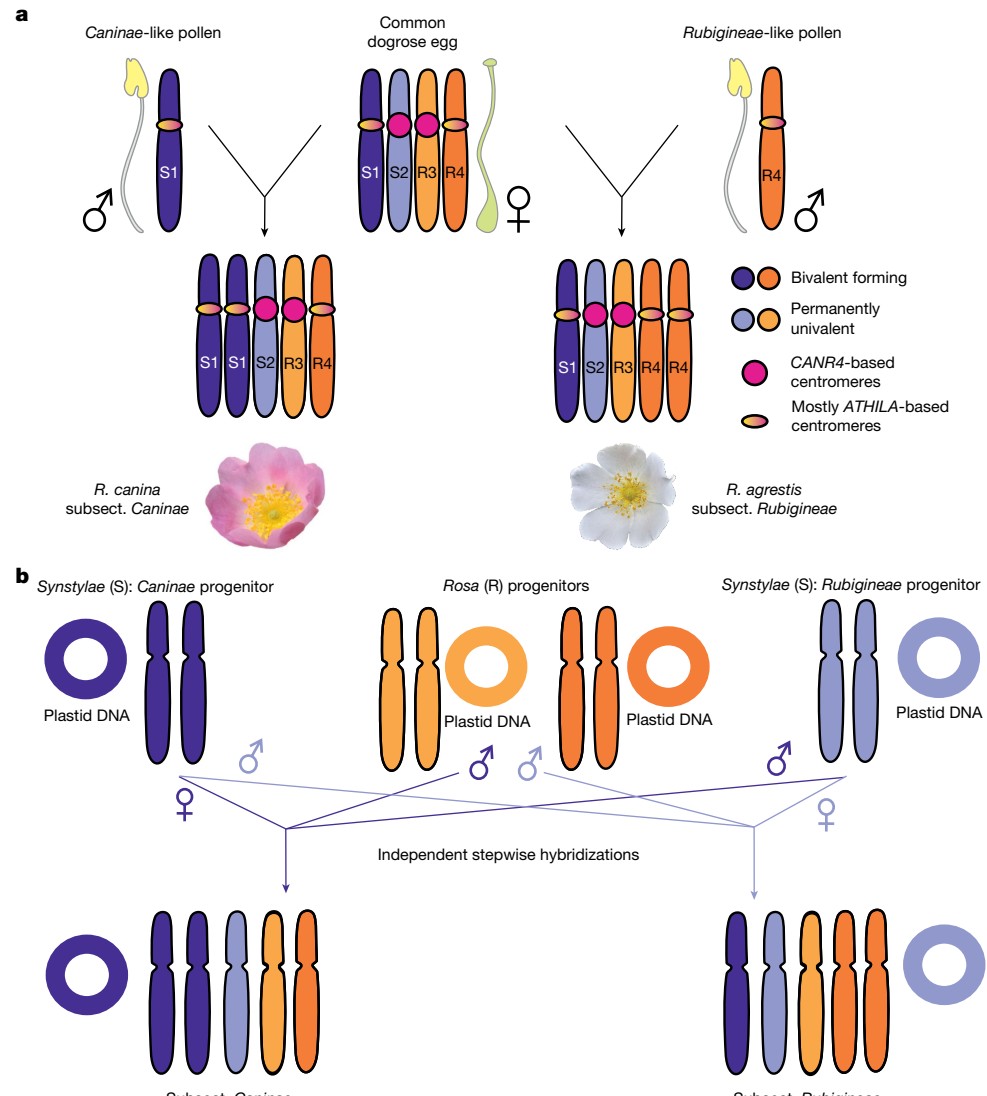

**Fig. 5 | Model for the origin and evolution of dogrose centromeres and genomes. a**, Reproduction mode and centromere evolution of the pentaploid dogroses. Dogroses of subsect. *Caninae* have two copies of the S1 subgenome, while dogroses of subsect. *Rubigineae* have two copies of the R4 subgenome. The S2 and R3 subgenomes are found as a single copy only. During meiosis in subsect. *Caninae*, the S1_h1/S1_h2 chromosomes form bivalents and are transmitted to both pollen and egg cells. The remaining univalent subgenomes (S2, R3 and R4) are transmitted by the egg cell only. By contrast, in subsect. *Rubigineae*, the R4_h1/R4_h2 chromosomes form bivalents and are transmitted to both pollen and egg cells, while the remaining univalent subgenomes S1, S2 and R3 are transmitted only by the egg cell. The analyses of centromeric

sequence composition revealed a dominance in *ATHILA* LTRs in most of the bivalent-forming centromeres of S1 and R4 subgenomes. By contrast, larger *CANR4*-based centromeres were found in all chromosomes of the permanently univalent S2 and R3 subgenomes. **b**, The model for the origin of dogrose subsections. On the basis of the findings that representatives of subsect. *Caninae* and subsect. *Rubigineae* contain phylogenetic distant plastids from the *Synstylae* clade[13,31–33] (Extended Data Fig. 5b), we propose that two progenitors of the *Synstylae* clade formed reciprocal hybrids and additionally incorporated R genomes through pollen donors. Subgenomes are represented by one chromosome. *Synstylae* subgenomes S1 (violet) and S2 (light blue) and *Rosa* clade subgenomes R3 (light orange) and R4 (dark orange) are shown.

Data Fig. 8a,b; *n* = 23). Furthermore, in situ hybridization with probes for *CANR4* and *ATHILA* on mitotic (*n* = 15) and meiotic (*n* = 18) chromosomes of *R. canina* confirmed the predominance of large *CANR4* signals at the centromeres of univalents, while all bivalent and almost all univalents showed *ATHILA* centromeric signals (Extended Data Fig. 8c–k). In the diploids *R. chinensis* and *R. rugosa*, *CANR4* centromeric signals were observed in only three pairs of chromosomes while, again, *ATHILA* was found in all centromeric regions (Extended Data Fig. 8l,m). Thus, the observed difference in the size of CENH3 centromeric signals is probably associated with the accumulation of *CANR4* satellite repeats, supporting our ChIP–seq analysis (Fig. 3e). These results confirm the bimodal architecture of *R. canina* centromeres, which are preferentially *ATHILA*-based in bivalents and *CANR4*-based in univalents,

with the caveat that our ChIP–seq experiment was performed using leaf tissue.

Next, we investigated the behaviour of centromeres and spindle dynamics during male meiosis of *R. canina*. Our immunostaining analysis using antibodies against CENH3 and alpha-tubulin clarified the asymmetric distribution of chromosomes during meiosis. In early stages of male meiosis, both small and large centromeres were visible (Fig. 4b–d and Supplementary Videos 2–4; *n* = 31). At onset of metaphase I, we observed seven bivalents organized at the centre of the cell surrounded by 21 univalents (Fig. 4e and Supplementary Video 5; *n* = 31), a configuration that was first proposed over a century ago[2,3,22,23]. During this stage, microtubules facilitated the separation of the homologous pairs through bipolar attachment, while univalents also showed bipolar

attachment (Fig. 4f and Supplementary Video 6; $n = 31$). In early and late anaphase I, homologous chromosomes migrated first, while single chromatids derived from univalents lagged behind (Fig. 4g,h and Supplementary Videos 7 and 8; $n = 31$). Notably, we observed two groups of univalents exhibiting different timing in sister-chromatid separation (Fig. 4g,h (arrows)). During metaphase and anaphase II, we frequently observed both normal and abnormal spindles, resulting from single chromatids lagging behind from anaphase I (Fig. 4i,j (arrowheads)). Homologous pairs derived from bivalents separated normally at end of anaphase II, forming haploid nuclei with seven chromosomes, while single chromatids from univalents lagged behind and were probably eliminated (Fig. 4i,j and Supplementary Videos 9 and 10; $n = 31$). Despite the apparent irregularities in male meiosis, viable pollen grains in *R. canina* (S27) were produced at a rate of approximately 20% (Supplementary Fig. 16a,b). Notably, while at the end of meiosis polyads are formed showing nuclei with varying number of centromere foci (Extended Data Fig. 9 and Supplementary Video 11), the mature binucleate pollen grains seem to contain only a haploid vegetative ($1n = 1x = 7$ chromosomes) and generative ($2n$) nucleus, as confirmed by their genome size and composition (Fig. 2a and Supplementary Fig. 16c). This suggests that, despite meiotic irregularities, a selective mechanism ensures the formation of haploid pollen. Our findings highlight how the atypical centromere behaviour and spindle dynamics during male meiosis in *R. canina* deviates from canonical meiotic processes but can still result in viable pollen production.

## Discussion

By leveraging newly generated genome assemblies of dogroses, we shed light onto the long-standing century-old mystery of the unique *Canina* meiosis[2,3,22,23]. Through structural analysis of subgenomes and centromeres, combined with pollen-derived genomics and hybridization experiments, we demonstrate that the bivalent-forming subgenomes in dogroses evolved independently and exhibit distinct interaction patterns during meiosis (Fig. 5).

The bimodal architecture of centromeres in *R. canina* is particularly intriguing when considered alongside its asymmetric female meiosis, in which univalent chromosomes are obligatorily transmitted through the egg cell[3,49]. The prevalence of *CANR4* repeats in univalent centromeres could possibly link centromere expansion with their drive in female meiosis (Fig. 5a), a phenomenon in which larger centromeres are preferentially transmitted during meiosis[37,50,51]. The structural divergence and selective enrichment of *CANR4* repeats in the centromeres of univalents may underpin their larger size, ensuring the preferential transmission of univalents through the egg cell and, therefore, maintaining the pentaploid genome structure. This may represent a rare case of an obligate drive mechanism, functioning in a 'drive or die' manner to maximize the transmission of univalents. However, the occasional presence of *ATHILA*-based centromeres in some univalent chromosomes suggests that *CANR4* expansion alone does not fully explain univalent drive. Furthermore, the structural divergence of centromeres seems also to influence their behaviour in male meiosis, as large *CANR4*-based centromeres in univalents could possibly promote bipolar orientation and premature chromatid separation in male meiosis. Notably, in *Arabidopsis thaliana*, bipolar orientation of univalents happens only when sister chromatid cohesion is defective[52], but it appears to be more common in wheat univalents[53]. This observation contrasts with that of female meiosis, in which univalents seem to have monopolar orientation in dogroses[3] (Supplementary Fig. 17). Thus, a potential role for sexual dimorphism in sister chromatid cohesion regulation could be part of the adaptations enabling *Canina* meiosis.

It is possible that the absence of a homologous pair (and therefore a competing centromere) in the obligatory univalents (S2 and R3 subgenomes) may facilitate the expansion of *CANR4* repeats. By contrast, the S1 and R4 centromeres, which exist within a competitive pairing environment, experience counterbalancing forces that limit *CANR4* accumulation (Fig. 5a). This is further supported by the absence of solely *CANR4*-based centromeres in diploid roses *R. chinensis* and *R. rugosa*. Furthermore, we provide strong evidence for the emergence of few tandem repeats originating from LTR sequences, which outcompete neighbouring *ATHILA* elements for CENH3 binding in two R4 univalent chromosomes (Rca1_R4 and Rca4_R4). These findings highlight the higher affinity of tandem repeats for centromere function in dogroses, further emphasizing the role of centromere composition in shaping meiotic behaviour. However, while the observed correlation is intriguing, we acknowledge that future studies will be essential to confirm whether the expansion of *CANR4* in univalent centromeres is directly linked to their drive during female meiosis.

Despite the lack of recombination, univalent chromosomes retain functional protein-coding genes, as evidenced by high BUSCO completeness and the absence of differential selection pressures. This supports a relatively recent origin of modern dogroses and highlights the resilience of their polyploid genome. Phylogenetic analyses using pollen SCOs and maternally inherited plastid markers further corroborate the polyphyletic origin of dogroses[13,31–33] with multiple hybridization events contributing to their evolutionary history (Fig. 5b). The distinct subgenome ratios observed in *R. canina* (3:2 *Synstylae* to *Rosa*) and *R. agrestis* (2:3 *Synstylae* to *Rosa*) underscore the complexity of their hybrid origins[30,32]. Pollen SCO data align with cytogenetic studies, indicating that bivalent-forming subgenomes in the subsections *Caninae* and *Rubigineae* are phylogenetically distant[34,35]. Moreover, multiple origins for *R. canina* are suggested, as the S1 subgenome clusters with five *R. canina* individuals, while the S2 subgenome clustered with another individual. This model aligns with reports of *Canina*-like meiosis arising spontaneously in hybrids of diploid sexual *Synstylae* species[54], further highlighting the complex hybrid origin and evolutionary dynamics of dogroses.

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

## Methods

### Plant material

For genome sequencing, we used the same individual of *R. canina* (S27), which has already been cytogenetically analysed[34] (voucher: GLM12396) from a natural stand (WGS84: 51.1732° N; 14.6271° E; Weißenberg, Saxony, Germany). A vegetative runner was dug on 28 March 2022 and planted in a pot. Clones of the collected plant specimen were cultivated in a greenhouse at the Max Planck Institute for Plant Breeding Research, Cologne, Germany.

### Whole-genome sequencing

High molecular mass genomic DNA was isolated from leaves using the NucleoBond HMW DNA Kit (Macherey Nagel). A HiFi library was prepared according to the manual of the HiFi SMRTbell Libraries using SMRTbell Express Template Prep Kit 2.0 (Pacific Biosciences) with initial DNA fragmentation using Diagenode Megaruptor 3 and final library size binning into defined fractions by Blue Pippin with 0.75% agarose cassettes (Sage Science). The size distribution was again controlled by a Femto pulse system (Agilent). Size-selected libraries were then sequenced on the Sequel II device with a Binding Kit 2.0 and Sequel II Sequencing Kit 2.0 for 30 h using two SMRT cells (Pacific Biosciences). Moreover, a chromatin-capture library was prepared from 0.5 g of fresh-weight leaf material input. All treatments were performed according to the recommendations of the Dovetail Omni-C kit for plants (Dovetail Genomics). As a final step, an Illumina-compatible library was prepared (Dovetail) and paired-end 2 × 150 bp deep-sequenced on the HiSeq 3000 (Illumina) device. All libraries were sequenced at the Max Planck Genome Centre Cologne at the Max Planck Institute for Plant Breeding Research.

### Genome assembly

A phased chromosome-level genome was assembled using the generated PacBio HiFi and Hi-C data. First, a phased primary assembly was obtained by running Hifiasm[55] using 50 Gb of PacBio HiFi reads in combination with Dovetail Omni-C reads with the following command: hifiasm -o out.phased.asm.hic --h1 hic.R1.fastq.gz --h2 hic.R2.fastq.gz hifi.reads.fastq.gz. In the default diploid mode, we generated two sets of phased contigs. Each set was further scaffolded to the chromosome scale using Salsa2[56], followed by successive rounds of manual curation and rescaffolding. We then identified 14 and 21 pseudochromosomes, respectively.

We used Benchmarking Universal Single-Copy Orthologues (BUSCO, v.5.4.0)[57] to evaluate the completeness of 35 chromosome-level scaffolds and for each of the four subgenomes. The lineage database used for running BUSCO was eudicots_odb10. The protein sequences were converted from the assembly using the GFF analysis toolkit AGAT: agat_sp_extract_sequences.pl -g annotation.gff -f genome.fasta -p --type cds -o protein.fasta. The *k*-mer-based tool Merqury (v.1.3)[58] was used to estimate both the completeness and base quality of the chromosome assembly. The quality value of the chromosome assembly was greater than 66.6, and the quality value of each chromosome was at least 62. Read *k*-mers were built from HiFi sequences by Meryl (v1.3) with a *k*-mer size of 31 bp.

**Rca_S2 assembly correction.** During the scaffolding step, we noted the absence of approximately 20 Mb (including the centromere) on chromosome Rca2_S1_h2. Further validation using fluorescence in situ hybridization (FISH) showed that this chromosome should indeed have a large array of *CANR4*, as found in its homologous chromosome Rca2_S1_h1 (Supplementary Fig. 1). We noted that this assembly error was probably generated by the presence of a small translocation found at the start of the missing region in Rca_S1_h2 compared with Rca_S1_h1. We further mapped the HiFi reads to the region present in Rca_S1_h1 and found robust evidence for the presence of five copies of this region in the genome. We then concluded that this region was incorrectly missing from the Hifiasm assembly due to its high degree of similarity

between the S1_h1 and S1_h2 haplotypes. We therefore duplicated the Rca2_S1_h1 region (37265454–54065178 bp) and manually assigned it to the expected position on Rca_S1_h2 (from Rca2_S1_h2: 33518152 bp).

**Assembly of DToL datasets.** We also downloaded available data from the Darwin Tree of Life (DToL) project for another accession of *R. canina* (PRJEB79801) and for *R. agrestis* (PRJEB79880) and performed phased chromosome-level genome assemblies as described above. *R. agrestis* from DToL revealed two copies of the R4 subgenome; our previous studies suggested that some accessions of *R. agrestis* were of hybrid origin, which should then have two copies of S1 subgenome[45,59].

### *k*-mer analysis for genome size and ploidy level estimation

*k*-mer analysis to estimate genome size was performed using jellyfish (v.2.3.0)[60] and Genomescope (v.2.0)[61]. The pentaploidy of *R. canina* was further confirmed and analysed using Smudgeplot (v.0.2.5)[61].

### Chloroplast genome assembly and phylogeny

To clarify the maternal lineage of the allopolyploid *R. canina* (S27), we assembled the plastid genome of the sequenced individual. GetOrganelle (v.1.7.7.0)[62] was used to de novo assemble the first draft of the plastid genome using 2× 150 bp Illumina short-read data (Sequence Read Archive (SRA): ERS1370372). This toolkit implements Bowtie 2[63] to initially find reads mapped to a plant chloroplast database and SPAdes[64] for de novo assembly and iterative extension. During the assembly and iteration process, BLAST+[65] was used to identify off-target contigs, which were then removed or trimmed. The resulting plastid genome was then used as a reference for mapping the original reads back using Geneious Prime v.2023.2.1 (Biomatters), allowing only mapping of paired reads mapped nearby with a minimum overlap of 75 bp and a minimum overlap identity of 98%. The results were manually examined and corrected where necessary.

The initial annotation of the chloroplast genome was performed using GeSeq (v.2.03)[66]. The annotation included the chloroplast inverted repeats (IRs), *rps12* interspersed gene, protein-coding sequences, tRNAs and rRNAs using 55% identity as thresholds for annotation of proteins and 90% for DNA as well as RNAs. Furthermore, tRNAscan-SE (v.2.0.7)[67] and Chloë (v.0.1.0)[68] were used as additional annotators within GeSeq. The annotations were manually edited using Geneious Prime v.2023.2.1 (Biomatters). The presence of chloroplast genomes differing in the orientation of the single-copy units (large single-copy (LSC) region, small single-copy (SSC) region) was checked by selecting motifs from the border region of the IR and the single-copy units (LSC-trnH-GUG 5′-GGTTCAATTCCCGTCGTTC-3′ or LSC-rps19 5′-GTGACACGTTCACTGAAAAAA-3′ and IRb-rps19-rpl2-IGS 5′-AGACGAAGAAACAAATTCTAT-3′; SSC-ndhF 5′-TGTAAT AATATAATAATTGAA-3′ or SSC-ycf1 5′-CGACCCTAAACGATGGAATCG-3′ and IRa-ycf1 5′-TTGAAAAACCCGTTGTAACTAT-3′), noting their relative orientation to each other on the same reads using SeqKit (v.2.6.1)[69].

The assembled *R. canina* chloroplast genome had a length of 156,650 bp and a classical quadripartite structure (Supplementary Fig. 18): a LSC of 85,634 bp (~56.57% of the plastid genome), a SSC of 18,878 bp (~12.05%) and two IR regions of 26,069 bp (~16.64% each). Different isomers were found to differ in the orientation of the SSC and LSC (flip-flop configuration).

We computed a chloroplast phylogeny using 37 samples, including sequences downloaded from GenBank and newly assembled data (Supplementary Data 17). The alignment was performed with MAFFT[70], and the phylogenetic tree was calculated using IQ-TREE[71] with the following settings: -m MFP --con-tree --burnin 250 -B 1000 -T 36 --wbtl.

### Identification of the bivalent-forming subgenome and comparative analysis

The assembled chromosomes were subjected to pairwise comparisons presented as dot plots using the Synteny Mapping and Analysis (SyMAP)

tool[72]. Multiple alignments within the synteny groups comprising five *R. canina* chromosomes plus *R. chinensis* and *R. rugosa* assemblies were carried out in the CLC Genomics Workbench (CLC) using the 'whole genome alignment' plugin with the following parameters: minimum initial seed length of 250; minimum alignment block length of 250. The aligned chromosomes were subjected to pairwise comparisons. The similarity values were calculated as the block fraction of the two genomes that were aligned (that is, the alignment percentage) or as the percentage of exactly matching nucleotides within the aligned blocks (the average nucleotide identity).

Multifasta files containing assembled short-read sequences of pollen SCO loci from eight different dogrose individuals (three samples of *R. canina*, two samples of *R. corymbifera* (subsect. *Caninae*) and three samples of *R. rubiginosa* (subsect. *Rubigineae*)) that were not sequenced by long reads (Supplementary Data 17) were used as queries to map them with the software BWA[73] with the aln command to the *R. canina* chromosome assembly. From the sequence alignment map (.sam) file, those chromosome hits with only one alternative were filtered according to the 'XA:Z:' flag using a Python script written by GPT-4 (ChatGPT Plus, OpenAI). A bubble map displaying the mean counts of chromosome pairs within different subsections was drawn with ggplot2[74].

### Synteny analysis

Chromosome synteny was analysed with the Synteny and Rearrangement Identifier (SyRI)[75]. For this purpose, chromosomes of subgenomes S1_h1, S1_h2, S3, R3 and R4 were aligned against each other within each linkage group (Rca1–Rca7) by minimap2[76,77] using the following command: minimap2 -ax asm5 --eqx -t 16 genome1.fa genome2.fa | samtools sort -@ 8 > aln.sorted.bam. Moreover, subgenome S1 was also aligned with *R. chinensis* (NCBI: GCA_041222415)[16] and *R. canina* subgenome R4 was aligned with *R. rugosa* (NCBI: GCA_958449725.1; https://www.darwintreeoflife.org/) to analyse its synteny. To keep all of the chromosomes arranged in the same order as *R. canina* and for better visualization, chromosomes 2, 5 and 7 of *R. chinensis* were inverted, and the chromosomes of *R. rugosa* were reordered to 6, 1 (inverted), 7, 5 (inverted), 2, 3 and 4, corresponding to chromosomes 1–7 in *R. canina*. SyRI was implemented for all of the aligned genome pairs using the following command: syri -c aln.sorted.bam -r genome1.fa -q genome2.fa -k -F B --nc 16. Visualization revealed only syntenic blocks over 50 kb, which was performed by plots: Python PLOTsr --sr rugosa_R4/syri.out --sr R4_R5/syri.out --sr R5_s3/syri.out --sr S3_S2/syri.out --sr S2_S1/syri.out --sr S1_chinensis/syri.out --genomes genomes.txt -o out_50k.pdf -S 0.7 -W 10 -H 9 -f 10 --itx -s 50000.

Syntenic orthologues among the primary annotations of diploid strawberry *Fragaria nilgerrensis*[78], *R. chinensis*[16], *R. rugosa* (GCA_958449725.1, DToL; https://www.darwintreeoflife.org/) and the five sets of chromosomes of *R. canina* were inferred using GENESPACE (v.1.2.3)[79] pipeline with the default parameters. In brief, GENESPACE compares protein similarity scores into syntenic blocks using MCScanX[80] and Orthofinder (v.2.5.4)[81] to search for orthologues/paralogues within synteny constrained blocks. Syntenic blocks were used to query pairwise peptide differences among progenitor alleles, determine divergence among progenitor orthologues using *R. chinensis* syntenic anchors and search for specific orthogroups.

### Self-synteny and fractionation bias

Synteny information was obtained using the SynMap tool on the CoGe platform[82,83]. Only genes within synteny blocks were considered, including not only gene pairs but also singleton genes in each genome that lost their counterpart in the other genome due to fractionation or other gene loss. The identification of syntelogues between species was performed using SynMap2 (https://genomevolution.org/wiki/index.php/SynMap2), which internally uses LAST for sequence alignments[84], and then fractionation bias was analysed with FractBias[85].

### d*N*/d*S* analysis

Protein-coding sequences (CDSs) were extracted for each *R. canina* subgenome according to coordinates from the gene structural annotation file using GffRead (v.0.12.6)[86] and translated into amino acid sequences using the transeq command from EMBOSS[87]. Additional amino acid sequences and CDSs of *R. rugosa* (BioProject: PRJNA1061178; https://www.darwintreeoflife.org/), *R. chinensis* (NCBI: GCF_002994745.2)[14] and *Fragaria vesca* (NCBI: GCF_000184155.1)[88] were downloaded. The CDS and amino acid sequences were validated, for example, for correct start codons or methionine as the first amino acid in the proteins, using Python scripts. The confirmed proteomes were subsequently analysed using OrthoFinder[81] to identify common single-copy orthologues. According to the protein IDs, FASTA files for each orthologue gene containing five proteins of *R. canina* together with the three of outgroups were aligned with MAFFT (v.7.490)[70]. On the basis of the aligned proteins, corresponding CDSs were codon based aligned using PAL2NAL[89] and DNA alignments were transformed into PHYLIP format. The PAML pipeline[90] with yn00 was used in a looped pairwise mode over all PHYLIP files for each subgenome and outgroup to estimate the nonsynonymous (d*N*) and synonymous (d*S*) substitution rates, as well as their ratio (d*N*/d*S* = ω). The results based on the Yang–Nielsen[91] method were extracted from PAML output files and combined using a Python script and graphical visualized with ggplot2[74] in the R environment[92]. To covert the relative evolutionary time (Time$_t$) from yn00 into absolute divergence time ($T_{Ma}$) in millions of years ago (Ma), we used *F. vesca* as a fixed calibration point with its fossil record of 2.96 Ma (refs. 93,94). The relative divergence time Time$_t$ for each pairwise compared gene was multiplied with a scaling factor as follows:

$$T_{Ma} = Time_t \times \left( \frac{2.96}{\text{Mean time}_t \text{ for } F. \text{ vesca versus } Rosa} \right)$$

All scripts were developed with the help of ChatGPT-4o (ChatGPT Plus, OpenAI).

### Chromosome-level phylogenetic reconstruction

We first generated whole-chromosome multiple alignments in synteny groups 1–7 using the Whole Genome Alignment tool in ClC workbench (Qiagen). The algorithm identifies seeds, that is, short stretches of nucleotide sequences that are shared between multiple genomes but not present multiple times in the same genome. These seeds were then extended using a HOXD scoring matrix, and the HOXD substitution score was combined with an adjustment term based on *k*-mer frequency to avoid spurious matches to repetitive regions in the genome[95]. The program parameters were as follows: minimum initial seed length, 250; minimum alignment blocks, 250; and mismatches in seeds, allowed. The chromosome phylogenies were constructed from multiple alignments using RAxML (v.8.2.12)[96] with the GTRGAMMAI model. The diploid accessions were chromosome assemblies of *R. chinensis*[16] and *R. rugosa*. For *R. rugosa*, the original chromosomes were renamed to fit the *R. chinensis* synteny.

### Subgenome-aware phasing of *R. canina*

We used SubPhaser[41] (default parameters) to phase and partition the subgenomes of the pentaploid *R. canina* and *R. agrestis* by assigning chromosomes to subgenomes based on differential repetitive *k*-mers. These were assumed to have expanded during the period of independent evolution after divergence from the nearest common ancestor and before the stepwise hybridization events (the divergence–hybridization period). A subgenome is considered to be well phased when it displays distinct patterns of both differential *k*-mers and homoeologous chromosomes, confirming the presence of subgenome-specific features, as expected. As the S1_h1 and S1_h2 chromosomes represent

haplotypes of the S1 genome, only the S1_h1 haplotype was used in the phasing analysis together with the sets of S2, R3 and R4 chromosomes.

LTR insertion times were calculated by Subphaser as follows: LTR-TRs were de novo detected using LTRharvest (v.1.6.1)[97] and LTRfinder (v.1.07)[98]. To reduce false positives, TEsorter (v.1.3.0)[99] was used to reconstruct the classification of LTR-RTs and further refine this classification. The subgenome-specific $k$-mer sequences were mapped to the LTR-RT sequences using a substring match procedure to identify the subgenome-specific LTR-RTs using the Fisher's exact test. Two LTRs of each subgenome-specific LTR-RT were retrieved and the nucleotide divergence was estimated using the Jukes–Cantor 1969 model. The insertion time ($T$) was calculated using the equation $T = K/2r$, where $r = 1.3 \times 10^{-8}$ substitutions per year (default)[100], and $K$ represents the divergence of the LTRs from the LTR-RT.

## Flow cytometric determination of the endosperm/embryo ratio

To isolate the nuclei from embryo and endosperm tissue, nutlets from fruits of the sequenced individual of $R. canina$ S27 (voucher: GLM12396) were first cracked with pliers. The embryo and endosperm were then carefully transferred into a droplet of nuclei isolation buffer (CyStain PI Absolute P; Sysmex-Partec) in a Petri dish and chopped with a sharp razorblade. After adding additional nuclei isolation buffer to a final volume of 500 μl, the nuclei suspensions were filtered through 50 μm disposable filters (CellTrics, Sysmex-Partec), stained with 4′,6-diamidino-2-phenylindole (DAPI) to a final concentration of 1.5 μg ml$^{-1}$ and stored on ice until use. The measurements were performed on a CyFlow Space flow cytometer (Sysmex-Partec) equipped with a high-power UV LED (365 nm).

## SCOs

**Plant material.** To analyse the phylogenetic origin of the subgenomes of allopentaploid $R. canina$, we sampled 30 rose individuals of 24 diploid species across the genus $Rosa$. Thus, $R. stellata$ (subgen. $Hesperhodos$), 10 species from sect. $Synstylae$, seven from sect. $Rosa$ ($Cinnamomeae$, including sect. $Carolinae$), four from sect. $Pimpinellifoliae$ and one individual of $R. chinensis$ (sect. $Chinensis$) were sampled from the living collection of the Europa-Rosarium Sangerhausen. Moreover, one accession of $R. majalis$ (sect. $Rosa$) was collected from the Botanical Garden Würzburg, and $R. persica$ (subg. $Hulthemia$) from Botanical Garden Jena. Species were rechecked using their respective floras[101–103], and the material was compared with available herbarium specimens available online (JSTOR Global Plants, https://plants.jstor.org/; Moscow Digital Herbarium, https://plant.depo.msu.ru). Herbarium vouchers were deposited in the Herbarium Senckenbergianum Görlitz (GLM; Supplementary Data 17).

To determine bivalent-forming genomes, we sampled pollen from several individuals of $Rosa$ sect. $Caninae$ (subsect. $Caninae$: three $5x R. canina$, two $5x R. corymbifera$; subsect. $Vestitae$: one $5x R. pseudoscabriuscula$; subsect. $Rubigineae$: three $5x R. rubiginosa$; Supplementary Data 17). We collected anthers from 50 to 100 freshly opened flowers under dry weather conditions in early May 2021 in the field, stored them in open glass for 1 day to allow the anthers to open and subsequently transferred them to a 50 ml tube. Owing to electrostatic attraction, the pollen deposited on the walls of the tube. Anthers were carefully removed, and the pollen powder was collected at the bottom of the tube by gentle centrifugation. The pollen powder was then tapped out over clean paper and transferred to tubes with the help of a spatula. This procedure was repeated three times. Pollen grains were stored in a refrigerator until use.

**Isolation and flow sorting of pollen nuclei.** Nuclei of mature pollen grains were isolated by the filter bursting method[104] using the nuclear isolation buffer as described previously[105]. Pollen grains were burst on the surface of a 20 μm disposable CellTrics filter (Sysmex-Partec). The resulting nuclear suspension was stained with propidium iodide

(50 μg ml$^{-1}$, PI) and run on a BD Influx cell sorter (BD Biosciences). Nuclear populations were identified in a dot plot showing the PI fluorescence signal (log scale) versus side scatter signal (SSC, log-scale). A sort gate was defined based on the corresponding fluorescence intensity (lin-scale) histogram. A total of 200,000 individual generative nuclei (volume, around 400 μl) were collected into a 1.5 ml reaction tube using the '1.0 Drop Pure' sorting mode of the BD FACS software (BD Biosciences). After adding 50 μl of 1× TE and 50 μl of NaN$_3$, the nuclei were sedimented by centrifugation (1,000$g$ for 10 min at 4 °C). Next, 300 μl of the supernatant was removed, and the nuclei with the remaining liquid were stored at −20 °C. The gating strategy to isolate generative nuclei of $R. canina$ is presented in Supplementary Fig. 19.

**DNA extraction.** DNA from diploid rose species was first extracted from 20 mg of silica-dried leaf tissue according to the ATMAB protocol[106] and subsequently purified using the Mag-Bind Total Pure NGS Kit (Omega Bio-Tek, Nocross) according to the manufacturer's manual. DNA from flow-sorted pollen nuclei was extracted using the Mag-Bind Plant DNA DS Kit with the modification that permanent but careful mixing was performed during binding and elution because the DNA quantities ranged from 37 ng to 236 ng. The DNA yield was quantified using the Qubit 4 Fluorometer (Thermo Fisher Scientific).

**Target construction.** To analyse nuclear single-copy regions in rose genomes, we used published SCO tags[107]. The SCO tags were originally developed to be amplifiable by PCR and covered coding as well as non-coding regions. We used the 29,000 sequences from additional file 3 from ref. 107, which consisted of SCO tags of 17 rose species and seven outgroup species of the Rosaceae family. These sequences were filtered for uniqueness so that duplicates were removed and searched with BLAST in the $R. chinensis$ haploid line genome (v.1.0)[15]. Owing to the structural gene model annotation of the $R. chinensis$ genome, we were able to identify 923 full-length nuclear genes with single-copy characteristics. The target-capturing baits were designed by the Agilent bioinformatics service (I. Kisakesen, Agilent Technologies) and covered exons + UTRs with flanking regions and small introns of the selected genes in the $R. chinensis$ genome. Finally, the target consisted of 5,794 sequences of different lengths (the shortest at 179 bp and the longest at 6,544 bp) named according to $R. chinensis$ gene prediction and had a total size of 2 Mb (Supplementary Data 12). All target sequences were covered by 2× tiling with a total of 85,670 specific baits.

**Sequencing.** For target enrichment, we used the SureSelect XT HS2 DNA system with precapture pooling (Agilent Technologies) and target design as described above. For diploid roses, 200 ng of input DNA was used, and for pollen DNA, 36–200 ng of input DNA was sheared with a Bioruptor Pico sonication device (Diagenode) to a recommended fragment size of 180–250 bp. The Illumina short-read libraries were amplified for 9 cycles after adapter ligation, pooled for precapture to 16 samples and then postcapture library pools were amplified again with 12 cycles of PCR amplification. The library pools were sequenced in 150 bp paired-end mode on the Illumina NovaSeq 6000 system by Novogene with approximately 1 GB of data output per sample.

To analyse the ploidy of the samples, in vitro flow cytometry was performed on silica-dried leaflets according to a protocol described previously[45] using $R. arvensis$ ($2n = 2x = 14$) as an internal standard. The fluorescence intensity was measured using the CyFlow Ploidy Analyser (Sysmex Partec), and the data were analysed using Flowing Software v.2.5.1 (Turku Bioscience Centre). Each sample was measured three times with a minimum of 3,000 particles.

To estimate the ploidy of the samples in silico, we used K-Mer Counter (KMC) (v.3.1.1)[108,109] to generate a $k$-mer database from FASTQ sequence files containing short-read data covering SCOs. The setting was a $k$-mer size of 21, a minimum count for a $k$-mer to be included of 1 and an upper limit for $k$-mer counts of 5,000. To avoid noise, KMC database reduction

was performed using the transform operation with the L 30 and U 5000 settings. With smudgeplot[61] analysis and its hetkzer operation, the coverages of the identified *k*-mer pairs were written to a '_coverages. tsv' file. A custom R script with ggplot2[74] and data.table packages[92] was used to plot the distribution of frequencies of different SNP ratio classes. For each sample, the ploidy level was then estimated by visual inspection of the plots.

**Target back-mapping, variant calling and creating a sample-specific reference.** The raw SCO reads were trimmed using Trimmomatic (v.0.39)[110] with the following settings: 2:30:8 LEADING:13 TRAILING:13 SLIDINGWINDOW:4:19 MINLEN:36. We updated the script from a previous study[111] to run it with current package versions and used it for mapping, variant calling and sample-specific reference building. In brief, the trimmed short reads from the target enrichment sequencing were mapped against the SCO targets of the *R. chinensis* reference genome (5,794 sequences) using the BWA program[73]. Using SAMtools (v.1.16.1)[112], the reads were sorted and indexed, and duplicates were removed. Notably, approximately 98% of trimmed reads were successfully mapped to the target. Hits with exactly one alternative mapping position were subsequently filtered. After mapping, the Genome Analysis Toolkit (GATK) (v.4.1.9.0)[113] was used with the operation HaplotypeCaller[114] for variant calling, BaseRecalibrator and ApplyBQSR were used to realign around SNPs and indels, and FastaAlternateReferenceMaker was used to create a sample-specific consensus sequence as a reference for each SCO locus in each sample. The provided ploidy level for the HaplotypeCaller was 2 (diploid) for both the pollen and diploid roses with regular meiosis, and the --max-alternate-alleles flag was set to 6, so that although the pollen is monoploid, it would be possible to call potential variances.

**Phylogenetic reconstructions based on SCO markers.** The SCO target was used as a query for a local search with BLAST+[65] in our *R. canina* S27 genome assembly with a customized output table (-outfmt 6 qseqid sseqid pident length qstart qend sstart send evalue bitscore) and additional in the DToL *R. canina* (PRJEB79801) and *R. agrestis* (PRJEB79880; https://www.darwintreeoflife.org/) genomes also assembled by us (see below). Those SCO loci that had only five hits, one each on subgenomes S1_h1, S1_h2, S2, R3, R4 and R4_h1, R4_h2, R3, S2 and S1 for *R. agrestis*, respectively and within the same linkage group, were filtered and considered single copies. A main list of common single-copy loci for all three genomes was created to preserve the correct order and used to extract the filtered loci from the BLAST outputs with the grep command. The filtered BLAST output was then converted into a BED file containing the sequence coordinates using a bash script written with the help of GPT-4 (ChatGPT Plus, OpenAI). Using the BEDtools (v.2.30.0)[115] command getfasta, sequences for each SCO locus were extracted from the *R. canina* genome assembly and written into a multifasta file. To obtain sequences with the same strand orientation, two locus lists were also created: one of the loci with a positive strand orientation and one with a negative-strand orientation. Loci with negative-strand orientation were identified by calculating the end coordinates minus the start coordinates and filtering according to negative values. According to both lists, the sequences were extracted and stored in two separate multi-FASTA files. Sequences with negative-strand orientation were reversed and complemented with SeqKit[116] and combined with the positive strand-oriented SCO sequences in one fasta file. Finally, for each subgenome (S1_h1/S1_h2, S2, R3, R4 for *R. canina* and R4_h1/h2, R3, S2, S1 for *R. agrestis*), the extracted SCO sequences were concatenated in the same order according to the main locus list and written to subgenome-specific fasta sequences. The same procedure was used for the haploid genome assemblies of *Rubus ideaus* (GenBank: GCA_030142095.1)[117] and three strawberry species, *F. vesca* subsp. *vesca* (GCA_000184155.1)[88], *Fragaria iinumae* (GCA_009720345.1)[118] and *F. nilgerrensis* (GCA_010134655.1)[78] as outgroups. Moreover, the same

single-copy loci considered in the genome assembly were extracted and concatenated in the same order with target enrichment samples from nine pollen samples and 30 leaf samples of 26 diploid rose species. The concatenated multilocus sequences were aligned using MAFFT (v.7.490)[70]. Finally, a maximum-likelihood phylogenetic tree was generated by applying IQTREE[71] with ModelFinder using the following settings: iqtree2 -s -m TEST --con-tree --burnin 250 -B 1000 -T 12 --wbtl. The tree figures were graphically finalized with MEGA X[119] and Inkscape v.0.92.3 (2405546, 2018-03-11) software.

## Analyses of synthetic hybrids

Synthetic hybrid *R. canina* (seed parent) × *R. rubiginosa* (pollen parent) (sample ID, D62b_2; SRA: SRR15033882) was a cross between *R. canina* (sample ID, D3b_2; SRA: SRR15033883) and *R. rubiginosa* (sample ID, D145b_2; SRA: SRR15033877), and the second synthetic hybrid *R. rubiginosa* (seed parent) × *R. corymbifera* (pollen parent) (sample ID, D166b_2; SRA: SRR15033879) was a cross between *R. rubiginosa* (D145b_2; SRA: SRR15033877) and *R. corymbifera* (sample ID, D81b_2; SRA: SRR15033881). These hybrids were originally produced by Wissemann and Hellwig[43] and kept as a living plant in the Botanical Garden Gießen, Germany. Whole-genome short-read sequencing was performed for both hybrids and their parental plants. The mean coverage of the maternal plant (sample ID, D3b_2) is ~27×, and the paternal plant (sample ID, D145_b2) is ~27×. The hybrid's (sample ID, D62b_2) coverage is ~29×. The reciprocal hybrid (sample ID, D166_b2) has an average of ~30× coverage, whereas its paternal plant (sample ID, D81_b2) is ~19×.

The reads from these six samples were mapped to the S1 subgenome of our assembled *R. canina*, respectively, using bowtie2 (v.2.5.1)[63] with the default parameters. Filtering was applied for all alignments with the same setup 'samtools view -F 3340 --min-MQ 1'. The coverage of each sample was calculated by 'bedtools coverage' (v.2.30.0) with a 100 kb window size. SNPs were called with the filtered alignments by bcftools (v.1.9)[112]. Specifically, 'bcftools mpileup' ran first with the minimum mapping quality 1, then 'bcftools call' ran with flags '--keep-alts --variants-only --multiallelic-caller'. In the end, only the unique SNPs in each parent were selected to calculate the SNP contribution in the hybrids.

## ModDotPlot analysis

Structural analysis of DNA sequences of whole chromosomes and centromere cuts were performed with ModDotPlot (v.0.9.0)[120] using the default parameters. ModDotPlot is a dot plot visualization tool designed for large sequences and whole genomes. The method outputs an identity heat map by rapidly approximating the average nucleotide identity between pairwise combinations of genomic intervals.

## Gene and repeat sequence annotation

The predicted gene model structures in the nuclear genome were annotated by applying the full-length chromosome sequences to Helixer[121]. Moreover, complete LTR retrotransposons were annotated with the DANTE and DANTE-LTR tools implemented in RepeatExplorer2[122,123]. *R. canina* short-read data (SRA: ERR1662939) were subjected to clustering analysis using the RepeatExplorer2 pipeline, and the output library of repeats was subsequently used to annotate the genome with the implemented RepeatMasker[124]. Tandem-repeat annotation and genome abundance estimation were performed using TAREAN and TideCluster implemented in RepeatExplorer2[122].

## RNA sequencing and analysis

Total mRNA was extracted from the leaf tissue of *R. canina* S27 using the Spectrum Plant Total RNA-Kit (Sigma-Aldrich). The RNA-sequencing library was prepared with poly(A) enrichment and then sent for sequencing on the NextSeq 2000 platform with 2 × 150 bp mode, resulting in 33,594,132 reads. For a more accurate mapping of RNA sequences, the annotated tandem repeats and transposable elements

were hard-masked from the genome. RNA alignment was done using hisat2 (v.2.1.0)[125] with the flag --no-mixed. The output was then filtered by only allowing for tag NM:0 and minimum mapping quality 2. To count the transcripts number for each gene, we converted the masked genome to protein sequences based on Helixer[121] structural annotation, and then functionally annotated the protein sequences by Mercator4 (v.7.0)[126] with both Prot-scriber and Swissprot databases, then htseq-count (v.2.0.1)[127] was applied to count the transcripts for all annotated proteins. The gene expression was analysed by DESeq2[128]. As the high homozygosity between the haplotypes of S1 subgenome, RNA reads were aligned to S1_h1, S2, S3, S4 genome, and the expression level of S1 was then halved (Supplementary Fig. 20 and Supplementary Table 4).

## CENH3 ChIP–seq experiment and analysis

For detecting the functional centromeres of *R. canina* S27, we designed a specific polyclonal antibody against its CENH3 protein (ARVKHTAARKDRIKTARRQP-C, AB016310), synthetized by LifeTein with immunization in rabbits. The *CENH3* gene of *R. canina* S27 was identified using BLASTP with the parameter '-evalue 1e-5 -qcov_hsp_perc 50' and the *A. thaliana* CENH3 protein HTR12 (AT1G01370) was used as the reference. The ChIP experiment was performed as described previously[129] with a few modifications. Young leaves (around 2–5 g) of *R. canina* S27 were collected and cross-linked in 4% formaldehyde in 1× PBS on ice with vacuum infiltration applied for 30 min. The quenching was performed applying 1 M glycine in each sample followed by vacuum infiltration at room temperature for 15 min. The material was then macerated in liquid nitrogen and the chromatin was extracted. After extraction, the chromatin was sonicated for 30 min on a Bioruptor (Diagenode) until fragments of around 200–600 bp length were achieved (30 s on; 30 s off; in high mode). The sonicated chromatin was incubated over night at 4 °C with 1 µg of each polyclonal antibodies (anti-CENH3 specific for *R. canina* (LifeTein, AB016310) raised in rabbit and anti-histone H3 (Active Motif, 39064) raised in mouse). Samples with no addition of primary antibodies were also incubated as input control samples and at least two experimental replications were used for each ChIP combination. After incubation, protein beads (anti-rabbit: rProtein A Sepharose FastFlow 50% slurry; anti-mouse: rProtein G Sepharose FastFlow 50% slurry (GE Healthcare)) were washed and added to each complex protein–antibody and incubated for at least 2 h at 4 °C in slow rotation. The final recovered chromatin was eluted from the beads, followed by a de-cross-linking step and final DNA extraction. After quality control using the 4200 TapeStation System (Agilent Technologies), the samples were forwarded for 150 bp paired-end Illumina sequencing. For the analysis, the raw 150 bp paired-end reads were quality checked and then mapped to the *R. canina* haplotype phased reference genome using the default parameters in bowtie2[63]. The BAM file was converted to bigwig using the bamCompare tool from deeptools2[130], and then normalized to reads per kilobase of transcript per million reads mapped. Peak calling was then performed using the MACS3 pipeline[131] with the inclusion of the parameters --broad -g 1.9e+9. The plots showing the distribution of different genomic features per chromosome or specific region were constructed using pyGenomeTracks[132]. The ChIP–seq signals in metaplots to compare chromosome (Extended Data Fig. 7 and Supplementary Fig. 14) and subgenome CENH3 enrichment (Fig. 3e) were calculated by bamCompare with parameters '--ignoreDuplicates --scaleFactorsMethod readCount --operation log2' to normalize the CENH3/H3 by read coverage.

## Functional centromere annotation

Functional centromere regions in the genome assembly of *R. canina* S27 were annotated based on the detection of CENH3 peaks with MACS3 (see above). The total centromere length was then calculated by the interval between the 5' and 3' CENH3 peaks. After alignment to the annotated functional centromeres in *R. canina* S27, comparable centromeric

regions were extracted from DToL *R. canina* and *R. agrestis* (https://www.darwintreeoflife.org). The repeat abundance of *CANR4* satellite repeats and *Ty3/Gypsy ATHILA* retrotransposons in the predefined centromeric regions were determined in base pairs for each chromosome of the three investigated *Rosa* genomes (*R. canina* S27, *R. canina* DToL and *R. agrestis* DToL). To reduce data skewness the data were log-transformed. A Shapiro–Wilk normality test was used to check normal distribution of the data with R (v.4.3.3)[92] (29 February 2024). A bivariate Bayesian generalized linear mixed model was implemented using the MCMCglmm package[133]. The model included pairing type (bivalent B, univalent U and univalent in *R. canina* but bivalent in *R. agrestis* Ub) as a fixed effect, while subgenome, genome, and synteny group were random effects, with an unstructured covariance structure (us(trait):random_effect) to account for correlations between response variables. MCMC settings included 100,000 iterations, with a 50,000 burn-in and a thinning interval of 50 and the family parameter was set according to 'gaussian'. Data visualization was performed using the ggplot2[74], patchwork[134], tidyr[135] and dplyr[136] packages (Supplementary Fig. 15; source data are available in Supplementary Data 13 and 16). The correlation of *CANR4* size with CENH3 abundance (Fig. 3e) was calculated by Spearman's rank correlation as the Shapiro–Wilk normality test resulted in $P \ll 0.5$. Linear regression model was fitted using the lm function in R (v.4.4.0), with multiple $R^2$ value as 0.842 and adjusted $R^2$ value as 0.836.

## *cenLTR* sequence characterization

*cenLTR* sequences were primarily annotated as tandem repeats using TAREAN and TideCluster implemented in RepeatExplorer2[122]. Further sequence similarity with LTR retrotransposons was performed using the transfer annotation tool of Geneious Prime v.2025.0.2 (https://www.geneious.com) with a minimum sequence similarity threshold of 75%. Using a Geneious Prime plugin for ClustalO[137], we performed alignments of consensus *cenLTR* sequences against the regions with the highest similarity found in the *R. canina* S27 genome, which all corresponded to different *ATHILA* elements on chromosomes Rca1_R4 and Rca4_R4. Consensus sequences of *cenLTR1–4* are available in Supplementary Dataset 14.

## DNA methylation sequencing and analysis

To investigate the methylome of *R. canina*, we performed enzymatic methyl-sequencing (EM-seq). For this, we extracted genomic DNA from young leaves and the samples were then prepared for an Illumina-compatible library using the NEBNext Enzymatic Methyl-seq Kit and further sequenced on the HiSeq 3000 device with paired-end orientation. We ended up with 68,632,618 pairs of 150 bp reads. EM-seq data were first aligned to the S1_h1, S2, R3, R4 combined subgenomes with Bismark (v.0.23.0) with the flag '--local' and duplications were removed by deduplicate_bismark. CpG-, CHG- and CHH-context methylations were then extracted by bismark_methylation_extractor (v.0.23.0). The output was converted to bedgraph by bismark2bedGraph (v.0.23.0) with the flag '--CX' activated for CHG and CHH contexts to visualize the methylations chromosome-wide and on the centromeres.

## Metaplots of CENH3 enrichment, DNA methylation, *ATHILA* and *CANR4* density

In the metaplots (Fig. 3d, Extended Data Fig. 7 and Supplementary Fig. 14), all signals were smoothed by the spline.smooth function with spar 0.3 in R (v.4.4.0). CENH3 enrichment was calculated by CENH3 ChIP–seq ($\log_2[\text{CENH3/H3}]$) signal normalized by coverage. CENH3 enrichment, DNA methylations, *ATHILA* density and *CANR4* were calculated in 50 kb adjacent windows and averaged by all chromosomes of the corresponding subgenome. All chromosome coordinates were scaled on the basis of their distance to centromere against the distance of centromere to telomere. Centromere position was defined

on the basis of where the maximum CENH3 enrichment was located. Mitochondrial sequences were masked when computing the CENH3 enrichment. All signal values ($y$ axis of metaplots) were scaled from 0 to 1 based on the global minimum to global maximum except for DNA methylations, for which the original percentage values were retained. The p- and q-arm values were averaged and mirrored.

### Immunodetection of CENH3 and microtubules

For immunodetecting the centromeres of *R. canina* S27, we used polyclonal antibodies against CENH3 protein (see above) and kinetochore protein KNL1 (C-EDHFFGPVSPSFIRPGRLSD, AB015677-3) described previously[48], also synthetized by LifeTein and raised in rabbits. To identify the microtubules, we used a commercial antibody against alpha-tubulin (Sigma-Aldrich, T6199) with immunization in mouse. For analysing the distribution of these markers in mitotic cells, root tips were fixed after a pretreatment in 0.2 mM 8-hydroxyquinoline for 4 h at 18 °C. For meiotic stages, the young anthers were directly fixed with no previous antimitotic pretreatment. The immunodetection experiment was performed according to a previously published protocol[138] with modifications to *R. canina* material. Young flower buds were collected on ice in buffer A (15 mM PIPES–NaOH, 80 mM KCl, 0.5 mM ethylene glycol tetraacetic acid, 80 mM sorbitol, 20 mM NaCl, 2 mM ethylenediaminetetraacetic acid (EDTA), 0.15 mM spermine, 0.5 mM spermidine and 1 mM dithiothreitol) and next incubated in 4% paraformaldehyde in buffer A for 1 h under vacuum infiltration on ice. After the fixation, the samples were washed three times with buffer A and then digested in enzymatic solution containing 1% cellulase-onozuka, 1% cellulase, 1% pectolyase Y23, 1% cytohelicase, 1% macerozyme and 10% pectinase in citrate buffer for 1 h in a humid chamber at 37 °C. To remove the excess of enzymatic solution, the material was gently washed with buffer A and left on ice until the preparation of the slides. A couple of anthers were placed and dissected in a drop of buffer A on a 18 × 18 mm high-precision coverslips, a few µl of polyacrylamide solution (25 µl 15% polyacrylamide (Sigma-Aldrich, A3574) in buffer A plus 1.25 µl of 20% sodium sulfite and 1.25 µl of 20% ammonium persulfate) were added to the dissected anthers, quickly mixed and a second coverslip was put above the first making a sandwich gently squeezing the anthers with a needle to liberate the meiocytes. The sandwiches were allowed to dry for up to 1 h until complete polymerization. After this, the coverslips were carefully separated and incubated in PBS with 1% Triton X-100 and 1 mM EDTA for at least 1 h, then more 2 h in blocking solution containing 3% BSA in PBS with 0.1% Tween-20. After this period, the primary antibodies were diluted in 1:500 (CENH3 and KNL1) and 1:200 (alpha-tubulin) ratios in blocking solution and applied on each sample, which were sequentially incubated at 4 °C for 48 h. After primary antibody incubation, primary antibodies were detected using secondary antibodies conjugated with specific fluorophores (Alexa Fluor 488 and Abberior StarRed and STAROrange for STED microscopy, also diluted in blocking solution in a proportion of 1:250) and incubated in a dark humid chamber at room temperature for at least 2 h. The material was then washed four to five times for 20 min each in 1× PBS + 0.1% Triton X-100 and then mounted in SlowFade Gold medium containing DAPI. The slides were photographed using a super-resolution STED microscope (Abberior instrument facility line; https://abberior-instruments.com/) and posterior brightness and contrast adjustments were done in Photoshop.

### Chromosome preparation and FISH

For mitotic chromosome preparations, root tips and young flower buds from *R. canina* S27 plants cultivated in the greenhouse were collected and then fixed in methanol:acetic acid solution (3:1 (v/v)) for 2–24 h at room temperature and then kept at −20 °C until use. After fixation, the root tips were pretreated with an enzymatic solution of 2% cellulase R10-onozuka (Duchefa Bioquemie)/20% pectinase (Sigma-Aldrich) in 0.1 M citric acid for 40 min at 37 °C in a humid chamber and then squashed in a drop of LB01 buffer (15 mM Tris, 2 mM Na$_2$EDTA, 80 mM KCl, 20 mM NaCl, 0.5 mM spermine, 15 mM β-mercaptoethanol, 0.1% Triton X-100 (pH 7.5)) and, after frozen in liquid nitrogen, the coverslips were removed.

For meiotic chromosome preparations, the anthers of *R. canina* C1 (GLM-P-0181117) were dissected from fixed flower buds around 0.5 cm in length. Anthers were washed with 1% (w/v) polyvinylpyrrolidone 40 (PVP-40; Sigma-Aldrich Chemie) and 0.5% (v/v) Triton X-100 for 15–20 min, followed by enzymatic digestion overnight in a humid chamber at 4 °C in 1% (w/v) cellulase Onozuka R-10 (Serva), 0.2% (w/v) pectolyase Y-23 (Sigma-Aldrich), 0.5% (w/v) hemicellulose (Sigma-Aldrich) and 0.5% (w/v) macerozyme R-10 (Duchefa Biochemie) dissolved in citric buffer (0.04 M citric acid and 0.06 M sodium citrate). The anthers were macerated on a slide, squashed in a drop of 70% acetic acid and fixed by freezing in liquid nitrogen.

For FISH experiments, a 22 bp oligo probe directly labelled with a Cy3 fluorophore at the 5′ terminus was designed, synthesized by Sigma-Aldrich and then used to detect the *CANR4* satellite repeat (Cy3-5′-ACCCTAGAAGCAAGAAGTTTGG-3′) or an insert of the plasmid carrying the *CANR4* dimer (GenBank MK069593) was used as a FISH probe[20]. For detection of the centromeric LTR *ATHILA* retrotransposon sequences, we designed a probe based on clustering analysis of Illumina reads (SRA: ERR1662939) using the RepeatExplorer2 pipeline. It was revealed that cluster 5 (CL5) contig contained *Ty3/Gypsy/ATHILA* sequences. The CL5 contig was used to design PCR primers amplifying a 180 bp product from *R. canina* genomic DNA. The primers were as follows: Rcan_centr_CL5_for: 5′- GCAAGCGCATAATTTAACC-3′ and Rcan-centr_CL5_rev: 5′-CAATCAAAAATATCCCCCC-3′. The PCR product was purified and cloned into the pDrive vector (Qiagen) and sequenced by the Sanger dideoxy method using the SP6 primer (Micosynth). Clone 11 was submitted to GenBank (PV030978). The inserts of plasmids were directly labelled in a nick translation reaction with Cy5 d-UTP or Cy3 d-UTP fluorochromes (Jena Bioscience) and used for FISH. To detect the 5S and 35S rDNA loci, the full-length 18S rRNA gene from tomato (GenBank: X51576.1) and the *Pta71* clone from *Triticum aestivum* were used to detect the 35S rDNA region, while a 5S rDNA unit (B variant) from *R. canina*[35] and the D2 clone from *Lotus japonicus* were used to detect the 5S rDNA locus. rDNA robes were directly labelled by Nick translation using Cy5 d-UTP (Jena Bioscience). The slides were prepared in accordance with the protocols described previously[35,139]. In brief, the slides were treated with pepsin solution (1 mg ml$^{-1}$ diluted in 0.01 N HCl) for 30 min at 37 °C in a humid chamber, washed with 2× SSC (saline sodium citrate, pH 7.0) solution, post-fixed with 4% paraformaldehyde for 10 min at room temperature, washed again with 2× SSC and then dried in 70% and 100% ethanol. After air drying for at least 30 min, the slides were denatured with hybridization mix (50% formamide, 2× SSC, 10% dextran sulfate and ~50 ng of each probe (15 µl per slide)) for 5 min at 75 °C and then incubated for at least 18 h at 37 °C. After hybridization, stringency washes were performed with 2× and 0.1× SSC solutions at 42 °C, achieving around 76% stringency. The slides were then washed at room temperature with 2 × SSC solution and mounted with DAPI in the antifade mounting medium Vectashield (Vector Laboratories).

### Alexander staining

Five mature and well open flowers were collected from the plant in the greenhouse. They were shaken above a microscope slide and their pollen was released on top of the slide. Then, 20 µl of Alexander staining solution (Morphisto, 13441.00250) was added and briefly mixed with the pollen by stirring with the pipette tip. A coverslip 24 × 40 mm was put on top of the mix. Pictures were taken with a Labscope microscope by Zeiss, using 10 × magnification. Five snapshots were counted with the help of the ZEN software.

### Reporting summary

Further information on research design is available in the Nature Portfolio Reporting Summary linked to this article.

## Data availability

All raw sequencing data (HiFi, Hi-C, RNA, CENH3-ChIP, DNA methylation, SCO of pollens) and genome assembly of *R. canina* S27 isolate are available under NCBI BioProject: PRJNA1111045. The chloroplast genome of *R. canina* S27 isolate available under GeneBank accession number PV550499. Raw sequencing data of *R. canina* DToL and *R. agrestis* DToL are available from Darwin Tree of Life (DToL) data portal (https://portal.darwintreeoflife.org/). The corresponding NCBI BioProject accession numbers are PRJEB79802 and PRJEB79880, respectively. Genome assemblies, the sample-specific SCO reference sequences, variant calling format files, annotations and alignments presented in this work are also available for download at Dryad[140] (https://doi.org/10.5061/dryad.cc2fqz6fh). The REXdb database Viridiplantae v.3.0 (http://repeatexplorer.org/?page_id=918) is publicly available. All other data needed to evaluate the conclusions in the paper are provided in the Article and its Supplementary Information.

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

**Acknowledgements** The authors thank A. Houben, H. Bruelheide, B. Usadel, M. Thines, M. Bagdevi and K. Wesche for discussions on the research; B. Schlitt, M. Schwager, S. H. Sourav, J. Wesenberg, A. Smolka and S. Stegmann for their technical assistance in the laboratory; colleagues F. Erdogan, I. Kisakesen and U. Abraham for their help with target enrichment; T. Schell, C. Sinai, C. Greve, L. Schardt, D. Baranski and A. Ben Hamadou for technical support; J. Bauernfeind for help with IT facilities; T. Hawel, G. Schulz, S. Arndt, G. Vogg, S. Ruge and Oberlausitz-Stiftung for providing access to their collections and plant material; M. Simon for providing rose photos; the staff at the Darwin Tree of Life Project at the Wellcome Sanger Institute for making the data of the additional accession of *R. canina* and *R. agrestis* available (https://www.darwintreeoflife.org/project-resources); and the staff at the Institute of Biophysics AS CR for internal support. This study was funded by the German Research Foundation (DFG Ri 2090/4-1, MA 9363/2-1 and MA 9363/3-1), the Czech Science Foundation (GAČR 22-16826S), the Max Planck Society (core funding to A.M.), and the European Union (European Research Council Starting Grant, HoloRECOMB, grant no. 101114879 to A.M.). The DFG also funded this work under Germany's Excellence Strategy—EXC 493 2048/1–390686111 (to A.M.). M.Z. is financially supported by the DFG (grant no. MA 9363/2-1).

**Author contributions** A.M., A.K. and C.M.R. conceived the research. V.H. designed and performed the SCO experiments and processed the data. M.Z. and A.M. performed the genome assembly and genomics. D.H. performed the plastome assembly and annotation. B.H. performed all nuclear sequencing libraries. T.N., M.Z. and A.M. performed the ChIP–seq analysis. T.N. and J.L. performed the chromosome preparations, immunodetection and FISH. J.F. performed the flow cytometry measurements and the pollen-nucleus sorting. A.K., J.L., R.K. and A.M. performed the centromere and repeat characterization. U.P. performed the pollen viability assay. V.W. created and curated the synthetic hybrids. A.M., A.K. and C.M.R. supervised the project. V.H., M.Z., A.M. and C.M.R. wrote the original draft, and all of the authors discussed the results and contributed to writing the manuscript.

**Funding** Open access funding provided by Max Planck Society.

**Competing interests** The authors declare no competing interests.

**Additional information**
**Correspondence and requests for materials** should be addressed to A. Kovařík, A. Marques or C. M. Ritz.

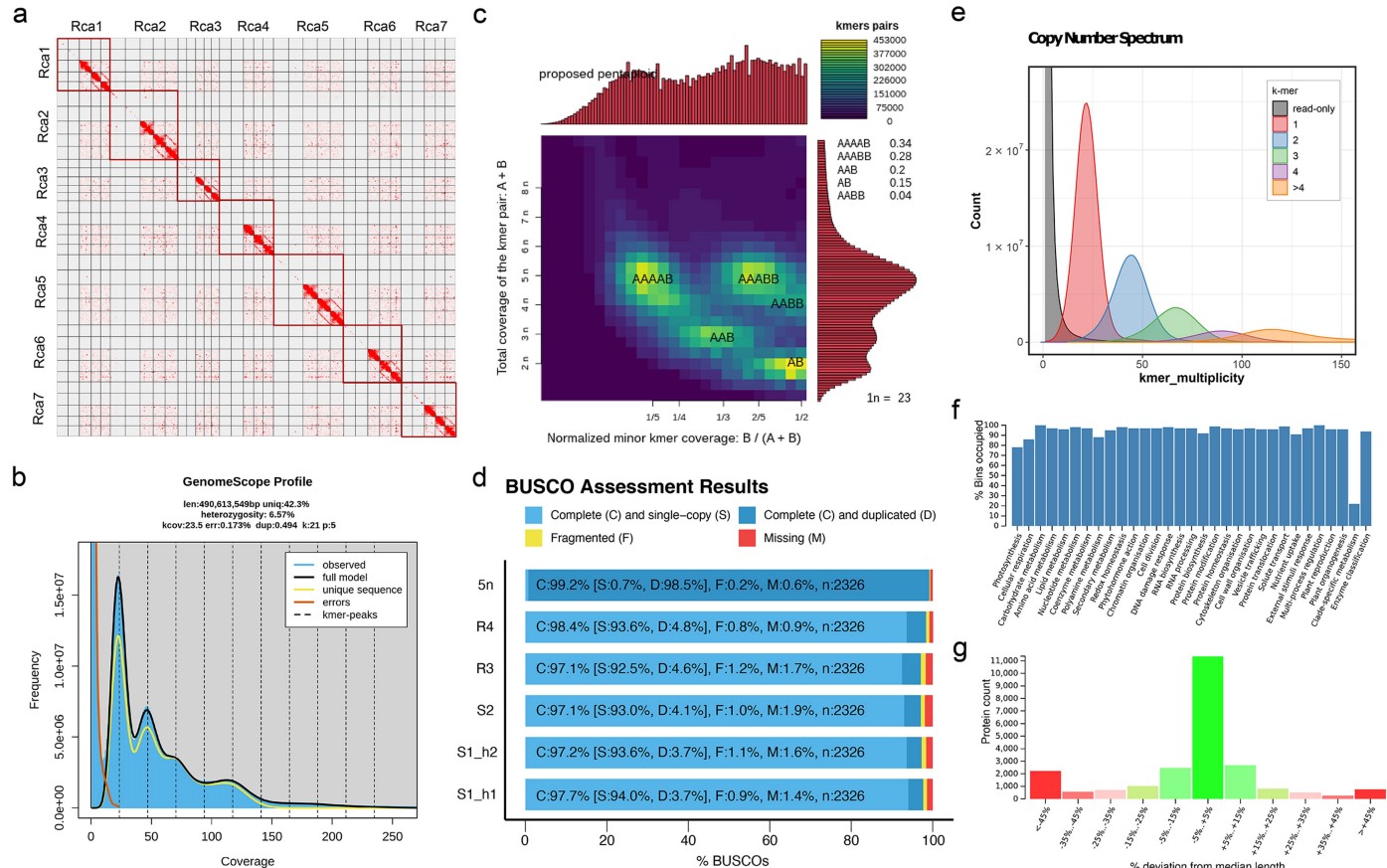

**Extended Data Fig. 1 | Genome assembly assessment of *R. canina* (S27).**
(**a**) Hi-C map of 35 chromosome-level scaffolds. Each synteny group contains five chromosomes, which were ordered as S1_h1, S1_h2, S2, R3, and R4 in sequence. The vanished Hi-C signals of the first two chromosomes in each linkage group suggest the high homozygosity of S1_h1 and S1_h2 haplotypes. (**b**) GenomeScope2 *21*-mer distribution from HiFi sequences and genome size estimation confirmed the expected *R. canina* genome size, i.e., 5× 490 Mbp. 21-mer coverage is estimated as 23.5× coverage, consistent with read coverage (23×). The 4*x* peak at coverage 94 is less invisible, indicating two out of five haplotypes are highly similar. (**c**) Ploidy and genome structure inference based on *21*-mer Smudgeplot analysis. A and B represent the number of heterozygous *k*-mers identified. Note the high amount of pentaploid heterozygous *k*-mer combinations AAAAB and AAABB. (**d**) BUSCO (Universal Single-Copy Orthologues) genes assessment results based on the annotated protein

sequences of the pentaploid genome and each individual subgenome set. (**e**) HiFi *k*-mer multiplicity frequency by Merqury. The number in the plot legend and different colours indicate the copy numbers found in the assembly. (**f**) and (**g**) are Mercator4 statistics of gene contents based on its functional annotation on *Rosa canina* protein sequences. (**f**) Percentage of genes occupying a certain gene category. (**g**) Protein length distribution based on the deviation to category-specific reference lengths. Each bar represents the number of proteins having a certain length difference to the reference length of the corresponding Mercator4 category. The overall small deviation of gene lengths in *R. canina* comparing to Mercator reference genes indicate most annotated genes in *R. canina* are complete. (**f**) and (**g**) show the results for the subgenome *S1* haplotype h1 only since the other three subgenomes show highly similar distribution.

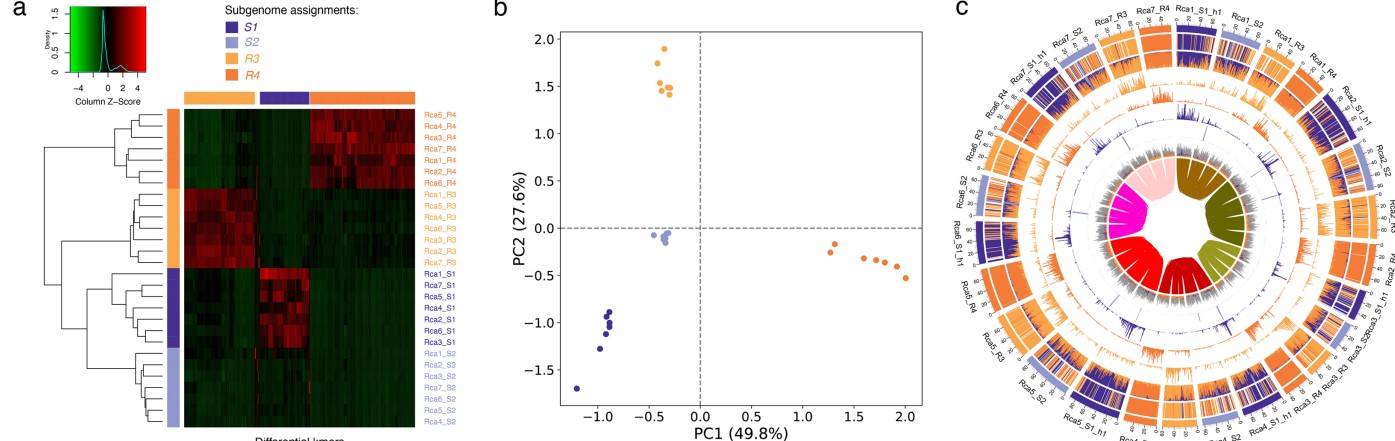

**Extended Data Fig. 2 | *K*-mer-based subgenome phasing and characterization of the pentaploid *R. canina* genome.** (**a**) Unsupervised hierarchical clustering (the horizontal colour bar at the top of the axis indicates to which subgenome the *k*-mer is specific; the vertical colour bar on the left of the axis indicates the subgenome to which the chromosome is assigned). The heatmap indicates the *Z*-scale relative abundance of *k*-mers. The larger the *Z* score is, the greater the relative abundance of a *k*-mer), and (**b**) Principal Component Analysis (PCA) of differential 15-mers confirmed that the genome was successfully phased into four subgenomes based on clearly distinct patterns of both differential *k*-mers and homoeologous chromosomes. (**c**) Chromosomal characteristics. From the outer to inner circles (1–9): (1) subgenome assignments based on the *k*-means algorithm; (2) significant enrichment of subgenome-specific *k*-mers – the same colour as the subgenome indicates significant enrichment for those subgenome-specific *k*-mers – white areas are not significantly enriched; (3) normalized proportion (relative) of subgenome-specific *k*-mers; (4–7) count (absolute) of each subgenome-specific *k*-mer set; (8) density of long terminal repeat retrotransposons (LTR-RTs) – if the colour is consistent with the subgenome, it indicates that LTR-RTs are significantly enriched in those subgenome-specific *k*-mers; grey indicates nonspecific LTR-RTs; and (9) homoeologous blocks. All statistics (2–7) are computed in sliding windows of 1 Mb. Notably, the fraction of subgenome differentiation between the S1 and S2 subgenomes was restricted to proximal regions, likely representing their different centromere compositions. Whereas R3 and R4 subgenomes are clearly distinct from each other in their specific *k*-mer spectrum.

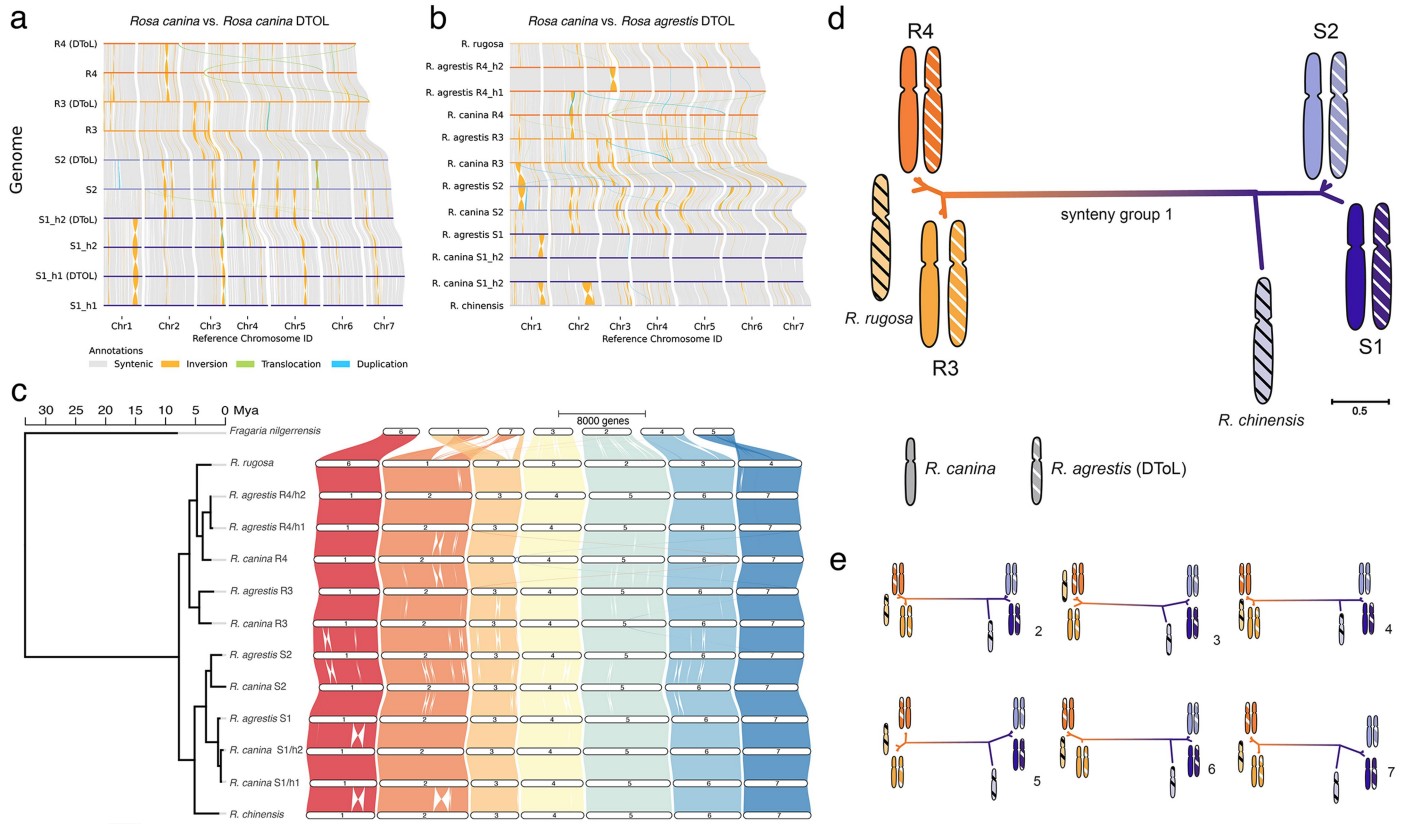

**Extended Data Fig. 3 | Comparative synteny analysis of the assembled R. canina (S27) with the R. canina DToL and R. agrestis DToL.** (**a**) Synteny and rearrangement analyses (SyRI) between our and the *R. canina* genome assembly from DToL. (**b**) Synteny and rearrangement analyses (SyRI) between our *R. canina* and the *R. agrestis* genome assembly from DToL. Pairwise comparisons of synteny of all *R. canina* subgenomes (S1_h1/S1_h2, S2, R3, and R4) and all *R. agrestis* subgenomes (S1, S2, R3, and R4_h1/R4_h2) juxtaposed against the corresponding chromosomes of *R. chinensis* (sect. *Synstylae*) and *R. rugosa* (sect. *Rosa*). Note that only synteny blocks and rearrangement blocks greater than 50 kb in length are shown here. (**c**) GENESPACE synteny and phylogenetic relationships of the five chromosome sets of our *R. canina* assembly, *R. agrestis*

from DToL and their close relatives *R. chinensis* and *R. rugosa*. Chromosomes are normalized by number of genes. (**d**–**e**) Unrooted Maximum Likelihood phylogenies of the homoeologous *R. canina* and *R. agrestis* chromosomes and chromosomes from the respective syntenic groups of the diploid species *R. chinensis* and *R. rugosa* based on multiple alignments of whole chromosome sequences. The upper panel (**d**) exemplarily depicts the phylogeny of chromosomes of synteny group 1. Synteny groups 2–7 are presented in (**e**). Filled chromosomes refer to subgenomes of *R. canina* belonging to the *Synstylae* clade (violet/light blue) and *Rosa* clade (dark/light orange). Chromosomes from *R. agrestis* are marked with hatched white filling, while the diploid roses are marked with black hatched filling.

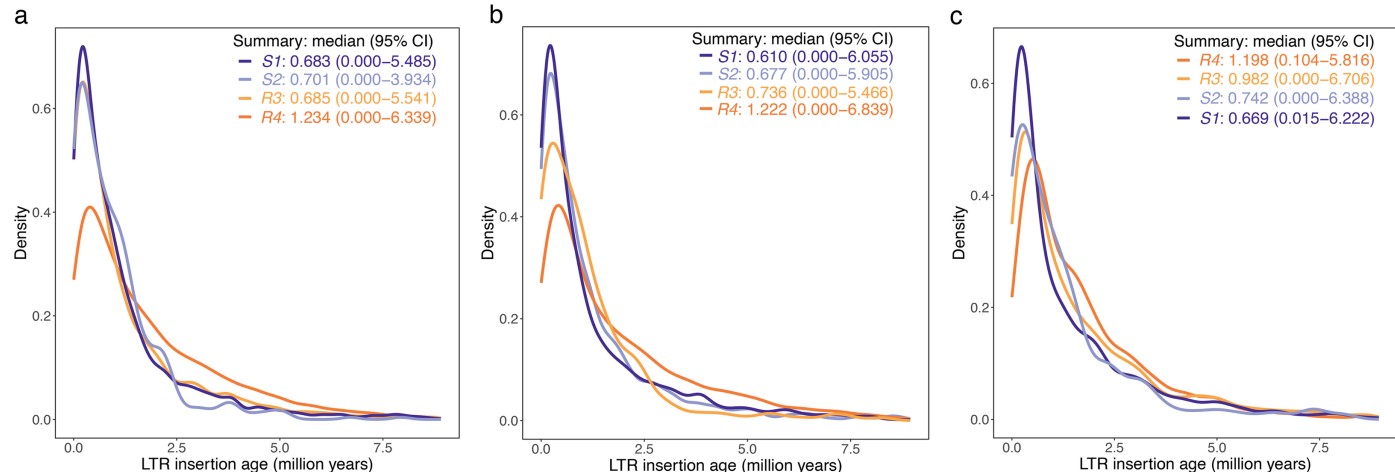

**Extended Data Fig. 4 | Dating subgenome-specific LTR-RTs insertion times in dogrose genomes.** The 95% confidence intervals (CIs) are marked in the upper right corner to predict the insertion time boundaries of LTR-RTs on the subgenome for *R. canina* (**a**), *R. canina* DToL (**b**) and *R. agrestis* DToL (**c**). The colours of the subgenomes are consistent throughout the Fig. panels.

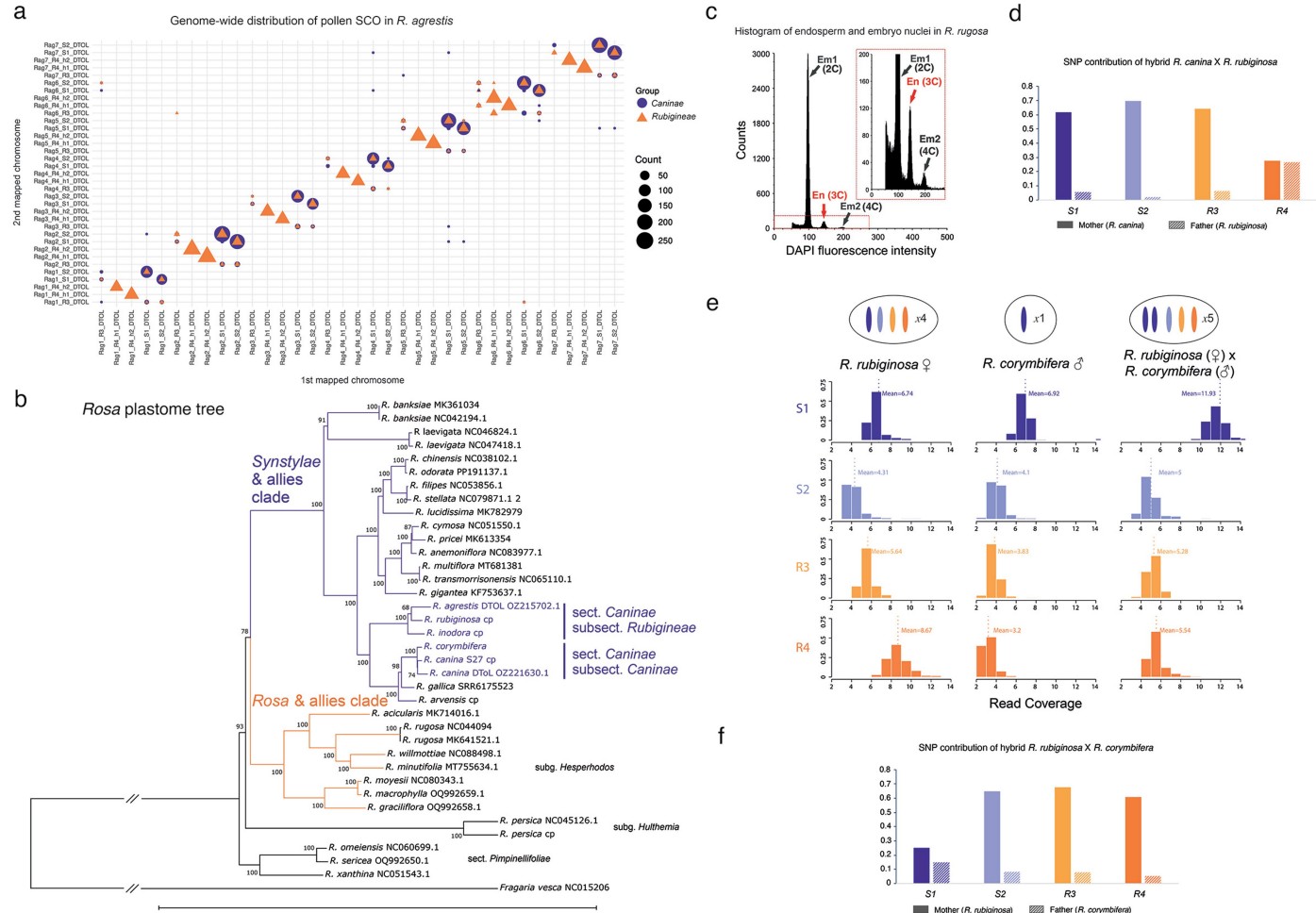

**Extended Data Fig. 5 | Experimental validation of the reproduction mode of dogroses.** (**a**) Genome-wide distribution of pollen SCOs from pollen of eigth dogrose species (subsect. *Caninae*: three samples of *R. canina*, two samples of *R. corymbifera*; subsect. *Rubigineae* three samples of *R. rubiginosa*) from two subsections in the *R. agrestis* genome assembly from DToL. The bubble map represents chromosomal hits from the SAM file output, which were selectively filtered to display loci with a single alternative hit. The size of the symbols corresponds to the mean count of pollen SCOs mapped to each chromosomal pair, identifying seven bivalent chromosome pairs within the *R. agrestis* (subsect. *Rubigineae*) genome assembly and seven different pairs in which *R. rubiginosa* (subsect. *Rubigineae*) pollen SCO mapped. (**b**) Plastome phylogeny of the genus *Rosa*. Maximum Likelihood phylogeny of the genus *Rosa* retrieved from plastome sequences obtained in this study and those available in Genbank (Supplementary Table 4). All nodes which were not supported by 100% bootstrap are indicated by dashed lines. (**c**) Representative histogram of flow cytometric measurements of nuclei isolated from nutlets of the sexual diploid *R. rugosa* to determine the endosperm/embryo ratios. The observed endosperm/embryo ratio 1.5, fits to the expectation of 3*x* endosperm and a 2*x* embryo originated by sexual reproduction including endosperm fertilization. Please note that the gain of the fluorescence was adjusted to position the embryo peak at channel 100. For a better visualization of the

endosperm peak the scale of the Y-axis was manually adjusted to 200 counts (red rectangles). Em1: embryo G0/G1, Em2: embryo G2, En: endosperm G0/G1. (**d**) The parental SNP ratios detected in each subgenome of the hybrid *R. canina* × *R. rubiginosa* (Sample ID: D62b_2). SNPs were called by aligning short reads to our assembled *R. canina* (S27) *S1_h1*. The y-axis displays the proportion of SNPs in each subgenome. For instance, ~62% SNPs detected in the alignments to S1 subgenome are the same as the maternal *R. canina* (Sample ID: D3b_2), while only around 5% SNPs are from the paternal *R. rubiginosa* (Sample ID: D145_b2), indicating the S1 subgenome of the hybrid was supposed to be contributed by the maternal parent. Overall, the hybrid subgenome S1, S2, and R3 are all from maternal parent *R. canina*, while one R4 subgenome is from maternal *R. canina* and the other R4 is contributed by paternal *R. rubiginosa*. So, the subgenome composition of the hybrid should be S1, S2, R3, R4_h1/h2. (**e**) Assessment of the parental genomes contribution of a synthetic hybrid between *R. rubiginosa* (female donor) and *R. corymbifera* (male donor), confirming the sexual reproduction and subgenome's inheritance trough male and female meiosis. Note, that *R. corymbifera* belongs to subsect. *Caninae* and is a very close relative of *R. canina* and so it has two copies of S1 subgenome. (**f**) The parental SNP ratios detected in each subgenome of the hybrid *R. rubiginosa* × *R. corymbifera* (Sample ID: D166b_2). The plot interpretation is the same as (**d**).

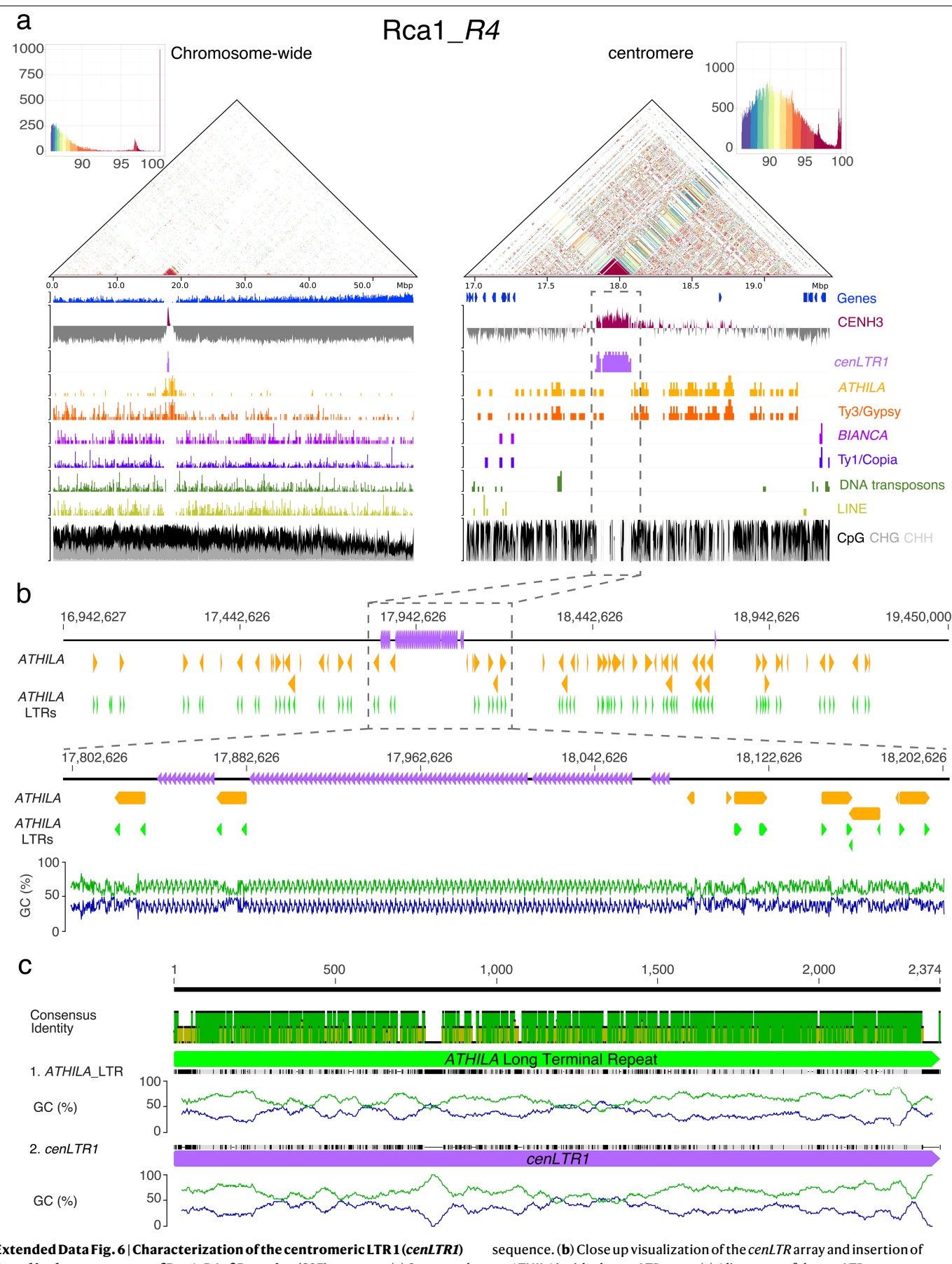

**Extended Data Fig. 6 | Characterization of the centromeric LTR 1 (*cenLTR1*) found in the centromere of Rca1_*R4* of *R. canina* (S27) genome. (a)** Structural visualization of the chromosome-wide (left) and centromere (right) close up. Please note the specificity of the CENH3 ChIP-seq signal only on the *cenLTR* sequence. **(b)** Close up visualization of the *cenLTR* array and insertion of an *ATHILA* inside the *cenLTR* array. **(c)** Alignment of the *cenLTR* consensus sequence with the LTR sequence from the inserted *ATHILA* element showing over 75% sequence similarity.

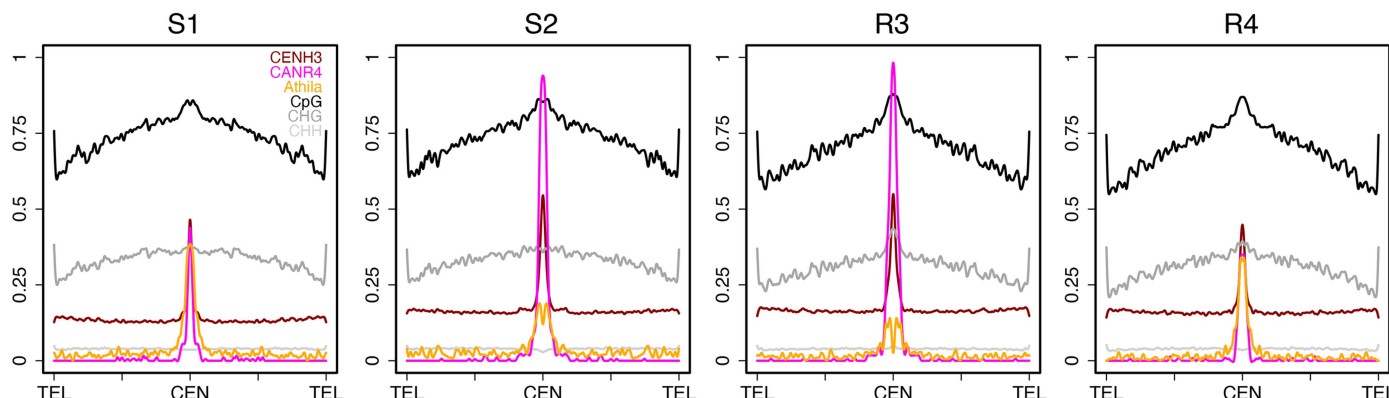

**Extended Data Fig. 7 | Genetic and epigenetic centromere variation in *R. canina* (S27).** Metaplot of CENH3 enrichment, DNA methylation, and centromeric elements–*ATHILA* and *CANR4* density–per subgenome. CENH3 enrichment marked in dark red was calculated by CENH3 ChIP-seq (log2 CENH3/H3) signal normalized by coverage. DNA methylations have three contexts with CpG marked in black, CHG maked in dark grey, and CHH marked in light grey. *ATHILA* (yolk yellow) and *CANR4* (magenta) were presented by their density. All signals were calculated in 50 kb adjacent windows.

All chromosome coordinates were scaled based on their distance to centromere against the distance of centromere (CEN) to telomere (TEL). Centromere position (CEN) was defined by where maximum CENH3 enrichment was located. All signal values (y-axis) were scaled from 0 to 1 by global minimum to global maximum except for DNA methylations, which retained the original percentage values. The p- and q-arm values were averaged and mirrored. Please note the higher CENH3 association with *CANR4*-based centromeres compared with *ATHILA*-based centromeres.

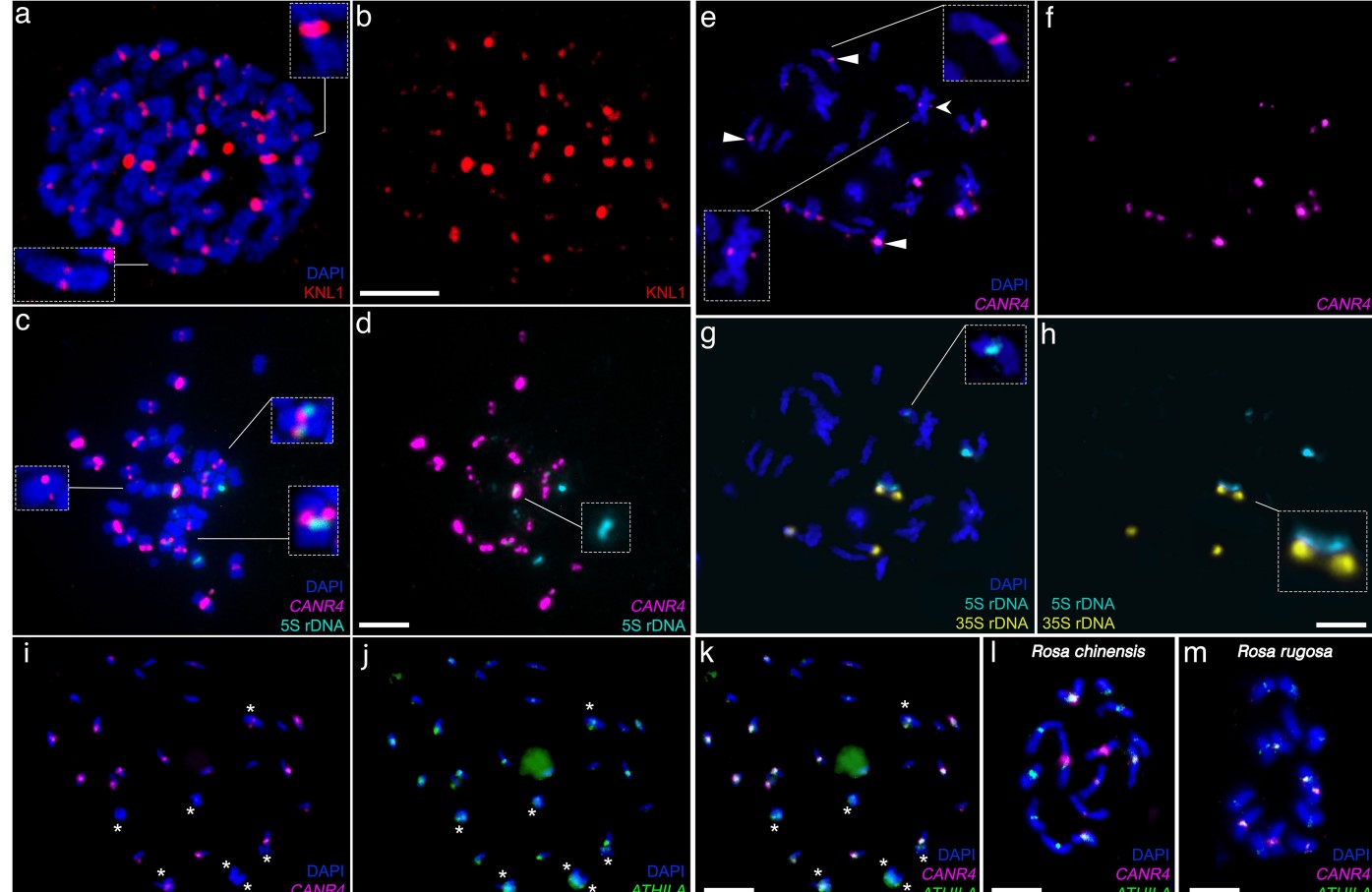

**Extended Data Fig. 8 | Detection of the kinetochore protein KNL1, *CANR4* satellite repeat, *ATHILA* and rDNA in the chromosomes of *R. canina*.** (**a–b**) Immunodetection of the KNL1 protein (in red) at all centromeres of *R. canina* revealed a differential distribution, with the presence of large and small centromeres, as shown in more detail in the upper and lower insets (**a**). (**c–d**) Distribution of the centromeric satellite repeats *CANR4* (in magenta) and 5S rDNA (in cyan) in the *R. canina* miotic chromosomes, while the *CANR4* repeat is present at the centromeres of a set of chromosomes. The 5S rDNA is located at the pericentromeric regions of chr1, insets in **c** and **d** highlight the pericentromeric location of 5S loci (right) and a weak *CANR4* signal (left).

(**e–h**) Distribution of the *CANR4* repeat in meiotic diakinesis, revealing its distribution mostly in univalents according (**e** and **f**), while the pattern of 5S and 35S rDNA (**g** and **h**) (cyan and yellow, respectively) hybridized to three sites, two of which were located near 35S sites on a bivalent (inset in **h**) corresponding to the Rca3_S1_h1/h2 chromosomes as found in our assembly annotation. Week signals are highlighted in the insets present in **g** and **h**. (**i–m**) FISH with *CANR4* and *ATHILA* on a diakinesis of *R. canina* (**i–k**) and mitotic cells of *R. chinensis* (**l**) and *R. rugosa* (**m**). Asterisks in **i–k** point to bivalent-forming chromosomes. The experiments for this set of data were repeated at least 10 times and independently presenting similar results. Scale bar = 5 μm.

| DAPI | CENH3 | MERGE |
|------|-------|-------|

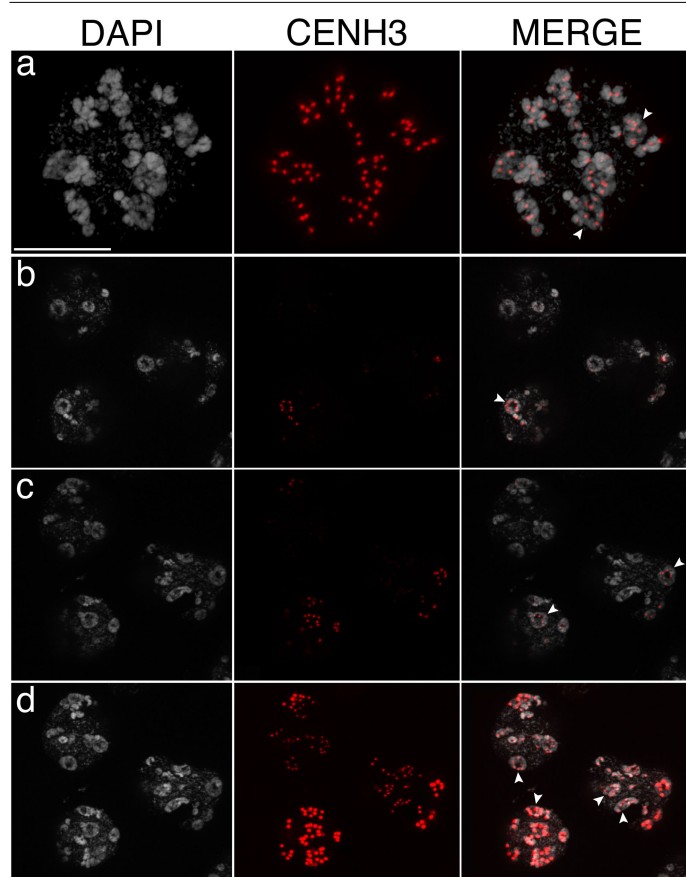

**Extended Data Fig. 9 | CENH3 immunostaining in polyads that are formed at the end of male meiosis in _R. canina_ (S27). (a)** A single polyad showing at least two nuclei with seven CENH3 signals (arrowheads), while several other nuclei show irregular number of centromeric foci. (**b**–**d**) Different Z-stacks of three polyads showing few nuclei with seven centromeric foci, while several other smaller nuclei are formed with irregular number of centromeres. Immunostaining experiments were repeated independently at least ten times outputting similar results. Scale bars = 10 μm.

# Reporting Summary

## Statistics

For all statistical analyses, confirm that the following items are present in the figure legend, table legend, main text, or Methods section.

| n/a | Confirmed | |
|---|---|---|
| ☐ | ☒ | The exact sample size (*n*) for each experimental group/condition, given as a discrete number and unit of measurement |
| ☐ | ☒ | A statement on whether measurements were taken from distinct samples or whether the same sample was measured repeatedly |
| ☐ | ☒ | The statistical test(s) used AND whether they are one- or two-sided<br>*Only common tests should be described solely by name; describe more complex techniques in the Methods section.* |
| ☒ | ☐ | A description of all covariates tested |
| ☐ | ☒ | A description of any assumptions or corrections, such as tests of normality and adjustment for multiple comparisons |
| ☐ | ☒ | A full description of the statistical parameters including central tendency (e.g. means) or other basic estimates (e.g. regression coefficient) AND variation (e.g. standard deviation) or associated estimates of uncertainty (e.g. confidence intervals) |
| ☒ | ☐ | For null hypothesis testing, the test statistic (e.g. *F*, *t*, *r*) with confidence intervals, effect sizes, degrees of freedom and *P* value noted<br>*Give P values as exact values whenever suitable.* |
| ☒ | ☐ | For Bayesian analysis, information on the choice of priors and Markov chain Monte Carlo settings |
| ☒ | ☐ | For hierarchical and complex designs, identification of the appropriate level for tests and full reporting of outcomes |
| ☒ | ☐ | Estimates of effect sizes (e.g. Cohen's *d*, Pearson's *r*), indicating how they were calculated |

*Our web collection on statistics for biologists contains articles on many of the points above.*

## Software and code

Policy information about availability of computer code

| Data collection | HiFi and Omni-C reads were obtained through own sequencing at the Max Planck Institute for Plant Breeding Research, Cologne, Germany. Images were analyzed using the ZEN software (Carl Zeiss GmbH) and the ZENBlack software (Carl Zeiss GmbH). |
|---|---|
| Data analysis | Available open source tools used in this study were:<br><br>Bedtools (v2.29.0)<br>bcftools 1.15.1<br>Bismark (v0.23.0)<br>BLAST 2.13.0+<br>Bowtie2 (2.5.4)<br>BUSCO (v5.1.2)<br>BWA (0.7.17)<br>CoGe (v7)<br>Cutadapt (v4.7)<br>DANTE_LTR (v0.3.5.2)<br>Deeptools (v3.5.1)<br>DESeq2 (1.46.0)<br>Dotter (v0.13.1)<br>EMBOSS (v2024.0419.155605)<br>findGSE_v1.94.R<br>GATK v4.1.9.0 |

GenomeScope2
Geneious (v2023.0.1)
GENESPACE (v1.3.1)
Helixer (0.3.4)
Hisat (2.2.1)
Hifiasm (0.19.8-r603)
htseq-count (v2.0.1)
IQ-TREE (2.4.0)
Jellyfish (v2.3.1)
Juicer (v1.6)
Kaks Calculator (v3)
MACS3 (3.0.1)
MAFFT (7.526)
minimap2 (v2.26)
ModDotPlot (v0.8.2)
PAML v4.10.6
plotsr (v0.5.3)
pyGenomeTracks (v3.8)
QUAST (v5.2.0)
RepeatExplorer2 (v2.3.7)
REXdb (v1.0)
SALSA2 (v2.3)
Samtools (v1.9)
StainedGlass (v0.6)
syri (v1.5.3)
VCFtools (0.1.16)
ZEN blue (3.1)

For manuscripts utilizing custom algorithms or software that are central to the research but not yet described in published literature, software must be made available to editors and reviewers. We strongly encourage code deposition in a community repository (e.g. GitHub). See the Nature Portfolio guidelines for submitting code & software for further information.

# Data

Policy information about availability of data

All manuscripts must include a data availability statement. This statement should provide the following information, where applicable:

- Accession codes, unique identifiers, or web links for publicly available datasets
- A description of any restrictions on data availability
- For clinical datasets or third party data, please ensure that the statement adheres to our policy

All raw sequencing data (HiFi, Hi-C, RNA, CENH3-ChIP, DNA methylation, SCO of pollens) and genome assembly of Rosa canina S27 isolate are available under NCBI BioProject: PRJNA1111045. The chloroplast genome of Rosa canina S27 isolate available under GeneBank accession number PV550499. Raw sequencing data of Rosa canina DToL and Rosa agrestis DToL are available from Darwin Tree of Life (DToL) data portal. The corresponding NCBI BioProject accession numbers are PRJEB79802 and PRJEB79880, respectively. Genome assemblies, the sample-specific SCO reference sequences, variant calling format files, annotations and alignments presented in this work are also made available for download at DRYAD: https://doi.org/10.5061/dryad.cc2fqz6fh. The REXdb database Viridiplantae v.3.0 [http://repeatexplorer.org/?page_id=918] is publicly available. All other data needed to evaluate the conclusions in the paper are provided in the paper and/or the supplemental information.

# Research involving human participants, their data, or biological material

Policy information about studies with human participants or human data. See also policy information about sex, gender (identity/presentation), and sexual orientation and race, ethnicity and racism.

| Reporting on sex and gender | n/a |
|---|---|
| Reporting on race, ethnicity, or other socially relevant groupings | n/a |
| Population characteristics | n/a |
| Recruitment | n/a |
| Ethics oversight | n/a |

Note that full information on the approval of the study protocol must also be provided in the manuscript.

# Field-specific reporting

Please select the one below that is the best fit for your research. If you are not sure, read the appropriate sections before making your selection.

☒ Life sciences ☐ Behavioural & social sciences ☐ Ecological, evolutionary & environmental sciences

For a reference copy of the document with all sections, see nature.com/documents/nr-reporting-summary-flat.pdf

# Life sciences study design

All studies must disclose on these points even when the disclosure is negative.

| | |
|---|---|
| Sample size | Sample-size calculation was performed based on assessment of the literature in the field, our own experience from previous studies and requirement for corresponding protocols. For Immunocytochemistry and in situ hybridisation analyes sample size was based on the number of cells obtained. The size of the sample used was performed according to the requirements for each protocol. For cytological analysis, different roots and anthers were collected and analysed to confirm the reproducibility of the results. For sequencing, sufficient coverage (>20x) was used to assemble and scaffold the R. canina genome. The sample size used for all experiments provided sufficient resolving power. |
| Data exclusions | No data was excluded form the analysis. |
| Replication | Cytogenetic analyses were performed on several cells, using the best superposition for the final figure. Experiments were independently repeated at least ten times with similar results, in order to track all meiotic stages. |
| Randomization | A randomization is not relevant for this study because no genotype or treatment were compared with each other. However, the tissues for cytogenetic and ChIPseq experiments were randomly collected from different plant individuals grown under the same condition in a greenhouse. |
| Blinding | The experiments were performed without knowing the final results. |

# Reporting for specific materials, systems and methods

We require information from authors about some types of materials, experimental systems and methods used in many studies. Here, indicate whether each material, system or method listed is relevant to your study. If you are not sure if a list item applies to your research, read the appropriate section before selecting a response.

## Materials & experimental systems

| n/a | Involved in the study |
|---|---|
| ☐ | ☒ Antibodies |
| ☒ | ☐ Eukaryotic cell lines |
| ☒ | ☐ Palaeontology and archaeology |
| ☒ | ☐ Animals and other organisms |
| ☒ | ☐ Clinical data |
| ☒ | ☐ Dual use research of concern |
| ☐ | ☒ Plants |

## Methods

| n/a | Involved in the study |
|---|---|
| ☐ | ☒ ChIP-seq |
| ☐ | ☒ Flow cytometry |
| ☒ | ☐ MRI-based neuroimaging |

## Antibodies

| | |
|---|---|
| Antibodies used | Customized Rosa canina-specific antibodies generated in this study:<br>rabbit anti-CENH3 (AB016310, LifeTein, generated in this study; dilution 1:500)<br>Commercially available antibodies:<br>mouse anti-alpha Tubulin (Sigma-Aldrich, St. Louis, MO; catalogue number T6199, Clone: MABI 0301, dilution 1:200)<br>rabbit anti-Histone H3 (Active Motif, cat. No. 39064, ChIP only, 1µg used)<br>Previously designed antibodies:<br>rabbit anti-KNL1 (AB015677-3; GenScript, Piscataway, NJ, USA; Oliveira et al. 2024, dilution 1:500) |
| Validation | Newly validated antibodies:<br>anti-Rosa canina CENH3 antibody was generated by the company LifeTein and validated by peptide ELISA tests and ChIP experiments. ELISA test information is available upon request. Furthermore, the observed indirect immuno-signals of anti-CENH3 on R. canina cells are compatible with centromere data previously reported in the published literature for other species.<br><br>Previously validated antibodies:<br>KNL1 was previously validated by Oliviera et al. (2024) by Immunostaining and Western blot.<br><br>Validation by commercial providers:<br>mouse anti-alpha Tubulin (validation; https://www.sigmaaldrich.com/DE/de/product/sigma/t6199?srsltid=AfmBOopZVI8rakd6EyMi2t1B9KcuqVPwcg6UFkhi3RDxHotJnOjCzZQW) |

rabbit anti-Histone H3 (validation: https://www.activemotif.com/documents/tds/39763.pdf; https://www.activemotif.com/catalog/details/39763)

# Plants

| Seed stocks | Rosa canina plants were cultivated under controlled greenhouse conditions (16h daylight, 26 °C, >70% humidity). |
| --- | --- |
| Novel plant genotypes | n/a |
| Authentication | n/a |

# ChIP-seq

## Data deposition

☒ Confirm that both raw and final processed data have been deposited in a public database such as GEO.

☒ Confirm that you have deposited or provided access to graph files (e.g. BED files) for the called peaks.

**Data access links**
*May remain private before publication.*

https://www.ncbi.nlm.nih.gov/sra/?term=SRR32424402

**Files in database submission**

6919_A_run867_AGGTTCCT_S30_L001_R1_001.fastq.gz
6919_A_run867_AGGTTCCT_S30_L001_R2_001.fastq.gz
6919_A_run867_AGGTTCCT_S30_L002_R1_001.fastq.gz
6919_A_run867_AGGTTCCT_S30_L002_R2_001.fastq.gz
6919_B_run867_GAACCTTC_S31_L001_R1_001.fastq.gz
6919_B_run867_GAACCTTC_S31_L001_R2_001.fastq.gz
6919_B_run867_GAACCTTC_S31_L002_R1_001.fastq.gz
6919_B_run867_GAACCTTC_S31_L002_R2_001.fastq.gz
6919_C_run867_AAGTCCTC_S32_L001_R1_001.fastq.gz
6919_C_run867_AAGTCCTC_S32_L001_R2_001.fastq.gz
6919_C_run867_AAGTCCTC_S32_L002_R1_001.fastq.gz
6919_C_run867_AAGTCCTC_S32_L002_R2_001.fastq.gz
6919_D_run867_CCACAACA_S33_L001_R1_001.fastq.gz
6919_D_run867_CCACAACA_S33_L001_R2_001.fastq.gz
6919_D_run867_CCACAACA_S33_L002_R1_001.fastq.gz
6919_D_run867_CCACAACA_S33_L002_R2_001.fastq.gz
6919_E_run867_ATAACGCC_S34_L001_R1_001.fastq.gz
6919_E_run867_ATAACGCC_S34_L001_R2_001.fastq.gz
6919_E_run867_ATAACGCC_S34_L002_R1_001.fastq.gz
6919_E_run867_ATAACGCC_S34_L002_R2_001.fastq.gz
6919_F_run867_CCGGAATA_S35_L001_R1_001.fastq.gz
6919_F_run867_CCGGAATA_S35_L001_R2_001.fastq.gz
6919_F_run867_CCGGAATA_S35_L002_R1_001.fastq.gz
6919_F_run867_CCGGAATA_S35_L002_R2_001.fastq.gz

**Genome browser session**
(e.g. UCSC)

no longer applicable

## Methodology

| Replicates | Two biological replicates of CENH3 sequenced as 6919_A and 6919_B; Two biological replicates of H3 sequenced as 6919_C and 6919_D; Two biological replicates of input control sequenced as 6919_E and 6919_F. |
| --- | --- |
| Sequencing depth | 6919_A: 21,902,101 pairs of reads (150bpx2); 30,512,245 reads uniquely mapped<br>6919_B: 21,890,693 pairs of reads (150bpx2); 31,961,379 reads uniquely mapped<br>6919_C: 21,798,217 pairs of reads (150bpx2); 34,691,504 reads uniquely mapped<br>6919_D: 21,798,366 pairs of reads (150bpx2); 34,835,465 reads uniquely mapped<br>6919_E: 21,304,165 pairs of reads (150bpx2); 33,647,990 reads uniquely mapped<br>6919_F: 21,682,527 pairs of reads (150bpx2); 34,367,915 reads uniquely mapped |
| Antibodies | The CENH3 gene of Rosa canina was identified using BLASTP with the parameter "-evalue 1e-5 -qcov_hsp_perc 50" and the A. thaliana CENH3 protein HTR12 (AT1G01370) was used as the reference. A specific polyclonal antibody against its CENH3 protein (ARVKHTAARKDRIKTARRQP-C / AB016310) was designed and synthesised by LifeTein with immunisation in rabbits. |
| Peak calling parameters | Genome indexing and read mapping were both done by bowtie2 (v2.5.4) with `--sensitive-local` flag activated for all experiments. |

| Peak calling parameters | After comparing the CENH3 domains with H3 or input samples woth bamCompare, peak calling was performed using the MACS3 pipeline with the inclusion of parameters --broad -g 1.9e+9. |
| --- | --- |
| Data quality | Mapped reads were not filtered by mapping quality considering the high similarity of four subgenomes. 251 out of 6105 broad peaks were 5 fold enriched when setting FDR value as 0.05. |
| Software | The raw 150bp pair-end ChIP -seq reads were checked by FastQC and then mapped to the R. canina haplotype phased reference genome using bowtie2 (as described in "Peak calling parameters"). The BAM file was converted to bigwig using the bamCompare tool from deeptools2, and then normalized to RPKM (reads per kilobase of transcript per million reads mapped). After this a peak calling, comparing the CENH3 domains with H3 or input samples, was performed using the MACS3 pipeline with the inclusion of parameters --broad -g 1.9e+9. |

# Flow Cytometry

## Plots

Confirm that:

☒ The axis labels state the marker and fluorochrome used (e.g. CD4-FITC).

☒ The axis scales are clearly visible. Include numbers along axes only for bottom left plot of group (a 'group' is an analysis of identical markers).

☒ All plots are contour plots with outliers or pseudocolor plots.

☒ A numerical value for number of cells or percentage (with statistics) is provided.

## Methodology

| Sample preparation | Nuclei of mature pollen grains were isolated by applying the filter bursting method63 using the nuclei isolation buffer according to Galbraith et al. (1983)64. Pollen grains were burst on the surface of a 20 μm disposable CellTrics filter (Sysmex-Partec). The resulting nuclei suspension was stained with propidium iodide (50 μg/ml, PI) and run on a BD Influx cell sorter (BD Biosciences). After identifying the nuclear populations in a dotplot displaying the PI fluorescence signal (log-scale) versus side scatter signal (SSC, log-scale) a sort gate was defined in the corresponding fluorescence intensity (lin-scale) histogram. Per individual 200,000 generative nuclei (volume ca. 400 μL) were collected into a 1.5 ml reaction tube using the '1.0 Drop Pure' sort mode of the BD FACS Software (BD Biosciences). After adding 50 μL 1× TE and 50 μL NaN3 nuclei were sedimented by centrifugation (1000 × g for 10 min at 4°C). Afterwards 300 μL of the supernatant was removed and the nuclei with the remaining liquid stored at -20°C |
| --- | --- |
| Instrument | BD Influx cell sorter |
| Software | BD FACSDiva™ Software (v9.0) |
| Cell population abundance | Per individual 200,000 generative nuclei (volume ca. 400 μL) were collected into a 1.5 ml reaction tube using the '1.0 Drop Pure' sort mode of the BD FACS Software (BD Biosciences). |
| Gating strategy | After identifying the nuclear populations in a dotplot displaying the PI fluorescence signal (log-scale) versus side scatter signal (SSC, log-scale) a sort gate was defined in the corresponding fluorescence intensity (lin-scale) histogram. |

☒ Tick this box to confirm that a figure exemplifying the gating strategy is provided in the Supplementary Information.

