## [Peer Review File · Nature]

Bimodal centromeres in pentaploid dogroses shed light on their unique meiosis

Corresponding Author: Dr André Marques

Parts of this Peer Review File have been redacted as indicated to remove third-party material

Please note that there is a attachment of reviewer 3 initial version at the end of the file.

Version 1:

Reviewer comments:

Referee #1

(Remarks to the Author)

This study assembles the dog rose genome and reveals an unusual pentaploid genome, as well as striking mixed univalent and bivalent modes of inheritance during meiosis, which associates with distinct centromere structures and sequence composition. This reveals a previously unknown genome architecture. Although the genome architecture is unusual, it has important implications for genome organisation, evolution and centromere function. I found the most interesting aspect is the 'drive or die' model, but further evidence would be required to support this.

Rosa canina (dog rose) is unusual in having 7 chromosomes that show typical bivalent inheritance through meiosis, in addition to 21 chromosomes that show univalent inheritance and appear to show strong maternal drive. HiFi sequencing and Omni-C data are used to assemble the 2.4 Gb genome – this appears to be performed well and carefully. The authors combine this analysis with comparison of gene orthology, kmers, and LTR divergence to develop a model for the hybridization history of dog rose. While the specifics of the hybrid and polyploid origin of this genome are studied in great detail, I personally found some of these details less interesting and are less likely to be general interest in my opinion, apart from having a pentaploid structure, which is unusual.

The most interesting aspects of the study are that the univalent and bivalent chromosomes have markedly different repeat sequence compositions, and this correlates with very distinct inheritance behaviour through male or female meiosis. In particular, the univalents appear to show obligate maternal drive. This is certainly the most novel aspect of the study, but the model presented is also the most speculative aspect, and I feel further work needs to be performed to prove the model.

The bivalent chromosome centromeres are shown to be dominated by ATHILA retrotransposons, whereas the univalent centromeres are dominated by satellite CANR4 repeats. Using KNL1 immunostaining evidence is provided that the univalent centromeres have 'smaller' kinetochore foci. It would be beneficial if the 'size' of the centromere could be studied in further ways – eg ChIP-seq of CENH3, or possibly the KNL1 protein, could be informative about the amount and arrangement of kinetochore loading on the bivalent versus univalent chromosomes. Is anything known about the relative epigenetic states, specifically DNA methylation, on the univalent versus bivalent centromeres?

This difference in centromere sequence, and presumably kinetochore loading, is then correlated with regular versus driving inheritance in female meiosis. Minimally, further study of female meiosis needs to be performed to confirm for example that cell death occurs in 3 of 4 meiocytes. This presumably must be associated with some cellular or spindle asymmetry that the CANR4 centromeres can sense somehow? Further data should be provided to compare CANR4 centromere behaviour in male versus female meiosis to prove their model, which would be novel and of wide interest and significance in the centromere field.

An interesting evolutionary question is what is the sequence composition and location of centromeres in *Rosa chinensis* and *Rosa rugosa*? Are their centromeres ATHILA or CANR4 dominated and are they located in syntenic regions?

In terms of repeat analysis, reading the methods, it seems that Illumina sequencing followed by RepeatExplorer2 analysis was performed? This seems surprising given a long read genome was produced – would it not be better to analyse the assembly directly using the wide range of transposon and tandem repeat mapping tools available? Indeed it seems that the LTR elements are classified into the main families present in plants, but this process is not explained clearly in the methods,

or does RepeatExplorer2 provide this information? Further detail on tandem repeat and transposon identification should be provided in the Methods and explained in the main text briefly.

Figure 1 seems rather simplistic for a main figure, and appears to summarise published knowledge. Perhaps this could be combined with Figure 2?

Minor points – when I first read the title I read ‘odd’ as meaning strange, rather than numerically odd. One alternative might be to use pentaploid instead?

This sentence is confusing – particularly the usage of ‘non-random’ and ‘highly specific’. ‘As expected, the number of SNPs observed in pollen 234 samples was considerably lower than in the respective somatic tissues (Figure 6b), which supports the 235 hypothesis that bivalent formation is non-random but highly specific.’

Referee #2

(Remarks to the Author)

This manuscript describes sequencing of a polyploid plant with a unique chromosomal architecture that permits the retention of univalent, and thus non-pairing and non-recombining, subgenomes. This makes possible a fascinating exploration of the consequences of carrying both sexual and non-sexual chromosomes simultaneously. The work is competently done, and the data is well presented. As is the nature of purely descriptive work, it lacks tested hypotheses. For instance, the bimodal distribution of centromeric repeats is fascinating but purely correlative. I'd be interested in the authors idea of a test of the hypothesis that the repeats are providing a distinct meiotic drive system. The writing is generally serviceable, although I think it places the points could be made with more clarity. I've suggested some example in my detailed comments, some of which are likely due to my misunderstanding of the text, but should help the authors by identifying text that could be rephrased for clarity.

Line 68. Just so you don't lose non-specialist, can you clarify what you mean by tetrasomically inherited means. That pairing and recombination can occur between any one of the homoeologous chromosomes and that all four copies are reliably inherited?

Line 72. Again, it would be helpful to clarify exactly what you mean here for the non-specialist. As in, “because...x, y”. Here, you are saying that preferential pairing can lead to apomyxis with respect to one set of chromosomes and not another, but you don't make clear why this is. You don't want your reader to get lost so early in your explanation.

Line 78. Ah, so preferential pairing only matters when you have an odd number of chromosomes. If this is the case, perhaps you could set up this section saying that preferential pairing isn't a big deal if you have an even number, but it is necessary with an odd number?

Line 88. Among eukaryotes that have been examined.

Line 90. I think you could be more explicit about what it is you are investigating. Certainly, preferential pairing is a must, but the biased apportionment of the univalent is also very interesting. The way this is worded suggests to me that this question wasn't really addressed.

Line 127. Any evidence for recombination between the more divergent chromosomes early following polyploidization?

Line 144. I'm confused. I thought they were nearly identical, although I guess they could be nearly identical and scrambled relative to each other?

Figure 2e. Perhaps a bit more explanation of the y axis here? I'm sure it is in the M&M, but a brief description would be helpful. Syntenic regions can be defined in many different ways, which makes it difficult to know what “proportion of syntenic regions” means. For instance, I would say that an inversion only changes synteny at the boundaries of the inversion, but you could also say the whole region is now out of synteny. Also, what does the shading in the boxes in the box plots indicate?

Line 153. Why is this remarkable?

Line 167. Again, why is this remarkable? Aren't these gene poor regions that are much more likely to accumulate differences?

Line 173. What does “similarly high levels of BUSCO genes” mean? BUSCO was mentioned earlier, but it isn't clear what you mean by it here.

Line 174. How do you know it wasn't an ancient autopolyploidy, such as hypothesized for soybean?

Line 181. “...between the estimated S1/2-s3-R5 from R4 subgenome-specific LTR-RTs insertion times, ranging from...” Awkward wording. between the estimated what? divergence times? How does a boundary range from x to y?

Line 183. Last, not “at last”.

Line 184. Can you make this assertion without reference to the known progenitor of each subgenome? That is to say, can you make any assertion about the timing of the allopolyploidy just by looking at divergence of the subgenomes?

Line 186. I think you mean the number of substitutions, not the rate, which generally refers to time. Also, rather than a differential rate, perhaps you could simply talk about degree of divergence. The writing here is unnecessarily confusing. I'm also not clear about the point being made here.

Line 193. I'm not sure reference 39 actually shows this. Many plant genomes have very large proportion of retro-elements. Reference 39 shows phylogenies of these, but I don't see a discussion of relative amounts of different types in different genomes.

Line 201. Again, not remarkable. There are many examples of both types of repeats being enriched in centromeres.

Line 205. Awkward wording. I think you mean, “Although tandem repeats are common in the S1/S2 genomes, they are not enriched in (peri)centromeres.”

Line 209. How are you defining centromeres here?

Line 226. The introduction would have benefited with a short discussion of this kind of meiotic drive.

Line 263. “Thus, bivalent-forming chromosomes in subsect.” is not a sentence and is unclear. Perhaps this is a term of art, but I don't understand what subsect means here.

Line 273. Or just differences in divergence times?

Line 299. Of course this meiotic drive is also likely to have negative consequences given that lack of recombination, as mentioned later. It is surprising that there aren't higher levels of fractionation in the univalent, given that the drive mechanism retains the centromere, not necessarily the genes on the chromosomes. Is expression data available? I would expect homoeologous genes on the univalent to express at lower levels, on average, and they should also have higher K_a/K_s values.

Referee #3

(Remarks to the Author)

Key results:

In this manuscript, Herklotz et al, made crucial step into solving the mystery of how pentaploid dogroses (*Rosa canina*) maintain their odd ploidy level across generations of sexual reproduction and going through meiosis. In this work the authors assembled a haplotype-phased genome of *R. canina* and uncovered the presence of four distinct constitutive subgenomes. This high-quality genome allowed a detail profile of the repeatome of each subgenome and a description of their satellite repeats and transposable elements, plus an analysis of the centromeres. In addition, this data unlocked an analysis of the origin and phylogenetic relationships between the constitutive subgenomes and related species. Importantly Sequencing of pollen nuclei enabled a robust identification of the subgenome(s) that form bivalents during male meiosis, conformed by two sets of highly syntenic and homozygous chromosome sets. Finally, the authors also provided a description of the localization patterns of KNL1 in mitotic chromosomes.

Validity:

As implied in the abstract, this work address very well one of the unknowns of dogrose reproduction: its chromosome properties. Remarkably, the heroic assembly of an extremely complex genome such as that of dogrose deserves credit, along with the robust and highly convincing identification of the bivalent- and univalent-forming genomes. Unfortunately, it was not possible—in my opinion—to provide valid and decisive insights into the unique chromosome segregation of this system. For further explanation see the Conclusions section below) Nevertheless, this goes way beyond an incremental progress and I strongly believe that the genome biology deciphered here, along with the genome references and other resources makes this work a qualitatively important milestone that will be essential to decipher this exciting mystery in future.

Originality and significance:

Although previous work already gave an idea of the nature of bivalent- and univalent-forming genomes (Kovarik et al., 2008; Lim et al., 2005) and the phylogenetic relationships with other species (Herklotz et al., 2018), the contribution to these important questions of this work is on completely different scale.

Data and methodology:

Overall, I am satisfied with the methodology of this work, especially regarding the aspects of the genome biology of *R. canina*. However, I believe that to address the chromosome biology mysteries, the mechanisms underlying non-symmetrical meiosis, a deeper cytological work might have been beneficial. For instance, immunostaining of the synaptonemal complex protein ZYP1 could have provided useful information into the level of suppression of recombination and pairing (the strict definition of pairing, see below in the Clarity section) during early meiosis for univalent-forming chromosomes. Moreover, a follow-up of the fate of univalent-forming chromosomes from anaphase I until pollen mitosis (when the authors nicely demonstrated that the univalent chromosomes are already excluded). This could have been achieved, marking some centromeres with CARN4 FISH or all of them by centromere-specific histone variant CENH3. Also, using these cytological markers in metaphase I could have been a nice validation that the regions identified as centromeres do show centromeric activity (e.g. stretching during metaphase I). Finally, I wonder if sequencing of endosperm is possible in this system and which constitutive genomes would be present in this tissue. This could provide some insights into the meiotic drive mechanism that in female meiosis makes one copy of each chromosome from each subgenome present in the egg cell.

Appropriate use of statistics and uncertainties:

Overall, statistics is properly used in this manuscript. I would maybe improve it by including a method section on statistics even if there are not a lot of test. I think this section is required to state things like that the homogeneous variance across groups were verified before proceeding to test like ANOVA/Tukey require (as in L146). Apart from this I have a couple of minor comments that I include among suggested improvements.

Conclusions: robustness, validity, reliability:

I missed further justification of how regions were assigned as centromeres. In other studies CENH3-ChIP was used (Ahmed et al., 2023; Hofstatter et al., 2022; Naish et al., 2021). I believe this is not a trivial question because, centromere position sometimes is not obvious simply from genomic signatures are not always an ultimate proof of centromere location (Liu et al., 2023). Also, based on the text and Figure 4 and supplementary dataset 2, it is not clear whether the localization of CARN4 is centromeric, or peri-centromeric (Lunerová et al., 2020 described it as “strictly peri-centromeric”).

One of the conclusions stated in the abstract (L45-47), is that “Distinct structural features of centromeres suggest a role in the segregation bias of univalents during asymmetric female meiosis...”. I have some concerns regarding the validity and relevance of those distinct features. First, it is explained that CARN4 repeats show preference for univalent-forming chromosomes (implying a potential role in their mysterious segregation). However, I do not understand why, as I feel that preference is not that strong or relevant. As an example. It is mentioned in L209-211 that “Large centromeric arrays of CARN4 were found in 16 out of 21 univalent chromosomes from the S3, R5 and to a lesser extent R4 subgenomes, while six (in syntenic group 2, 4, 5) out of 14 centromeres in the S1/S2 bivalent-forming chromosomes”. This means that in univalent-forming chromosomes ~5.3 chromosomes per genome show large CARN4 centromeric arrays while in bivalent-forming ones ~3 chromosomes per genome carry the centromeric arrays. Figure 4, showing the example of the homoeologous group 6, gives the impression that there is a clear preference of CARN4 arrays for univalent-forming chromosomes. However, this does not hold true for any other homoeologous groups (see Supplementary dataset 2). Indeed, if we carefully observe each genome separately, those arrays are only in 2 out of 7 chromosomes in the univalent-forming genome R4. Moreover, the fact that different chromosomes show distinct centromere repeat organizations has already been observed in other models like soybean (Liu et al., 2023) and maize (Gent et al., 2017) without that implying functional different during segregation. Therefore, I have problems to see a trend for CARN4 expansion based on whether chromosomes form bivalents or univalents, and the hypothesis that these arrays drive the differential behavior of the genomes during meiosis gets less likely.

Similarly, I wonder if the differences in KNL1 signal size could be simply due to differences in centromere size. For instance, based on Supplementary Dataset 2, the size of most centromeres is about 2Mb, but it can range from 4 (Chromosome 3 of genome R5) to 0.5 (Chromosome 5 of genome S2). Actually, if we assume that CARN4 is at centromeres (please see Clarity section below), it might seem that the size of the FISH signal does vary substantially among chromosomes. In any case, I believe that some quantification of the KNL1 signal intensity or sharing additional images in the supplementary might reinforce the observation.

References:

I found the use of references appropriate in this manuscript.

Clarity and context:

Overall, the manuscript was clear, but I have a few comments on this regard.

First of all, although this is probably a matter of taste, I found the first and second sections of the results (specially the first one) a bit too technical for the main text of an article in general journal like this. I would rather prefer to use the main text to focus on the genome biology and leave some of the technical metrics for the supplementary (maybe using tables where all the assembly metrics are gathered, or supplementary notes if descriptions are required). However, I fully understand that assembling a so complex genome like this calls for sharing the details (two good examples of this are Ahmed et al., 2023 and Hofstatter et al., 2022). If the authors prefer to keep the technical tone of the first part of the results I would suggest to include a few more sentences explaining the implications of the metrics (I have included a few suggestions in this regard in the suggested improvements below).

In addition, I have some issues with the use of the word pairing. My understanding is that there is a consensus in the meiosis community nowadays that pairing is the alignment between homologous regions of homologous chromosomes during early stages of meiosis that precedes synapsis and crossover formation (Zickler & Kleckner, 2015, 2023). According to this definition, we cannot speak about metaphase I bivalents using the word "pairing". I know that, traditionally pairing was used for metaphase I bivalents, especially among plant cytogeneticist. However, I would say that some statements with this use of the term can be imprecise. For instance, it would be imprecise to say that homoeologous pairing is prevented in allopolyploids like wheat and oilseed rape. Yes, metaphase I bivalents are only formed between homologs (and prevented between homoeologs) but pairing (*sensu stricto*) does occur between homoeologs during early meiosis (Grandont et al., 2014; Martinez et al., 2001). Similarly, the results of this manuscript support that bivalent formation occurs between homologs in dogrose metaphase I, but it is totally possible that pairing (*sensu stricto*) is also happening at early stages of meiosis between homoeologs (as in wheat and oilseed rape). There is a clear explanation of this distinction in Lloyd & Bomblies, 2016 (Box 1). therefore, I would suggest to differentiate between the two terms: pairing and bivalent formation at metaphase I. Or at the very least, clarify the definition of pairing that the authors will be using.

Finally, sometimes I get confused about subgenomes S1 and S2. In the abstract it is clearly stated that this work uncovered four subgenomes. However, at the beginning of the results it is explained that the sequencing data was assembled into 35 chromosomes. This means that the chromosomes from the S1 and S2 genomes were assembled separately. However later they are always mentioned as a one S1/S2 and explained as highly homozygous. Could the Ks analysis distinguish between the two s1 and s2? I would not expect so, and indeed, from the text I assume that the k-mer approach did not distinguish between S1/S2... In such a homozygous context, could the two genomes have been assembled without haplotype phasing (that is explained later)? Another contradiction is in Supplementary Dataset 2, where the chromosomes S1 and S2 are shown separately (although it becomes obvious that not many differences exist between them). Also, in L 127 "chromosome-based phylogenies" are mentioned as a method to allocate each chromosome to their respective subgenome, but I could not find further details in the methods. In addition, the genomes S1 and S2 look pretty empty in the heatmap of the figure S1a. More importantly, since the subgenomes S1 and S2 form bivalents and segregate, I wonder if they can be called subgenomes. I guess that it could be the case that some plants do not have any S2 chromosome at all (i.e. they are S1/S1/S3/R4/R5), right? If that is the case, wouldn't it make more sense to directly say that S1 and S2 are simply haplotypes of the same S1/S2 subgenome (subgenome in singular)?

Suggested improvements: Main text

-L34 "...achieved by forming seven bisexually inherited bivalents..." I would say that this is a bit misleading; bivalents cannot be inherited. Since a bivalent is the structure formed by two chromosomes bounded by chiasmata, this sentence can be misinterpreted as both chromosomes segregating together toward the same pole. I would rather say "seven bisexually inherited bivalent-forming chromosomes".

-L70 are-> there is??

-L51 "...X in the figure."

-L77-80. The text says: "bivalent-forming chromosomes are passed on to the pollen grains and egg cells, whereas unpaired chromosomes (univalents) are passed on to the egg cells but are excluded from viable pollen nuclei." What is that exclusion mechanism? I wonder if a sex-specific genome elimination of particular chromosomes is totally ruled out. I bring this up because, the univalents seem to segregate randomly in anaphase I (Lim et al., 2005). The results of this work, seem to virtually do rule it out for the male side (unless it happens in the short lapse between meiosis and pollen formation), but I wonder if previous literature consider that possibility.

-L86. Could it be that the words "and there" should not be there?

-Figure 1. I just had in mind a slightly different interpretation of the bivalents and univalents. Please check image attached. I know that the authors have a lot of experience in quantifying DAPI signal to figure bivalents and univalents out. This is just a suggestion.

-L107. It is not clear if only 35 scaffolds were assembled. Maybe a table summarizing all the assembly metrics in the supplementary would help.

-L111-112. I would suggest to clarify the meaning of some of the results presented, same as the BUSCO score is shown as a completeness indication, the same explanation should be given for Mercator4 annotation. I think this is important in a generalist journal like this; many non-genomics experts will read this manuscript.

-L114. Is it possible to express that these parameters indicate completeness of the "single copy" genome? From those we don't know how completed are duplicated regions...

-L122-123. I am a bit confused with these lines. I feel that the words linkage and synteny are used as synonyms here.

However, linkage can also refer to genetic linkage (genetic maps) which are also used for other assembly strategies such as trio binning.

-L122-123. Again, I would include a sentence summarizing the conclusion from that data, for instance "...this result uncovered the identity of the 5 homoeologs for each synteny group".

-145-146. I would include a reference to figure S2e

-Figure 2. I would be a bit more explicit about the color code

-L146. I would mention that ANOVA was done first.

-L151. Please make a reference to the figure where the highest synteny of S1-S2 is shown, and where the letters AB and X are shown

-L151-155. I do not understand why the comparisons were denoted with letters. Is it based on the statistical difference significance?

-L161-169. I find this part more related to the subgenome assignment already explained in the previous section. Also, a

couple of explanatory sentences justifying the k-mer approach, and how it is needed to identify subgenome-specific signatures might be positive.

-L178. I missed the information about the insertion time calculation in the methods.

-L199. When Figure S8 is mentioned, did the authors mean Figure S9, instead?

-Figure 4. Please explain that the meaning of the identity heatmaps. Also explain the color code for the density plots are the same for the heatmaps. According to my interpretation of the color code, the identity between two random windows is on average about 90%... Isn't that too much for the chromosome arms? Also, a little detail: in panel C the axis of the histogram of chromosome 6 of S3, is missing some text.

-L210-211. "As frequent across both S1 and S2 chromosomes, we did not find any obvious enrichment for tandem repeats but a high enrichment for ATHILA elements at their centromeres" I don't understand why it is said that there is no obvious enrichment. There are regions with obvious overrepresentation of some repeats in figure 4 and Supplementary dataset 2 in all the chromosomes of the S1 and S2 genomes.

-L231. I would include a bit more information about these species as most readers would not know if they are diploids, tetraploid, pentaploids, etc. Also, I would also explain in the text, although that the information can be found in the Figure 6d, what phylogenetic relationship those species have with the constitutive subgenomes of *R. canina*. I believe this would facilitate the understanding of the controls and results presented.

-L232. I think this is the first time the acronym SCO is used in the main text, please spell it out.

-L235-236. The text says "the number of SNPs observed in pollen samples was considerably lower than in the respective somatic tissues (Figure 6b), which supports the hypothesis that bivalent formation is non-random but highly specific". I think that if bivalents were formed by synthetic pairs of random homoeologous chromosomes, and therefore, pollen was formed by random genomes, then the number of SNPs found would be equally lower than in leaf sample. I think that, if I understood the approach properly, what the lower number of SNPs indicates is that the pollens contain much less chromosomes than the (half of) the leaf sample, which strongly support that only a few chromosomes segregate to the future sperm cells. I think that the proof that bivalents are formed by two specific chromosome sets is more strongly supported by the results presented subsequently (L236-267).

-Figure 6. Why in the panel C subsections names are shown instead of the species cited in L231.

-L242. I am a bit confused. This is the first time that *R. pseudoscabriuscula* is cited, but it is not mentioned in L231. Also, subsect *Vetistae* is mentioned in Figure 6C. Is *R. pseudoscabriuscula* actually *R. sherardii*? Why *R. corymbifera* is not shown in Figure 6C?

-L258-259. I think that this contradictory and unexpected result calls for some some explanation or hypothesis. It might help to repeat the analyses with chromosome-wise concatenated SCO loci (instead of genome-wide). Also, I think it should be explained in the methods section what is the origin and relationship of these plant materials.

-L265-266. I think that this conclusion can be only understood once it is also understood that the species whose pollen was analyzed are also pentaploid with the same asymmetric meiosis mechanism. As mentioned above, please explain the ploidy of those species.

-L271-273. Regarding the interpretation that: "This difference in branch lengths within our phylogenetic framework might reflect the differential evolutionary selection pressures with respect to their ability or lack of meiotic pairing and recombination.", could not also be a matter of how related they are with their closest branch?

-L294-295. Again, not sure of this conclusion.

-L308-309. "However, the high level of BUSCO completeness in all subgenomes does not indicate a significant loss of protein-coding genes from univalent chromosomes, suggesting their potential functionality." Actually, fractionation is normally a property of duplicated genes in polyploids (by definition non-BUSCO). I wonder if the authors could say something about fractionation in NON-BUSCO.

-L380. It is not clear what these diploid rose DNAs were used for.

-L390-392. Also, please explain the ploidy of those species here in the methods.

-

Suggested improvements: Supplementary

-L55: would it be possible to summarize in one sentence what information is provided by this plot? I am talking specially about the A/B etc...

-Figure S3. Further explanation of the figure should be provided in the caption of this figure, in my opinion. What is this dot plot expressing? Where is this data coming from? What kind of linkage is plotted? Genetic linkage? Synteny? Identity? Hi-C contacts? What is the meaning of the comparisons highlighted in blue?

-Figures S4 and S5. Please, explain more explicitly where the color code is and include a Y-axis label in the plots on the right.

-Supplementary dataset 1. Please verify if Mann-Withney test is appropriate in this situation. My understanding was that in this context a Kruskal-Wallis plus Dunnett's test.

Final suggestion. (the authors are not required by any means to do this). I really wonder if the bivalent- and univalent-forming genes evolve differently... Would it be possible to analyze K_a/K_s patterns in both kinds of genomes.

Version 2:

Reviewer comments:

Referee #1

(Remarks to the Author)

Thank you – the authors have addressed my concerns well, both by rewriting the text, and adding the new CENH3 ChIP-seq and immunostaining data, as well as the work on synthetic hybrids. Together these new data provide strong support for the proposed model of univalent and bivalent inheritance, and provide a much higher resolution picture of centromere structure and function.

Abstract line 39 – should this be ‘role of centromeres’? The use of ‘chromosomes’ here doesn't seem quite right to me.

Referee #2

(Remarks to the Author)

The authors have done an excellent job responding to all of my comments. I have no further issues with this manuscript.

Referee #3

(Remarks to the Author)

After working on this version of the manuscript, I really appreciate the changes, and I feel the manuscript is more readable and better explained. I am also very happy about the new results included, particularly the inclusion of new genomes, the CENH3 ChIP-Seq, and especially the cytology in male meiosis. However, I still have important concerns about the main claim that differences in centromere structure between subgenomes explain their different behaviors during meiosis.

I am aware that the text shows some caution regarding this claim in some parts (for instance, in L361-362: “CANR4 enrichment alone does not fully explain univalent drive.”), but other statements are, in my opinion, not as cautious. For instance, L50-51: “These structural differences likely drive the asymmetric segregation of univalents during female meiosis” or L352-354: “The dominance of CANR4 repeats in univalent centromeres suggests a potential mechanism for centromere drive – a phenomenon in which certain centromeres are preferentially transmitted during meiosis – leading to biased segregation (Figure 6a).” Also, in L362-366: “the structural divergence of centromeres is also closely linked to their behavior in male meiosis.”

Future results might prove me wrong, but with the current evidence, in my view, the association between centromeric structure and meiotic behavior is at most correlative. I made a contingency table for meiotic behavior (bivalent vs. univalent) against centromere composition (satellite-based vs. retrotransposon-based), and if I perform a statistical test (chi-square or Fisher's exact), it is not significant. This suggests that the observed combinations of meiotic behaviors with centromere compositions could occur by chance, assuming independence between the two factors. Additionally, in *R. agrestis*, where the bivalent- and univalent-forming chromosomes change, the compositions of centromeres remain the same, making it difficult to assume that one determines the other. Overall, I think the text does not clearly acknowledge the lack of evidence that differences in centromere structure between subgenomes determine their different behavior during meiosis.

On the other hand, I am okay with the word “bimodal” — I think it is a great choice, actually — because it implies a trend that is not always fulfilled. Additionally, I would be very curious about differences in the relative coverage of CENH3 ChIP reads, so that centromere function can be compared across chromosomes and subgenomes. Perhaps CENH3 abundance might be better correlated with differences in meiotic behavior.

Additional Comments

Although I am very glad that the authors included CENH3 ChIP-Seq data in this version, I think it is fair to explicitly mention the limitation of using data from leaves to study meiotic behavior.

L227-246: These crosses demonstrate that the subgenome that can be in double dosage varies depending on the parents used for the cross, but unless I am missing something, this experiment shows that bivalents are formed during meiosis. I assume that to support this claim, cytological observations of meiosis or sequencing from sorted pollen would be needed. Probably, the word “potentially” in L244 is meant to indicate this, but I think it should be clearer that whether bivalents can be formed is not directly tested.

I am wondering how exceptional the result found in CAN2 is. Could it be that the bivalent-forming subgenome is highly variable? Perhaps depending on which haploid subgenome came from the pollen donor? This would explain why the univalent-forming subgenomes do not degenerate (e.g., through fractionation and higher Ka/Ks), as all subgenomes have a chance to recombine occasionally. In other words, could it be that the claim that the bivalent-forming subgenomes are interchangeable is true not only for interspecies crosses but also within species? Of course, this is just speculation, but is there any previous data or evidence to rule out this scenario?

I also have some difficulty understanding the model of how male gametes only transmit bivalent-forming chromosomes while univalents are lost. L335-336 states: “while single chromatids from univalents lagged behind and were likely eliminated (Figure 5i–j; Supplementary Movies 9–10, n = 31).” It is not clear to me from the figure how univalents are eliminated; I just see a few chromosomes lagging behind. Since in metaphase II there are no more bivalents, only chromosomes, I do not think it is possible to determine which chromosomes originated from a bivalent and which from a univalent. Examining the telophase II stage might be more informative.

Actually, I find the observations in metaphase I in Figure 5 quite remarkable. It seems that univalents systematically display

bipolar orientation. In *Arabidopsis thaliana*, this only happens when sister chromatid cohesion is defective (D'Erfurth et al., 2009), but it appears to be more common in wheat univalents (Lukaszewski, 2010). This observation contrasts with that of female meiosis by Täckholm (1922), where univalents seem to have monopolar orientation. I wonder if it would be worthwhile to explicitly discuss the possibility that sexually dimorphic adjustments in sister chromatid cohesion regulation could be part of the adaptations enabling canina meiosis.

Minor Comments

L34: "relies on meiotic chromosome pairing" → I would add "forming bivalents" if the word count allows.

L35: "Ploidy asymmetry" is not a well-known term. The same applies to "polyploid hemisexual" in L96.

Figure S7: Some axis labels (e.g., FC, RC) are not clarified in the legend. Comparisons with DToL genomes would be interesting (e.g., *R. agrestis* R4 vs. *R. canina* R4 or S27 subgenomes vs. DToL subgenomes).

L97-99: The introduction more cautiously states the correlation: "accumulates in dogroses, especially in univalents, possibly linking centromere expansion with their drive in female meiosis."

L185-200: It would be nice to specify how many individuals from each species were used for DNA extraction. Were different *R. canina* individuals analyzed?

Figure S10: The meaning of SG1, SG2, and SG3 is not included in the legend.

Figure 3D: The "5C" label looks a bit like "6C."

L265-266: "The total centromere length per subgenome confirmed that the S2 and R3 centromeres were larger compared to those of S1 and R4." → It would be more accurate to state that the length of the centromeric repeat array is larger, as in many centromeres the repeat array is larger than the CENH3-positive region. Later, once immunostaining results are presented, would be a better time to discuss centromere size.

L267-268: "CANR4-based centromeres were also found in six (syntenic groups 2, 4, and 5) out of 14 centromeres of the S1_h1/h2 bivalent-forming chromosomes and in two R4 chromosomes." → It might be clearer to state "3/7 syntenic groups" rather than "6/14," as this better conveys the deviation from the expected pattern.

L324-325 and Figure 5e: "At onset of metaphase I, we observed seven bivalents organized at the center of the cell surrounded by 21 univalents." → This may not be metaphase I but rather pro-metaphase I or late diakinesis.

Comments on Responses to Reviewers

Figures S19 and S20: Scale bars are not explained

Comment R2.6) Line 127. Any evidence for recombination between the more divergent chromosomes early following polyploidization?

Answer to R2.6: As far as we know there is no evidence available. Also based on our data, we could not detect any sign of recombination between univalents and the bivalent-forming subgenomes. This greatly supports that hybridisation events most likely happened in an asymmetric genome contribution, with two copies of a subgenome hybridised with an additional copy of a different subgenome, potentially forming an allotriploid.

My comment -> I would emphasize this in the text.

Answer to R3.7: The reviewer is entirely right in that the R4 genome does not show expansion of centromeric satellite as opposed to R3 and S2 genomes. We now quantified CANR4 repeats in each *R. canina* chromosome including the two new genomes available from DToL. We used bivariate Bayesian Generalized Linear Mixed Model (GLMM) using Markov chain Monte Carlo techniques to statistically analyse the proportion of ATHILA and CANR4 across subgenomes in *R. canina* (S27 and DToL) and *R. agrestis* (Supplementary Figure 14). The model included subgenome as a fixed effect, while pairing type (Univalent/Bivalent) among others were set as random effects. This experiment clearly showed that the bivalent-forming subgenomes (S1 and R4) harbour ATHILA-type or mixed ATHILA/CANR4 satellite-type centromeres irrespective of genetic background. The proportion of CANR4 satellite is however low. In contrast, the S2 and R3 subgenomes, which always form univalents, show high content of CANR4 satellite. The genome space of the satellite is extraordinary in some of these chromosomes. For example, in Rca3_S2, Rca7_S2, Rca3_R3 and Rca4_R3 chromosomes it far exceeds 2 Mb which is equivalent to >10000 copies of CANR4 monomers. Given that the chromatin of CANR4 satellite is more enriched for CENH3 (Figure 4), we believe that the R3 and S2 chromosomes harbour strong centromeres supporting our hypothesis about the evolution of centromeric drive in *R. canina*.

My comment -> If I understand mixed models properly I thought that pairing type as a fixed effect (and not sub-genome) since pairing type is the explanatory variable whose effect needs to be analyzed. If subgenome is used as a fixed effect, I am not surprised that a correlation is found as there are clear differences between subgenomes. What I am not so sure about if there is a correlation between pairing type and centromere type.

Version 3:

Reviewer comments:

Referee #3

(Remarks to the Author)

I am fully satisfied with this latest version as now it expresses more caution in some of the proposed conclusions. From my perspective, this is a great piece of work and the article is now of the highest standard.

I have a few really minor suggestions though, that I would leave up to the authors and the editor if they want to implement or not:

-L162-163: "subgenomes, which were distinctly separated in both species (Supplementary Fig. 8c–d). This pattern suggests that the univalent subgenomes, S2 and R3, are accumulating more divergence over time." I think it is a bit early to speak about univalent-forming sub-genomes at this point of the manuscript, since this conclusion came later.

-I would also suggest a bit more of extra caution while interpreting the results in the section entitled "Deciphering bivalent-forming subgenomes and meiotic asymmetry in dogroses". I feel that this nice piece of data shows unambiguously which genomes are transmitted from the male and the female side. Which genomes are forming bivalents, can only be unambiguously observed cytologically. I totally agree that based on previous literature and on the new results provided by this work. Proposing that the bivalent forming chromosomes are the one that are transmitted by pollen is a very solid model, but I would maybe state the slight difference between the two things.

-I would add scale bars on figures 2A and 2D.

Response letter to Referees' comments:

Referee #1 (Remarks to the Author):

This study assembles the dog rose genome and reveals an unusual pentaploid genome, as well as striking mixed univalent and bivalent modes of inheritance during meiosis, which associates with distinct centromere structures and sequence composition. This reveals a previously unknown genome architecture. Although the genome architecture is unusual, it has important implications for genome organisation, evolution and centromere function. I found the most interesting aspect is the 'drive or die' model, but further evidence would be required to support this.

Comment R1.1) *Rosa canina* (dog rose) is unusual in having 7 chromosomes that show typical bivalent inheritance through meiosis, in addition to 21 chromosomes that show univalent inheritance and appear to show strong maternal drive. HiFi sequencing and Omni-C data are used to assemble the 2.4 Gb genome – this appears to be performed well and carefully. The authors combine this analysis with comparison of gene orthology, kmers, and LTR divergence to develop a model for the hybridization history of dog rose. While the specifics of the hybrid and polyploid origin of this genome are studied in great detail, I personally found some of these details less interesting and are less likely to be general interest in my opinion, apart from having a pentaploid structure, which is unusual.

Answer to R1.1: We thank the reviewer for the thoughtful feedback on the narrative. In response, we have revised the results to improve readability and moved some of the technical details to the Methods and Supplementary Materials sections. Instead, we have placed greater emphasis on the biological implications of the findings, particularly the unusual pentaploid structure the bimodal centromere architecture and their implication for dogroses unique sexual reproduction. We hope these changes make the article fit a broader audience's interests.

Comment R1.2) The most interesting aspects of the study are that the univalent and bivalent chromosomes have markedly different repeat sequence compositions, and this correlates with very distinct inheritance behaviour through male or female meiosis. In particular, the univalents appear to show obligate maternal drive. This is certainly the most novel aspect of the study, but the model presented is also the most speculative aspect, and I feel further work needs to be performed to prove the model.

Answer to R1.2: We sincerely thank the reviewer for their insightful comments and constructive feedback on our study. We agree that the intriguing aspect of work is to validated the distinct inheritance behaviour of univalent and bivalent chromosomes through male or female meiosis, particularly the obligate maternal drive observed in univalents. We also acknowledge that the model we initially proposed was speculative and required further experimental validation.

In response to the reviewer's concerns, we have conducted additional experiments to strengthen our findings. Specifically, we performed CENH3 ChIP-seq and immunostaining of meiotic chromosomes, which provided clear evidence of a bimodal centromere architecture distinguishing univalents and bivalents. This structural difference aligns with their distinct inheritance patterns and supports the idea that centromere composition and organisation play a critical role in driving the observed maternal bias in univalents. In addition, centromere dimensions could be experimentally proved by CENH3 ChIP-seq. A strong centromeric activity was related to *CANR4* repeats (Fig. 4 revised). We also added a FISH experiment of a male diakinesis of *R. canina* showing *ATHILA* signals on all bivalents (co-localization of *ATHILA* and *CANR4* on two bivalents).

Furthermore, by analysing synthetic hybrids, we were able to trace the genomic contributions of four subgenomes from the female and one subgenome from the male donor (Figure 3e, Suppl. Figure 11c–e). This analysis confirmed the inheritance patterns we initially hypothesised, with univalents showing a strong maternal drive, while bivalents exhibited more balanced inheritance. These results provide robust experimental support for our model and underscore the importance of centromere architecture and repeat composition in shaping meiotic outcomes. We believe these additional experiments and results adequately support our model and its implications for understanding chromosome inheritance and centromere evolution.

Meanwhile the consortium Darwin Tree of Life (DTOL) published two dogrose genomes (*Rosa canina* of subsect. *Caninae* and *Rosa agrestis* of subsect. *Rubigineae*). Including these data in the revised manuscript offered us the great possibility to compare the bioinformatic results among several species. We found strong congruence of results between two individuals of *R. canina* in terms of genome (Suppl. Figure 8a, 9a) and centromere (Suppl. Figure 15) composition of bivalents and univalents. In addition, the proposed genome composition of subsect. *Rubigineae* from pollen data was also supported by data from the *R. agrestis* genome (Figure 3c; Suppl. Figure 11a). Across the three genomes we also observed some variation in centromere size between bivalents and univalents (Suppl. Figures 14–16, Suppl. Dataset 2, 7–9). The high *CANR4* and low *ATHILA* abundance was most strongly pronounced in subgenomes *S2* and *R3* which act in both species as univalents only (Figure 4d).

Comment R1.3) The bivalent chromosome centromeres are shown to be dominated by *ATHILA* retrotransposons, whereas the univalent centromeres are dominated by satellite *CANR4* repeats. Using *KNL1* immunostaining evidence is provided that the univalent centromeres have ‘smaller’ kinetochore foci. It would be beneficial if the ‘size’ of the centromere could be studied in further ways – eg ChIP-seq of *CENH3*, or possibly the *KNL1* protein, could be informative about the amount and arrangement of kinetochore loading on the bivalent versus univalent chromosomes. Is anything known about the relative epigenetic states, specifically DNA methylation, on the univalent versus bivalent centromeres?

Answer to R1.3: We agree that the centromere sizes need a more accurate quantification. We thus performed *CENH3* ChIP-seq as the reviewer suggested and our data well support that the univalents have significantly bigger centromeres than bivalents (Supplementary Figure 14; Supplementary Dataset 7-9). The revised version of Fig. 4d now contains DNA methylation in all three different contexts. As expected, there is a slight increase of methylation towards the centromere region reflecting accumulation of repeats in that region (chromosome-wide analysis in the upper row). However, we found no obvious difference in methylation patterns between bivalents and univalents, which is also supported by relatively uniform immunostaining with the *H3K4me3* (Figure R1) from previous study (Kalfusová *et al.*, 2024). A high-resolution analysis of centromeric region revealed that *CANR4*-based centromeres harbour a conspicuous dip in DNA methylation at the centromere mid-point, which as not observed in *ATHILA*-based centromeres. Such dip in DNA methylation has been also observed in other satellite-based centromeres in other plants (Naish *et al.*, 2021; Hofstatter *et al.*, 2022; Dias *et al.*, 2024). Altogether, these results point to a similar epigenetic regulation of bivalent-forming and univalent chromosomes.

Redacted

Comment R1.4) This difference in centromere sequence, and presumably kinetochore loading, is then correlated with regular versus driving inheritance in female meiosis. Minimally, further study of female meiosis needs to be performed to confirm for example that cell death occurs in 3 of 4 meocytes. This presumably must be associated with some cellular or spindle asymmetry that the CANR4 centromeres can sense somehow? Further data should be provided to compare CANR4 centromere behaviour in male versus female meiosis to prove their model, which would be novel and of wide interest and significance in the centromere field.

Answer to R1.4: We thank the reviewer for their insightful comments and for highlighting the importance of further investigating the behaviour of CANR4-based centromeres during female meiosis. We agree that understanding the mechanisms driving the obligate maternal inheritance of univalents, including potential cell death in 3 of 4 meiocytes and the role of spindle asymmetry, is critical to fully validate our model. However, despite intensive attempts we have not been able to capture the female meiosis in dogroses, which reveals to be rather challenging. We did, however, characterised the centromere and spindle dynamics during male meiosis and through the analysis of controlled hybrid crossings proved the 4:1 to inheritance in female versus male meiosis. Below, we address these points in light of our new experimental results and findings.

- 1) The death of 3 out of 4 haploid products has been already described by Täckholm (1922). We have now added a new suppl. figure (Supplementary Figure 21) of his initial findings of dogrose female meiosis where he provided exact drawings of nearly all stages of female meiosis (Täckholm, 1922). Indeed, in this case he has reported that only the apical cell of the female tetrad generates the egg cell.
- 2) Our immunostaining and CENH3 ChIP-seq data revealed that CANR4-based centromeres are larger and show stronger CENH3 binding compared to ATHILA-based centromeres, which may contribute to their distinct behaviour during meiosis. In male meiosis, CANR4-based univalents experience bipolar orientation and early sister chromatid separation in meiosis I instead of meiosis II leading to lagging chromatids, which are ultimately eliminated. In contrast, bivalents segregate normally. This suggests that CANR4-based centromeres may interact differently with the spindle apparatus compared to ATHILA-based centromeres (Kalfusová *et al.*, 2024). We predict that in female meiosis, CANR4-based univalents experience monopolar orientation due to spindle asymmetry (as shown by Täckholm 1922, Supplementary Figure 21), leading to their obligate maternal inheritance. The hypothesis is fully consistent with the structure of centromeres in the S2 and R3 subgenomes, whereas R4 centromeres are CANR4-poor and ATHILA-rich. Still R4 chromosomes are transmitted in female meiosis. Thus, additional factors are likely playing a role in female meiosis for univalent segregation. Furthermore, we discovered several newly arisen tandem repeats from LTR sequences in two R4 centromeres (Rca1_R4 and Rca4_R4), which show enhanced CENH3 binding compared to neighbouring ATHILAs (Suppl. Figures 17 and 18). These findings highlight the higher affinity of tandem repeats for centromere function in dogroses, further emphasizing the role of centromere composition in shaping meiotic behaviour in these species.
- 3) Offspring of the sequenced individual of *R. canina* S27 clearly originated by sexual reproduction from reduced embryo sacks according to expectations from Canina meiosis: We performed Flow Cytometric analyses of the endosperm/embryo ratio in *R. canina* (Fig. 3d) and compared with the ratio in the sexual diploid *R. rugosa* (Supplementary Fig. 11b). These analyses showed that in *R. canina* 9C/9x endosperm and the 5C/5x embryo originated from a reduced embryo sac mother cell and merged with one pollen nucleus each because the endosperm/embryo ratio matched the expectation of $9C/5C = 1.8$.
- 4) Our new analysis of the genomic composition of two reciprocal synthetic dogrose hybrids between subsect. *Caninae* and *Rubigineae* originally produced by (Wissemann & Hellwig, 1997) provided strong support for the inheritance patterns predicted by our model. In crosses involving *R. canina* as the female parent and *R. rubiginosa* (subsect. *Rubigineae*) as male parent, the resulting hybrids showed a 4:1 subgenome contribution (Figure 3e). This is consistent with the formation of a 4x egg cell (containing one copy of each S1, S2, R3, and R4 subgenome) and a 1x pollen nucleus (containing the R4 subgenome; Figure 3e; Suppl. Figure 11c). Accordingly, we obtained reciprocal crossing results in *R. rubiginosa* (female parent) × *R. corymbifera* (subsect. *Caninae*, male parent, Supplementary Figure 11d–e). Hybridisation, and thus, sexual reproduction is frequent in dogroses and has been observed in many natural populations (e.g. (Ritz *et al.*, 2011; Herklotz & Ritz, 2017).

In addition, previous studies have estimated only a very low frequency of apomictically produced offspring (Nybom *et al.*, 2004; Nybom *et al.*, 2006).

Comment R1.5) An interesting evolutionary question is what is the sequence composition and location of centromeres in *Rosa chinensis* and *Rosa rugosa*? Are their centromeres ATHILA or CANR4 dominated and are they located in syntenic regions?

Answer to R1.5: The reviewer is entirely right in that the comparison of polyploid and diploid species would be informative. To address this point, we carried out genomic and cytogenetic analyses of *Rosa chinensis* ($2n = 2x = 14$, section *Synstylae*, GCA_041222415, released in 2024) and for *R. rugosa* ($2n = 2x = 14$, section *Rosa*, DTOL GCA_958449725.1). At the genomic level, we now show: (i) Similarity ModDotPlot profiles and quantitative analyses of centromeric repeats in *R. chinensis* and *R. rugosa* centromeres (Supplementary Datasets 3 and 4). Interestingly, in contrast to *R. canina*, in both diploids most chromosomes were composed of ATHILA based centromeres, while CANR4 arrays were relatively shorter compared to those in *R. canina*. Putting these results into the phylogenetic frame, it seems that the centromere composition of *R. chinensis* is similar to that of the bivalent-forming S1 subgenome, and the centromere composition of *R. rugosa* is close to that of R4 subgenome. In these two subgenomes, centromeres were mostly composed of Ty3/Gypsy/ATHILA retroelements. In contrast, centromeres in the S2 and R3 subgenomes are almost exclusively composed of tandem repeats from the CANR4 family. The CANR4 satellite did not arise “*de novo*” in *R. canina* since it was detected in all *Rosa* species we analysed so far (Lunerová *et al.*, 2020) while its abundance is relatively low in diploids. These results allowed us to draw a hypothesis on centromere evolution in *Rosa*. We suggest that there might be a balance between CANR4 satellite and ATHILA elements. Such a balance is probably maintained by meiotic recombination, a process which reduces and expands tandem arrays in *R. canina* bivalent-forming subgenomes. However, in non-recombining univalent subgenomes the balance is disrupted by failing removal of satellite DNA by the absence of this process. This model is now further elaborated in the manuscript.

Comment R1.6) In terms of repeat analysis, reading the methods, it seems that Illumina sequencing followed by RepeatExplorer2 analysis was performed? This seems surprising given a long read genome was produced – would it not be better to analyse the assembly directly using the wide range of transposon and tandem repeat mapping tools available? Indeed it seems that the LTR elements are classified into the main families present in plants, but this process is not explained clearly in the methods, or does RepeatExplorer2 provide this information? Further detail on tandem repeat and transposon identification should be provided in the Methods and explained in the main text briefly.

Answer to R1.6: We thank the reviewer for pointing this out. Indeed, we have used both short reads and the long-read assembled chromosomes to perform the repeat annotation of the genome. Both strategies delivered similar results, confirming the high quality of our assemblies. We have now improved the method description of the repeat annotation. For tandem repeat discovery we have used TideCluster and for Transposable elements DANTE and DANTE-LTR, all of which are implemented within the RepeatExplorer2 galaxy server. Furthermore, we have done similar annotations for our newly generated assemblies for *R. canina* and *R. agrestis* from DTOL and have compared them with our initial genome assembly.

Comment R1.7) Figure 1 seems rather simplistic for a main figure, and appears to summarise published knowledge. Perhaps this could be combined with Figure 2?

Answer to R1.7: We agree with the reviewer. We have now added the figure as panel **a** in the revised Figure 1.

Comment R1.8) Minor points – when I first read the title I read ‘odd’ as meaning strange, rather than numerically odd. One alternative might be to use pentaploid instead?

Answer: We thank the reviewer for this suggestion. We completely changed the title to be more explicit with our findings.

Comment R1.9) This sentence is confusing – particularly the usage of ‘non-random’ and ‘highly specific’. ‘As expected, the number of SNPs observed in pollen 234 samples was considerably lower than in the respective somatic tissues (Figure 6b), which supports the 235 hypothesis that bivalent formation is non-random but highly specific.’

Answer to R1.9: During revising the manuscript we deleted the entire part of the figure and its respective description.

Referee #2 (Remarks to the Author):

This manuscript describes sequencing of a polyploid plant with a unique chromosomal architecture that permits the retention of univalent, and thus non-pairing and non-recombining, subgenomes. This makes possible a fascinating exploration of the consequences of carrying both sexual and non-sexual chromosomes simultaneously. The work is competently done, and the data is well presented. As is the nature of purely descriptive work, it lacks tested hypotheses. For instance, the bimodal distribution of centromeric repeats is fascinating but purely correlative. I’d be interested in the authors idea of a test of the hypothesis that the repeats are providing a distinct meiotic drive system. The writing is generally serviceable, although I think it places the points could be made with more clarity. I’ve suggested some example in my detailed comments, some of which are likely due to my misunderstanding of the text, but should help the authors by identifying text that could be rephrased for clarity.

Comment R2.1) Line 68. Just so you don’t lose non-specialist, can you clarify what you mean by tetrasomically inherited means. That pairing and recombination can occur between any one of the homoeologous chromosomes and that all four copies are reliably inherited?

Answer to R2.1: We cited the paper by Bourke, P. M. *et al. Plant J.* **90**, 330-343 (2017) which said: “We found conclusive evidence for parent-specific preferential pairing behaviour on three chromosomes whereas the other chromosomes showed tetrasomic behaviour.” This implies that 4 out of 7 chromosomes exhibit multisomic inheritance and 3 disomic inheritance. In our opinion, this only illustrates that *R. hybrida* is an allotetraploid with a complex genetic history involving numerous introgression events which can, in fact, be expected since it has widely been used by the breeders. We modified the sentence as follows: We rephrased the sentence avoiding the specialised terminology: “that most genomic markers were recombined freely from all four chromosome sets”.

Comment R2.2) Line 72. Again, it would be helpful to clarify exactly what you mean here for the non-specialist. As in, “because...x, y”. Here, you are saying that preferential pairing can lead to apomixis with respect to one set of chromosomes and not another, but you don’t make clear why this is. You don’t want your reader to get lost so early in your explanation.

Answer R2.2: We rephrased the sentence: “Within the *Synstylae* clade, allopolyploid dogroses [*Rosa* sect. *Caninae* (DC.) Ser.] exemplify a fascinating reproductive strategy, where the selective pairing of specific recombination partners results in a mixed mode of chromosome inheritance – combining both biparental and uniparental transmission – in the same nucleus.”

Comment R2.3) Line 78. Ah, so preferential pairing only matters when you have an odd number of chromosomes. If this is the case, perhaps you could set up this section saying that preferential pairing isn't a big deal if you have an even number, but it is necessary with an odd number?

Answer to R2.3: We thank the reviewer for pointing this. However, as also raised by Reviewer 3 (see below), we have removed the use of preferential pairing, as we do not really know how the different subgenomes are being paired in early meiotic prophase. However, indeed a preferential pairing is likely connected to their allopolyploid nature, as some subgenomes are only found in single copy and thus lack a partner homolog to pair too. This is particularly interesting as asymmetrical meiosis is observed not only in odd chromosome number species but also in tetraploids and hexaploids (e.g. Täckholm 1922). This suggests that despite an apparent even number of chromosome sets, subgenomes might still be present in single copy.

Comment R2.4) Line 88. Among eukaryotes that have been examined.

Answer to R2.4: Changed into “eukaryotes examined so far”

Comment R2.5) Line 90. I think you could be more explicit about what it is you are investigating. Certainly, preferential pairing is a must, but the biased apportionment of the univalent is also very interesting. The way this is worded suggests to me that this question wasn't really addressed.

Answer to R2.4: We changed this sentence to be more explicit about asymmetric univalent inheritance. Although we do not have cytological data on female meiosis, we provide evidence from reciprocal synthetic hybrids between subsect. *Caninae* and *Rubigineae* that *Caninae* egg cells provide one bivalent-forming set (S1) and all univalent-forming genomes (S2, R3, R4; Fig. 3e, Suppl. Figure 11c). Accordingly, *Rubigineae* egg cells provide one bivalent-forming set (R4) and all univalent-forming genomes (S1, S2, R3; Supplementary Figure 11d–e).

Comment R2.6) Line 127. Any evidence for recombination between the more divergent chromosomes early following polyploidization?

Answer to R2.6: As far as we know there is no evidence available. Also based on our data, we could not detect any sign of recombination between univalents and the bivalent-forming subgenomes. This greatly supports that hybridisation events most likely happened in an asymmetric genome contribution, with two copies of a subgenome hybridised with an additional copy of a different subgenome, potentially forming an allotriploid.

Comment R2.7) Line 144. I'm confused. I thought they were nearly identical, although I guess they could be nearly identical and scrambled relative to each other?

Answer to R2.7: We thank the reviewer for identifying this confusing naming of subgenomes. To improve the clarity of subgenome relationships, and as also suggested by referee 3, we changed the naming of subgenomes: There are only four subgenomes S1, S2, R3 and R4 with the R subgenomes originating from the *Rosa* clade and the S subgenomes originating from the *Synstylae* clade. The S1

genome is an “internal diploid genome” presenting two sets of homozygous homologous chromosome sets, which are now called haplotypes S1_h1 and S1_h2. Similarly, the R4 subgenome is an “internal diploid genome” in *Rosa agrestis*, which are now called haplotypes R4_h1 and R4_h2. We have changed this throughout text, figures and data hoping it is clearer now.

Comment R2.8) Figure 2e. Perhaps a bit more explanation of the y axis here? I’m sure it is in the M&M, but a brief description would be helpful. Syntenic regions can be defined in many different ways, which makes it difficult to know what “proportion of syntenic regions” means. For instance, I would say that an inversion only changes synteny at the boundaries of the inversion, but you could also say the whole region is now out of synteny. Also, what does the shading in the boxes in the box plots indicate?

Answer to R2.8: We thank the reviewer for this suggestion. However, since this plot was not really adding any new information to the analysis in this figure and to keep the clarity of the manuscript narrative, we decided to delete this plot.

Comment R2.9) Line 153. Why is this remarkable?

Answer to R2.9: The reviewer is right; this was redundant information.

Comment R2.10) Line 167. Again, why is this remarkable? Aren’t these gene poor regions that are much more likely to accumulate differences?

Answer to R2.10: We thank the reviewer for pointing this out. In order to keep the clarity of the text we have reformulated most of the manuscript and reformulated this entire section. We hope the reviewers appreciated our efforts in improving the overall writing of the manuscript. The reviewer is right about that the accumulation of mutations at proximal regions is expected. We should have chosen the words more cautiously to minimize the misunderstanding. We have rephrased the sentence and emphasize more on the differences of *k*-mer distribution and density between different subgenomes.

Comment R2.11) Line 173. What does “similarly high levels of BUSCO genes” mean? BUSCO was mentioned earlier, but it isn’t clear what you mean by it here.

Answer to R2.11: We deleted the term and replaced it by the more general term gene content. BUSCO is the abbreviation for “Benchmarking Universal Single-Copy Orthologues” and a software tool to provide quantitative measures assessing genome assemblies. Moreover, we have completely rewritten this section for sake of clarity.

Comment R2.12) Line 174. How do you know it wasn’t an ancient autopolyploidy, such as hypothesized for soybean?

Answer to R2.12: We rephrased the entire section. The here presented phylogenies (Fig. 1 and 3) clearly indicate an allopolyploid origin for the R and S genomes, which was also assumed in earlier publications, see Introduction, l. 99ff.

However, the reviewer is right: Ancient partial autopolyploidy in the evolutionary history of *R. canina* cannot be totally excluded. We can, for example, imagine that R genomes may have diversified into R3 and R4 in an ancient hybridisation, since they are sister to each other in the phylogeny (Fig. 3c). The same situation for S1 and S2 genomes could be possible because only the S2 genome is sister to a diploid species of *Synstylae* (*R. arvensis*). So, a hypothetical autopolyploid (e.g. S1S1S1 genotype) could give rise to S1S1S2 subgenomes in modern *R. canina*. If so, the question arises of why the two S1

subgenomes remain homologous while the third one diverged and evolved into the S2 subgenome. For such reasons we consider the allopolyploidy involving repeated hybridisations events as a more likely evolutionary scenario and proposed a rather coarse model in terms of ordering the events in Figure 6b.

Comment R2.13) Line 181. "...between the estimated S1/2-s3-R5 from R4 subgenome-specific LTR-RTs insertion times, ranging from..." Awkward wording. between the estimated what? divergence times? How does a boundary range from x to y?

Answer R2.13: We agree that the wording was a bit confusing. However, in light of our new restructuring and additional analysis in the manuscript, we have rewritten this part. We hope the text has now improved for clarity.

Comment R2.14) Line 183. Last, not "at last".

Answer R2.14: This part has been rewritten.

Comment R2.15) Line 184. Can you make this assertion without reference to the known progenitor of each subgenome? That is to say, can you make any assertion about the timing of the allopolyploidy just by looking at divergence of the subgenomes?

Answer R2.15: We thank the reviewer for pointing out this issue in the first version of our manuscript. In light of our new results, which we now include the analysis of other two dogroses assembly from DTOL, we have completely rewritten this part. Our new results suggest that the hybridisation history of dogroses involved stepwise and independent events in *R. canina* and *R. agrestis*. The distinct insertion times of subgenome-specific LTR-RTs suggest that the combination of the S1, S2, and R3 subgenomes occurred at different time points in each species (~0.7 MYA for S1-S2-R3 in *R. canina* vs. ~0.7 MYA for S1-S2, ~0.9 MYA for R3 in *R. agrestis*), while the R4 subgenome (~1.2 MYA) appears to have been incorporated later in both species. The differentiation of the R3 subgenome between the two species further supports the idea of independent hybridisation events, highlighting the complex and polyphyletic origins of dogroses.

Comment R2.16) Line 186. I think you mean the number of substitutions, not the rate, which generally refers to time. Also, rather than a differential rate, perhaps you could simply talk about degree of divergence. The writing here is unnecessarily confusing. I'm also not clear about the point being made here.

Answer R2.16: We agree with the reviewer and we have now rewritten this part.

Comment R2.17) Line 193. I'm not sure reference 39 actually shows this. Many plant genomes have very large proportion of retro-elements. Reference 39 shows phylogenies of these, but I don't see a discussion of relative amounts of different types in different genomes.

Answer R2.17: We agree with the reviewer that this reference was not properly placed there. We decided to rephrase this sentence and remove this citation.

Comment R2.18) Line 201. Again, not remarkable. There are many examples of both types of repeats being enriched in centromeres.

Answer R2.18: We have now rephrased this sentence.

Comment R2.19) Line 205. Awkward wording. I think you mean, “Although tandem repeats are common in the S1/S2 genomes, they are not enriched in (peri)centromeres.”

Answer R2.19: We thank the reviewer for pointing this out. We have now rephrased this section in the new version of the manuscript.

Comment R2.20) Line 209. How are you defining centromeres here?

Answer R2.20: We agree that we lacked robust evidence for the centromere position and associated sequences in the first version of the manuscript. We have now carried out a comprehensive analysis of these elements that shape *Rosa canina* centromeres by CENH3 ChIP-seq. As expected, we found that the satellite CANR4 repeats which dominate centromeres in univalent S2 and R3 subgenomes show high levels of CENH3 association. In R4 subgenome, two newly identified repeats which arose from the LTR sequence of centrophilic ATHILA seem to be highly associated with CENH3 and outcompetes the ATHILA element. Furthermore, centromeres that lack any tandem repeat show moderate association with the centrophilic ATHILA elements.

Comment R2.21) Line 226. The introduction would have benefited with a short discussion of this kind of meiotic drive.

Answer R2.21: We have added a sentence about a possible link to centromere and meiotic drive to the introduction.

Comment R2.22) Line 263. “Thus, bivalent-forming chromosomes in subsect.” is not a sentence and is unclear. Perhaps this is a term of art, but I don’t understand what subsect means here.

Answer R2.22: This is a misunderstanding. “Subsect.” is the defined abbreviation of the taxonomic level subsection. Dogroses belong to *Rosa* section *Caninae* and are composed of several subsections (here subsect. *Caninae* and *Rubigineae*) which then contain several species. Since section and subsection share the name *Caninae* (because the type species *R. canina* is therein) we have to make the differentiation to make clear whether we refer on all dogroses (sect. *Caninae*) or only a subset (subsect. *Caninae*, here the ones with S1 bivalents).

Comment R2.23) Line 273. Or just differences in divergence times?

Answer R2.23: Thanks for the comment. A detailed discussion on branch lengths, especially when combining SCO data and whole genome sequences is possibly not sound enough. We deleted this sentence and rephrased the entire section. In addition, we performed an analysis on Ka/Ks ratio which did not point on subgenome-specific selective regimes (Supplementary Figure 7c), see also R3.37, R3.46.

Comment R2.24) Line 299. Of course this meiotic drive is also likely to have negative consequences given that lack of recombination, as mentioned later. It is surprising that there aren’t higher levels of fractionation in the univalent, given that the drive mechanism retains the centromere, not necessarily the genes on the chromosomes. Is expression data available? I would expect homoeologous genes on the univalent to express at lower levels, on average, and they should also have higher Ka/Ks values.

Answer to R2.24: This is a very valuable remark, which was also raised reviewer 3. In our group we produced a SCO target on exon sequences with flanking intron regions across the entire genus *Rosa*. While concatenating the sequences in the diploids was rather straight forward (see Fig. 3), phasing and concatenating haplotypes in polyploids was in most cases not feasible due to a nearly complete lack of sequence divergence in the coding regions within polyploids. The general low sequence divergence is expected due to the rather young age estimate of dogroses (based on LTR insertion times: 0.7-1.2 MYA, see Fig. 2d-f; based on Ks-values 0.3-0.6 MYA in Supplementary Figure 7). We did have performed RNAseq analysis of leaf tissue for the sake of functional annotation of the genome. We observed slightly higher gene numbers and transcripts on S1 and R4 subgenome than expected if we assume that transcripts number is proportional to gene number (Suppl. Table 6; Suppl. Figure 23 a-b), which suggests that bivalents have more active transcription in general. In addition, four subgenomes present distinct patterns on the top 100 genes with high expression (Supplementary Figure 23 c), which indicates the organism's normal functionality might require the orchestrated gene activities of all subgenomes. However, we agree that a more refined experiment using different replicates and tissues should be performed in order to identify slightly differentiation in expression activity, if incipient subgenome dominance is occurring. Therefore, we decided not to discuss these experiments in the main text and we only bring these results in the Online Methods section.

The Ka/Ks ratios have now been determined for a set of 7076 single-copy orthologues in each (S1, S2, R3 and R4) *R. canina* subgenome, *R. chinensis* and *R. rugosa* (Supplementary Figure 7). The low ω values indicate that most mutations have a symmetrical character and are not influencing the amino acid composition of encoded proteins. Hence, the genes in bivalent and univalent genes evolve under the purifying selection. There might be several interpretations: (i) Most selected genes are slowly evolving. (ii) Alternatively, hybridization events giving rise to the *R. canina* genome could be of recent origin, and hence there was insufficient time to accumulate non-synonymous mutations.

Referee #3 (Remarks to the Author):

Key results:

In this manuscript, Herklotz et al, made crucial step into solving the mystery of how pentaploid dogroses (*Rosa canina*) maintain their odd ploidy level across generations of sexual reproduction and going through meiosis. In this work the authors assembled a haplotype-phased genome of *R. canina* and uncovered the presence of four distinct constitutive subgenomes. This high-quality genome allowed a detail profile of the repeatome of each subgenome and a description of their satellite repeats and transposable elements, plus an analysis of the centromeres. In addition, this data unlocked an analysis of the origin and phylogenetic relationships between the constitutive subgenomes and related species. Importantly Sequencing of pollen nuclei enabled a robust identification of the subgenome(s) that form bivalents during male meiosis, conformed by two sets of highly syntenic and homozygous chromosome sets. Finally, the authors also provided a description of the localization patterns of KNL1 in mitotic chromosomes.

Validity:

Comment R3.1) As implied in the abstract, this work address very well one of the unknowns of dogrose reproduction: its chromosome properties. Remarkably, the heroic assembly of an extremely complex genome such as that of dogrose deserves credit, along with the robust and highly convincing

identification of the bivalent- and univalent-forming genomes. Unfortunately, it was not possible—in my opinion—to provide valid and decisive insights into the unique chromosome segregation of this system. For further explanation see the Conclusions section below) Nevertheless, this goes way beyond an incremental progress and I strongly believe that the genome biology deciphered here, along with the genome references and other resources makes this work a qualitatively important milestone that will be essential to decipher this exciting mystery in future.

Originality and significance:

Comment R3.2) Although previous work already gave an idea of the nature of bivalent- and univalent-forming genomes (Kovarik et al., 2008; Lim et al., 2005) and the phylogenetic relationships with other species (Herklotz et al., 2018), the contribution to these important questions of this work is on completely different scale.

Answer to R3.1, 2: We thank the reviewer for recognising the efforts made in our manuscript, as we now unravel the dogroses genomes in much more scientific depth. As stated above we tried to add more data and evidence from new cytogenetic and bioinformatics analyses to the revised version of the manuscript.

Data and methodology:

Comment R3.3) Overall, I am satisfied with the methodology of this work, especially regarding the aspects of the genome biology of *R. canina*. However, I believe that to address the chromosome biology mysteries, the mechanisms underlying non-symmetrical meiosis, a deeper cytological work might have been beneficial. For instance, immunostaining of the synaptonemal complex protein ZYP1 could have provided useful information into the level of suppression of recombination and pairing (the strict definition of pairing, see below in the Clarity section) during early meiosis for univalent-forming chromosomes. Moreover, a follow-up of the fate of univalent-forming chromosomes from anaphase I until pollen mitosis (when the authors nicely demonstrated that the univalent chromosomes are already excluded). This could have been achieved, marking some centromeres with CARN4 FISH or all of them by centromere-specific histone variant CENH3. Also, using these cytological markers in metaphase I could have been a nice validation that the regions identified as centromeres do show centromeric activity (e.g. stretching during metaphase I). Finally, I wonder if sequencing of endosperm is possible in this system and which constitutive genomes would be present in this tissue. This could provide some insights into the meiotic drive mechanism that in female meiosis makes one copy of each chromosome from each subgenome present in the egg cell.

Answer to R3.3: We thank the reviewer for her/his insightful suggestion to analyse the synaptonemal complex by the detection of ZYP1 protein. We carried out an immunostaining experiment of pollen mother cell pachytene chromosomes using antibodies against the ZYP1 and ASY1 proteins, which allows the detection of synapsed and non-synapsed chromosomes, respectively. Indeed, we observed a very distinct pattern of ZYP1 in only a subset of chromosomes, possibly representing bivalents, while ASY1 was consistently observed in other chromosomes till late pachytene, likely to represent non-synapsed univalents (Figure R2). As this is still a preliminary experiment, we think that it will be necessary to conduct further careful analysis, which is an ongoing project in the lab. Thus, we kindly ask the reviewer that we do not show these preliminary results in the current manuscript.

Redacted

Centromere validation: The comment on physical identification of centromeres using CENH3 has been made by the other reviewers too. We fully agree and are happy to present these results in the revised version (New Fig. 4 and supplements). In addition, we provide new cytogenetic data on the occurrence of CANR4 and ATHILA in the male diakinesis of *R. canina* (Suppl. Figure 19) and CENH3 immunostaining together with Alpha tubulin across most stages of the male meiosis (Figure 5 and Suppl Videos).

Sexual Reproduction Verification: While sequencing the endosperm is a promising approach, the limited number of SNPs between the two S1 haplotypes makes it difficult to distinguish between sexual and apomictic reproduction. To address this, we conducted an experiment to measure the ploidy ratio of endosperm to embryo (Fig. 3b, Suppl. Figure 11b) to confirm sexual reproduction in the female line. In apomictic reproduction, the endosperm would arise from duplicated somatic cells, resulting in a 10x endosperm. If fertilized by the second generative nucleus, this would yield an 11x endosperm. However, our results showed a clear 9x endosperm / 5x embryo ratio, consistent with sexual reproduction. This ratio suggests the endosperm originates from a duplicated 4x female gamete combined with a haploid gamete from the generative nucleus.

To further validate sexual reproduction in dogroses, we performed SNP detection analysis on reciprocal synthetic hybrids between subsect. *Caninae* and *Rubigineae*. The genome composition of these hybrids aligned with the expectations of Canina-meiosis (Figure 3e, Suppl. Figure 11). Together, these analyses confirm the 1x male gamete and 4x female gamete segregation mode in dogroses, reinforcing the sexual reproduction mechanism.

Appropriate use of statistics and uncertainties:

Comment R3.4) Overall, statistics is properly used in this manuscript. I would maybe improve it by including a method section on statistics even if there are not a lot of test. I think this section is required to state things like that the homogeneous variance across groups were verified before proceeding to test like ANOVA/Tukey require (as in L146). Apart from this I have a couple of minor comments that I include among suggested improvements.

Answer R3.4: We have skipped the ANOVA and boxplots from the figure because this was redundant with the panel with synteny analyses. We have done additional statistical analysis for CANR4 abundance, which are further detailed below in R3.7.

Conclusions: robustness, validity, reliability:

Comment R3.5) I missed further justification of how regions were assigned as centromeres. In other studies CENH3-ChIP was used (Ahmed et al., 2023; Hofstatter et al., 2022; Naish et al., 2021). I believe this is not a trivial question because, centromere position sometimes is not obvious simply from genomic signatures are not always an ultimate proof of centromere location (Liu et al., 2023). Also, based on the text and Figure 4 and supplementary dataset 2, it is not clear whether the localization of CANR4 is centromeric, or peri-centromeric (Lunerová et al., 2020 described it as “strictly peri-centromeric”).

Answer to R3.5: The reviewer is entirely right in that the exact position of a centromere cannot be determined solely based on repeat accumulation signatures. We therefore carried out immunostaining of chromosomes with antibodies to KNL1 and CENH3. We have now also performed CENH3 and alpha tubulin immunostaining in male meiotic chromosomes. However, the pericentromeric and centromeric positions of repeats cannot be discriminated from cytogenetic experiments due to low resolution. Also, immunostaining methods do not allow to determine which of the repeats are binding centromere-specific KNL1 and CENH3 proteins. In order to identify the sequences underlying functional centromeres in *R. canina* we carried out a chromatin immunoprecipitation (ChIP) experiment using our newly generated antibody for the *R. canina* CENH3. We show that both Ty3/Gypsy ATHILA and CANR4 satellite bind CENH3 and both repeats can be considered as “*bona fide*” centromeric. However, one of the most interesting aspects of CENH3-ChIP-seq experiment was an extraordinarily high association of CANR4 repeats with CENH3 chromatin in comparison with ATHILA-based centromeres (Figure 4). Additionally, we now identified several novel centromeric satellites specific for the R4 subgenome (Supplement Figure 17 and 18), which have independently originated from LTR sequences from full-length centrophilic ATHILA elements. Remarkably, these LTR-derived tandem repeats outcompete the original ATHILA sequences for CENH3 binding.

Comment R3.6) One of the conclusions stated in the abstract (L45-47), is that “Distinct structural features of centromeres suggest a role in the segregation bias of univalents during asymmetric female meiosis...”. I have some concerns regarding the validity and relevance of those distinct features. First, it is explained that CARN4 repeats show preference for univalent-forming chromosomes (implying a potential role in their mysterious segregation). However, I do not understand why, as I feel that preference is not that strong or relevant. As an example. It is mentioned in L209-211 that “Large centromeric arrays of CANR4 were found in 16 out of 21 univalent chromosomes from the s3, R5 and to a lesser extent R4 subgenomes, while six (in syntenic group 2, 4, 5) out of 14 centromeres in the S1/S2 bivalent-forming chromosomes”. This means that in univalent-forming chromosomes ~5.3 chromosomes per genome show large CARN4 centromeric arrays while in bivalent-forming ones ~3 chromosomes per genome carry the centromeric arrays. Figure 4, showing the example of the homoeologous group 6, gives the impression that there is a clear preference of CARN4 arrays for univalent-forming chromosomes. However, this does not hold true for any other homoeologous groups (see Supplementary dataset 2). Indeed, if we carefully observe each genome separately, those arrays are only in 2 out of 7 chromosomes in the univalent-forming genome R4. Moreover, the fact that different chromosomes show distinct centromere repeat organizations has already been observed in other models like soybean (Liu et al., 2023) and maize (Gent et al., 2017) without that implying functional different

during segregation. Therefore, I have problems to see a trend for CARN4 expansion based on whether chromosomes form bivalents or univalents, and the hypothesis that these arrays drive the differential behavior of the genomes during meiosis gets less likely.

Answer to R3.7: The reviewer is entirely right in that the R4 genome does not show expansion of centromeric satellite as opposed to R3 and S2 genomes. We now quantified CANR4 repeats in each *R. canina* chromosome including the two new genomes available from DToL. We used bivariate Bayesian Generalized Linear Mixed Model (GLMM) using Markov chain Monte Carlo techniques to statistically analyse the proportion of ATHILA and CANR4 across subgenomes in *R. canina* (S27 and DToL) and *R. agrestis* (Supplementary Figure 14). The model included subgenome as a fixed effect, while pairing type (Univalent/Bivalent) among others were set as random effects. This experiment clearly showed that the bivalent-forming subgenomes (S1 and R4) harbour ATHILA-type or mixed ATHILA/CANR4 satellite-type centromeres irrespective of genetic background. The proportion of CANR4 satellite is however low. In contrast, the S2 and R3 subgenomes, which always form univalents, show high content of CANR4 satellite. The genome space of the satellite is extraordinary in some of these chromosomes. For example, in Rca3_S2, Rca7_S2, Rca3_R3 and Rca4_R3 chromosomes it far exceeds 2 Mb which is equivalent to >10000 copies of CANR4 monomers. Given that the chromatin of CANR4 satellite is more enriched for CENH3 (Figure 4), we believe that the R3 and S2 chromosomes harbour strong centromeres supporting our hypothesis about the evolution of centromeric drive in *R. canina*.

Comment R3.7) Similarly, I wonder if the differences in KNL1 signal size could be simply due to differences in centromere size. For instance, based on Supplementary Dataset 2, the size of most centromeres is about 2Mb, but it can range from 4 (Chromosome 3 of genome R5) to 0.5 (Chromosome 5 of genome S2). Actually, if we assume that CARN4 is at centromeres (please see Clarity section below), it might seem that the size of the FISH signal does vary substantially among chromosomes. In any case, I believe that some quantification of the KNL1 signal intensity or sharing additional images in the supplementary might reinforce the observation.

Answer to R3.7: To clarify the size and position of centromeres in chromosomes, in addition to KNL1, we have now developed a *R. canina* specific CENH3 antibody and performed immunostaining on *R. canina* mitotic and male meiotic cells. Furthermore, the functional annotation of centromeres across the bivalent-forming and univalent chromosomes has been determined by CENH3-ChIP-seq (Figure 4 and supplements), highlighting the high affinity of CANR4-based centromeres in comparison to ATHILA-based ones (Figure 4d).

References:

I found the use of references appropriate in this manuscript.

Clarity and context:

Overall, the manuscript was clear, but I have a few comments on this regard.

Comment R3.8) First of all, although this is probably a matter of taste, I found the first and second sections of the results (specially the first one) a bit too technical for the main text of an article in general journal like this. I would rather prefer to use the main text to focus on the genome biology and leave some of the technical metrics for the supplementary (maybe using tables where all the assembly metrics are gathered, or supplementary notes if descriptions are required). However, I fully understand that assembling a so complex genome like this calls for sharing the details (two good examples of this are

Ahmed et al., 2023 and Hofstatter et al., 2022). If the authors prefer to keep the technical tone of the first part of the results I would suggest to include a few more sentences explaining the implications of the metrics (I have included a few suggestions in this regard in the suggested improvements below.

Answer to R3.8: We thank the reviewer for the thoughtful feedback on the narrative and good-example recommendations. We agree that the intensive technical terms and numbers might lose our potential readers. In response, we have revised the assembly section and moved most technical details to the Methods and Supplementary Materials sections, but stress the high quality and importance of the assembly. Detailed quality assessments have been organized as Supplementary Tables and Figures.

Comment R3.9) In addition, I have some issues with the use of the word pairing. My understanding is that there is a consensus in the meiosis community nowadays that pairing is the alignment between homologous regions of homologous chromosomes during early stages of meiosis that precedes synapsis and crossover formation (Zickler & Kleckner, 2015, 2023). According to this definition, we cannot speak about metaphase I bivalents using the word “pairing”. I know that, traditionally pairing was used for metaphase I bivalents, especially among plant cytogeneticist. However, I would say that some statements with this use of the term can be imprecise. For instance, it would be imprecise to say that homoeologous pairing is prevented in allopolyploids like wheat and oilseed rape. Yes, metaphase I bivalents are only formed between homologs (and prevented between homoeologs) but pairing (sensu stricto) does occur between homoeologs during early meiosis (Grandont et al., 2014; Martinez et al., 2001). Similarly, the results of this manuscript support that bivalent formation occurs between homologs in dogrose metaphase I, but it is totally possible that pairing (sensu stricto) is also happening at early stages of meiosis between homoeologs (as in wheat and oilseed rape). There is a clear explanation of this distinction in Lloyd & Bomblies, 2016 (Box 1). therefore, I would suggest to differentiate between the two terms: pairing and bivalent formation at metaphase I. Or at the very least, clarify the definition of pairing that the authors will be using.

Answer to R3.9: The reviewer is completely right. Our genomic and cytological data allow only conclusions about bivalent-formation not about chromosome pairing in the strict sense. Due to the lack of detailed immunostaining work with proteins involved in synapsis the chromosomal behaviour during late prophase I is still unknown. Thus, we replaced phrases like “preferential pairing” by “preferential recombination”, “choice of recombination partners” etc. according to Lloyd and Bomblies (2016) and Bomblies (2023) throughout the manuscript. We kept the term pairing at the beginning of introduction when describing the general outcomes of meiosis.

Furthermore, the issue of pairing between homoeologous chromosomes raised by the reviewer is interesting. In this context, we have limited microscopic data showing early stages of *R. canina* meiosis. However, our ZYP1/ASY1 staining showed above in R3.3 (Figure R2) shows that both thick and thin threads are observed in pachytene stage, suggesting that indeed pairing might be occurring only between bivalent-forming chromosomes.

Comment R3.10) Finally, sometimes I get confused about subgenomes S1 and S2. In the abstract it is clearly stated that this work uncovered four subgenomes. However, at the beginning of the results it is explained that the sequencing data was assembled into 35 chromosomes. This means that the chromosomes from the S1 and S2 genomes were assembled separately. However later they are always mentioned as a one S1/S2 and explained as highly homozygous. Could the Ks analysis distinguish between the two s1 and s2? I would not expect so, and indeed, from the text I assume that the k-mer approach did not distinguish between S1/S2... In such a homozygous context, could the two genomes have been assembled without haplotype phasing (that is explained later)? Another contradiction is in

Supplementary Dataset 2, where the chromosomes S1 and S2 are shown separately (although it becomes obvious that not many differences exist between them). Also, in L 127 “chromosome-based phylogenies” are mentioned as a method to allocate each chromosome to their respective subgenome, but I could not find further details in the methods. In addition, the genomes S1 and S2 look pretty empty in the heatmap of the figure S1a. More importantly, since the subgenomes S1 and S2 form bivalents and segregate, I wonder if they can be called subgenomes. I guess that it could be the case that some plants do not have any S2 chromosome at all (i.e. they are S1/S1/S3/R4/R5), right? If that is the case, wouldn't it make more sense to directly say that S1 and S2 are simply haplotypes of the same S1/S2 subgenome (subgenome in singular)?

Answer to R3.10: Thanks for this very helpful comment. Indeed, our assignment and naming were confusing. Following your suggestion we changed naming of chromosome sets: There are four subgenomes S1, S2, R3 and R4 with the R subgenomes originating from the *Rosa* clade and the S subgenomes originating from the *Synstylae* clade. The S1 genome is a “internal diploid genome” with two nearly identical homozygous chromosome sets, which are now called haplotypes S1_h1 and S1_h2. We changed this throughout text, figures and data and hope it is clearer now. Furthermore, for the genome assembly of *R. agrestis*, the R4 subgenome is the “internal diploid genome” in this species with two nearly identical homozygous chromosome sets, which are now called haplotypes R4_h1 and R4_h2.

Suggested improvements: Main text

Comment R3.11) -L34 “...achieved by forming seven bisexually inherited bivalents...” I would say that this is a bit misleading; bivalents cannot be inherited. Since a bivalent is the structure formed by two chromosomes bounded by chiasmata, this sentence can be misinterpreted as both chromosomes segregating together toward the same pole. I would rather say “seven bisexually inherited bivalent-forming chromosomes”.

Answer to R3.11: changed according to suggestion

Comment R3.12) -L70 are-> there is??

Answer to R3.12: changed to represent

Comment R3.13) -L51 “...X in the figure.”

Answer to R3.13: This analysis is skipped from the revised version (see ref. 2/9)

Comment R3.14) -L77-80. The text says: “bivalent-forming chromosomes are passed on to the pollen grains and egg cells, whereas unpaired chromosomes (univalents) are passed on to the egg cells but are excluded from viable pollen nuclei.” What is that exclusion mechanism? I wonder if it a sex-specific genome elimination of particular chromosomes is totally ruled out. I bring this up because, the univalents seem to segregate randomly in anaphase I (Lim et al., 2005). The results of this work, seem to virtually do rule it out for the male side (unless it happens in the short lapse between meiosis and pollen formation, but I wonder if previous literature consider that possibility).

Answer to R3.14: We thank the reviewer for their insightful comment. Using rDNA probes, Kalfusova et al. (2024) demonstrated that univalents persist throughout male meiosis until the late tetrad stage. Our new immunostaining experiments in male meiosis, using CENH3 and alpha-Tubulin, reveal that univalents exhibit bipolar orientation during metaphase I but lag during anaphase I and occasionally

form aberrant spindles in meiosis II. The final elimination of univalents occurs during meiosis II, where they fail to segregate properly due to their monopolar orientation and are lost between telophase II and tetrad formation. This is evidenced by the presence of several micronuclei, as previously observed in Kalfusová *et al.* (2024) (Figure R3). These micronuclei likely account for the high percentage of unviable pollen observed. To further investigate pollen viability, we performed Alexander staining on our sequenced accession and found that only 20% of pollen grains are viable (Suppl. Figure 20). This low viability is consistent with the observed lagging and mis-segregation of univalents, which likely contribute to the formation of micronuclei and the subsequent loss of univalent-containing pollen grains. Our SCO target dataset supports the conclusion that viable pollen grains are predominantly composed of 7 bivalent-forming chromosomes, as it confirms the presence of S1-only in *Caninae* dogroses and R4-only in *Rubiginae* dogroses. These results collectively support a mechanistic elimination of univalents from viable pollen grains.

However, we cannot rule out the possibility that genetic or epigenetic incompatibilities – e.g., similar to sperm-killing meiotic drivers, as for instance found in fission yeast (Nuckolls *et al.*, 2017) – may also contribute to the death of pollen grains containing univalents. This hypothesis is quite exciting as it would imply that bivalents are driving through male and univalents through female meiosis. Further studies could explore specific genetic or epigenetic markers to test this hypothesis.

Redacted

Comment R3.15) -L86. Could it be that the words “and there” should not be there?

Answer to R3.15: deleted (l. 91)

-Figure 1. I just had in mind a slightly different interpretation of the bivalents and univalents. Please check image attached. I know that the authors have a lot of experience in quantifying DAPI signal to figure bivalents and univalents out. This is just a suggestion.

Answer to R3.15: Thanks for the comment, we agree that the diakinesis shown here is not ideal. We exchanged the picture for another one easier for interpretation, now as panel a) in the revised Fig. 1.

Comment R3.16) -L107. It is not clear if only 35 scaffolds were assembled. Maybe a table summarizing all the assembly metrics in the supplementary would help.

Answer to R3.16: Thank you for pointing this out. We have now expanded the Supplementary Table 1 and included the statistics of contigs, scaffolds, and unscaffolded contigs. More details have also been added in Methods to improve reproducibility.

Comment R3.17) -L111-112. I would suggest to clarify the meaning of some of the results presented, same as the BUSCO score is shown as a completeness indication, the same explanation should be given for Mercator4 annotation. I think this is important in a generalist journal like this; many non-genomics experts will read this manuscript.

Answer to R3.17: We agree that some metrics need more interpretation to fit a broader readership. Aligning with reviewer’s R3.8), we have removed most assembly metrics in the main text and given a detailed description of assembly assessment and their indications in the legend of Supplementary Figure 1 and Suppl Table 1.

Comment R3.18) -L114. Is it possible to express that these parameters indicate completeness of the “single copy” genome? From those we don’t know how completed are duplicated regions...

Answer to R3.18: Thank you for raising this point and allowing us to clarify. To address the ambiguity, BUSCO and Mercator4 were run on the protein sequences of all five haplotypes respectively. BUSCO were run on the merged genome as well. The reported BUSCO gene completeness of 99.2% refers specifically to the merged genome. However, we also observed high completeness for each individual haplotype. Specifically, each “single copy” genome achieved a BUSCO completeness score of no less than 97.1% and a Mercator4 completeness score of no less than 94.5%. These results indicate strong overall genome quality, even at the level of individual haplotypes. Since the assembly section has been rephrased, these details have been clarified in the legend of Supplementary Figure 1.

Comment R3.19) -L122-123. I am a bit confused with these lines. I feel that the words linkage and synteny are used as synonyms here. However, linkage can also refer to genetic linkage (genetic maps) which are also used for other assembly strategies such as trio binning.

Answer to R3.19: We agree with the reviewer. To avoid confusion, we use “synteny” when describing chromosome comparisons. In addition, term “linkage group” was replaced with “synteny group”.

Comment R3.20) -L122-123. Again, I would include a sentence summarizing the conclusion from that data, for instance “...this result uncovered the identity of the 5 homoeologs for each synteny group”.

Answer to R3.20: We thank the reviewer for noticing the lack of such an important sentence. We have now added a sentence in the of this paragraph: “These findings resolve the long-standing question of the identity of the five homoeologous chromosomes within each syntenic group.”

Comment R3.21) -145-146. I would include a reference to figure S2e

Answer to R3.21: This figure has now been moved to MM and a reference has been added.

Comment R3.22) -Figure 2. I would be a bit more explicit about the color code

Answer to R3.22: We have now better explained the colour code in the figure legend, the box plots from the previous version are removed.

Comment R3.23) -L146. I would mention that ANOVA was done first.

Comment R3.24) -L151. Please make a reference to the figure where the highest synteny of S1-S2 is shown, and where the letters AB and X are shown

Answer to R3.23: Due to the redundancy of this figure, we have removed it.

Comment R3.25) -L151-155. I do not understand why the comparisons were denoted with letters. Is it based on the statistical difference significance?

Answer to R3.25: To streamline the manuscript, we skipped the figure.

Comment R3.26) -L161-169. I find this part more related to the subgenome assignment already explained in the previous section. Also, a couple of explanatory sentences justifying the k-mer approach, and how it is needed to identify subgenome-specific signatures might be positive.

Answer to R3.26: We thank the reviewer for this important suggestion. Following their suggestion we have now reformulated the entire results section. We hope the reviewer appreciate this new version.

Comment R3.27) -L178. I missed the information about the insertion time calculation in the methods.

Answer to R3.27: We thank the reviewer for pointing this out. As this has been done within the Subphaser pipeline, it was initially missing in our short version of the method description. We have now extended this part in the methodology. In addition, we performed a Ks-based analysis on divergence times (Supplementary Figure 7).

Comment R3.28.1) -L199. When Figure S8 is mentioned, did the authors mean Figure S9, instead?

Answer R3.28.1: Thank you for thorough reading: yes, there was a mistake there. We have now reformulated the results sections and several new supplementary figures are added and the numbering has now changed.

Comment R3.28.2) -Figure 4. Please explain that the meaning of the identity heatmaps. Also explain the color code for the density plots are the same for the heatmaps. According to my interpretation of the color code, the identity between two random windows is on average about 90%... Isn't that too much for the chromosome arms? Also, a little detail: in panel C the axis of the histogram of chromosome 6 of S3, is missing some text.

Answer R3.28.2: We thank the reviewer for bringing this point. In order to bring focus on the centromere characterisation, we show in the new figure 4 only the plots for centromere and not for the whole chromosome, which is now only shown in the supplements. Furthermore, we have replotted all chromosome-wide and centromere heatmaps using ModDotPlot (<https://github.com/marbl/ModDotPlot>), which is more efficient memory-wise and produces similar plots as StainedGlass. We have added this information to the methods section.

The colour gradient used in the heatmap is determined by the sequence identities of the alignments, which are calculated from the number of matches, the number of mismatches, the number of insertion events and the number of deletion events. The colour gradient ranges from 85 to 100% similarity. Alignments below 85% similarity are not visualised. This information has been added to the revised figure. When there are multiple alignments between the same two sequence fragments, all alignments other than the one with the most matches are filtered out regardless of their sequence identity. This explains why the identity between random windows may optically appear high while, in reality, the similarity between individual subregions is much lower. Nevertheless, low frequency hits (grey to blue) are visible in chromosome arms as the reviewer correctly noticed. Potentially this could be explained by the insertion of retroelements sharing sequence identity. In fact, the distribution plots indicate that the Ty1/Copia *BIANCA* family accounting for more than 8% of all repeats is evenly distributed across the chromosome arms. It is conceivable that similarity between these randomly integrated retroelement copies may generate hits in the alignments and the coloured dot plots.

Comment R3.29) -L210-211. "As frequent across both S1 and S2 chromosomes, we did not find any obvious enrichment for tandem repeats but a high enrichment for ATHILA elements at their centromeres" I don't understand why it is said that there is no obvious enrichment. There are regions with obvious overrepresentation of some repeats in figure 4 and Supplementary dataset 2 in all the chromosomes of the S1 and S2 genomes.

Answer to R3.29: This sentence was misleading, as also mentioned in Ref. R2.19 (please see above) and was newly formulated.

Comment R3.30) -L231. I would include a bit more information about these species as most readers would not know if they are diploids, tetraploid, pentaploids, etc. Also, I would also explain in the text, although that the information can be found in the Figure 6d, what phylogenetic relationship those species have with the constitutive subgenomes of *R. canina*. I believe this would facilitate the understanding of the controls and results presented.

Answer to R3.30: We added this information to l. 193ff.

Comment R3.31) -L232. I think this is the first time the acronym SCO is used in the main text, please spell it out.

Answer to R3.31: Thanks, this has been corrected now (l. 195).

Comment R3.32) -L235-236. The text says” the number of SNPs observed in pollen samples was considerably lower than in the respective somatic tissues (Figure 6b), which supports the hypothesis that bivalent formation is non-random but highly specific”. I think that if bivalents were formed by synthetic pairs of random homoeologous chromosomes, and therefore, pollen was formed by random genomes, then the number of SNPs found would be equally lower than in leaf sample. I think that, If I understood the approach properly, what the lower number of SNPs indicates is that the pollens contain much less chromosomes than the (half of) the leaf sample, which strongly support that only a few chromosomes segregate to the future sperm cells. I think that the proof that bivalents are formed by two specific chromosome sets is more strongly supported by the results presented subsequently (L236-267).

Answer to R3.32: To streamline the manuscript, we opted to omit this figure, as the reviewer correctly noted that the information it contained was less precise and somewhat redundant with revised Fig. 3b. Our rationale was based on the fact that leaf samples contain 35 chromosomes (5x7), while pollen contains only 7 chromosomes. If bivalent formation were random within each syntenic group, the SNPs in pollen should represent a random subset of the total SNP population found in leaves (5x). Given a sufficiently large number of nuclei (e.g., 200,000), the SNP distribution in pollen should not significantly differ from that in leaves.

Comment R3.33) -Figure 6. Why in the panel C subsections names are shown instead of the species cited in L231.

Answer to R3.33: We thank the reviewer for pointing this out. However, we think that adding species names would be puzzling, making the visualisation more complex. We have now added the information of the pollen species names in the legend.

Comment R3.34) -L242. I am a bit confused. This is the first time that *R. pseudoscabriuscula* is cited, but it is not mentioned in L231. Also, subsect *Vestitae* is mentioned in Figure 6C. Is *R. pseudoscabriuscula* actually *R. sherardii*? Why *R. corymbifera* is not shown in Figure 6C?

Answer: Thanks for thorough reading. Unfortunately, we mixed up species names between these two taxa both belonging to subsect. *Vestitae*. For streamlining the manuscript, we deleted the pollen data from Fig. 3b and concentrated on subsects. *Caninae* and *Rubigineae* from which we analysed whole genomes.

Comment R3.35) -L258-259. I think that this contradictory and unexpected result calls for some explanation or hypothesis. It might help to repeat the analyses with chromosome-wise concatenated SCO loci (instead of genome-wide). Also, I think it should be explained in the methods section what is the origin and relationship of these plant materials.

Answer R3.35: Thank for this suggestion. We have now performed addition chromosome-wise concatenated alignment phylogenies (Suppl. Dataset 2), which has indeed confirmed that the CAN2 samples always groups with S2 subgenomes from our three assemblies. Indeed, different microsatellite alleles carried by bivalent-forming chromosomes of *R. canina* were also observed in a study by Ritz & Wissemann (2011)³. Thus, we assume that there is variation in genome composition between different accessions of *R. canina*, which deserves future analysis at population genomic level. We hypothesise that S2 genome might form bivalents and being biparentally inherited in some accessions of *R. canina*. We have now added these new findings to the result and added a sentence to the last paragraph in the discussion. The origin of plant material used for pollen analyses is listed in detail in Suppl. Table 5.

Comment R3.36) - L265-266. I think that this conclusion can be only understood once it is also understood that the species whose pollen was analyzed are also pentaploid with the same asymmetric meiosis mechanism. As mentioned above, please explain the ploidy of those species.

Answer to R3.36: We added this information to l. 193f (see Ref. 3/30).

Comment R3.37) -L271-273. Regarding the interpretation that:” This difference in branch lengths within our phylogenetic framework might reflect the differential evolutionary selection pressures with respect to their ability or lack of meiotic pairing and recombination.”, could not also be a matter of how related they are with their closest branch?

Answer to R3.37: This was also raised by Ref. R2.23. A detailed discussion on branch lengths, especially when combining SCO data and whole genome sequences is possibly not sound enough. We deleted this sentence and rephrased the entire section. In addition, we performed an analysis on Ka/Ks ratio which did not point on subgenome-specific selective regimes (Supplementary Figure 7), see also R3.46.

Comment R3.38) -L294-295. Again, not sure of this conclusion.

Answer to R3.38: Unfortunately, we were unable to perform cytological analysis of female meiosis due to technical challenges and time constraints. Female meiosis requires specific stages of flower buds, which must be sectioned using a microtome. Since our plant material flowers only once per year for about two weeks, we could not obtain sufficient samples during the project. Literature review revealed that Täckholm (1922) is the only study to have successfully investigated embryo sac mother cells (EMC) and documented the key stages of female meiosis, while other studies focused solely on pollen mother cells (PMC) (e.g., Blackburn & Harrison 1921; Lim et al. 2005; Lunerova et al. 2020). We provide an additional supplementary figure with Täckholm’s drawing of female meiotic stages (Supplementary Figure 21).

To address this, we instead performed flow cytometric analysis of the embryo/endosperm ratio in *R. canina*, which clearly indicated sexual reproduction. We observed a 5x embryo/9x endosperm ratio, consistent with the predictions of *anina* meiosis (see also Kolarčik et al., 2014). While the occurrence of gametophytic apomixis in dogroses has been debated (e.g., Fagerlind 1940; Gustafsson & Hakansson 1942; Gustafsson 1944; Kroon & Zeilinga 1974), it has largely been dismissed based on the intermediate characteristics of synthetic hybrids. Nybom et al. (2004, 2006) provided genetic evidence for apomixis, estimating it at around 5% in open-pollinated offspring. This suggests that apomixis may occur occasionally, but the majority of offspring are produced sexually. Further supporting this, our analysis of reciprocal synthetic hybrids between subsect. *Caninae* and *Rubigineae* confirmed the expected genomic composition consistent with *Canina* meiosis (see also R1.4, R2.5, R3.3).

Comment R3.39) -L308-309. “However, the high level of BUSCO completeness in all subgenomes does not indicate a significant loss of protein-coding genes from univalent chromosomes, suggesting their potential functionality.” Actually, fractionation is normally a property of duplicated genes in polyploids (by definition non-BUSCO). I wonder if the authors could say something about fractionation in NON-BUSCO.

Answer to R3.39: We thank the reviewer for this. Indeed, we have performed genome-wide fractionation analysis and could show that there is no apparent fractionation among all five chromosome sets (Suppl. Figure 6).

Comment R3.40) -L380. It is not clear what these diploid rose DNAs were used for.

Answer to R3.40: We rephrased the sentence: “To analyse the phylogenetic origin of the subgenomes of allopolyploid *R. canina* we sampled 30 rose individuals of 24 diploid species across the genus *Rosa*.”

Comment R3.41) -L390-392. Also, please explain the ploidy of those species here in the methods.

Answer to R3.41: We added this information (all are pentaploids).

Suggested improvements: Supplementary

Comment R3.42) -L55: would it be possible to summarize in one sentence what information is provided by this plot? I am talking specially about the A/B etc...

Answer to R3.42: This figure shows the quantification of heterozygous k-mers identified in the *R. canina* genome using SmudgePlot (<https://github.com/KamilSJaron/smudgeplot/wiki/FAQ>). Based on this approach we “assume that at least some of the k-mers will be heterozygous - i.e. one of the k-mer is from one haplotype and the other k-mers is the other. If the ploidy is higher, like triploid, we expect to find locations that have the same k-mers in two of the haplotypes one SNP difference in the third haplotype. If the genome is tetraploid we will detect some pairs where two haplotypes are similar and two haplotypes are too diverged (looking like AB), some cases where one haplotype will be diverged but the three other will be "triploid like" and therefore it will look like AAB and finally majority of heterozygosity will be carried by either AABB or AAAB structures which will also tell us whether the genome structure is AA'BB' or AA'A"B (i.e. what is the branching of haplotypes)”. In the case of the pentaploid genome of *R. canina* a high amount of heterozygous k-mer combinations AAAAB and AAABB can be seen. We added an explanation to Supplementary Fig. 1.

Comment R3.43) -Figure S3. Further explanation of the figure should be provided in the caption of this figure, in my opinion. What is this dot plot expressing? Where is this data coming from? What kind of linkage is plotted? Genetic linkage? Synteny? Identity? Hi-C contacts? What is the meaning of the comparisons highlighted in blue?

Answer to R3.43: We apologize for confusing the reviewer by incomplete description of the figure. Figure S3 (now Suppl. Figure 2) shows all to all chromosome comparisons generated by a Dotplot in SyMAP. We have now elaborated further the legend of this figure as follows: “**Supplementary Figure 2.** All-to-all dotplot comparisons of the assembled *Rosa canina* chromosomes (n = 35). The plot shows the chromosomes showing self-synteny at a sequence similarity threshold of 98% across all the 35 scaffolded pseudochromosomes. Please note the high similarity between *S1* haplotypes and between *S1* and *S2* subgenomes, as well as between *R3* and *R4* subgenomes. In contrast, sequence similarity between *S* and *R* subgenomes is remarkably lower. Blue rectangles depict the syntenic regions identified that share > 98% sequence similarity.”

Comment R3.44) -Figures S4 and S5. Please, explain more explicitly where the color code is and include a Y-axis label in the plots on the right.

Answer to R3.44: Thanks for pointing this out. We have now added the y-axis label and colour scale in the plots and add detailed description in the legend.

Comment R3.45) -Supplementary dataset 1. Please verify if Mann-Whitney test is appropriate in this situation. My understanding was that in this context a Kruskal-Wallis plus Dunnett's test.

Answer to R3.45: The reviewer probably meant the CANR4 abundance comparisons. The reviewer is right that the Kruskal-Wallis test is more appropriate than the Mann-Whitney test at this place. We wish to mention that in the revised version that analysis of repeat abundances in the *R. canina* subgenomes has been markedly developed. Specifically, we quantified individual types of repeats in all centromeres (Supplementary Figure 14). We found that that the univalent-forming chromosomes are typically enriched for CANR4 and other satellite repeats while those of the bivalent-forming are enriched for Athila retroelements. For the statistical comparisons, we applied generalized Linear Mixed Models (GLMMs) using Markov Chain Monte Carlo (MCMC) methods. Further, we analysed the centromeres in the recently sequenced *R. chinensis* genome, which is close to the *Synstylae*-based S1 and S2 showing that its chromosomes are structurally related to S1 and not S2 subgenomes of *R. canina* (Supplementary Dataset 3).

Comment R3.46) Final suggestion. (the authors are not required by any means to do this). I really wonder if the bivalent- and univalent-t forming genes evolve differently... Would it be possible to analyze Ka/Ks patterns in both kinds of genomes.

Answer R3.46: This is a very valuable remark, which was also raised by reviewer R2.25 (please see there). We have now performed comparative pairwise Ka/Ks ratio between each pair of subgenomes with their closest diploid relative genomes available. The Ka/Ks ratios have now been determined for a set of 7076 genes in each (S1, S2, R3 and R4) *R. canina* subgenome (Suppl Figure 7). The low ω values indicate that most mutations have a symmetrical character and are not influencing the amino acid composition of encoded proteins. Hence, the genes in bivalent and univalent genes seem to be evolving under similar purifying selection. There might be several interpretations: (i) Most selected genes are slowly evolving. (ii) Alternatively, and most likely, hybridisation events giving rise to the *R. canina* genome could be of recent origin, and hence there was insufficient time to accumulate non-synonymous mutations.

References

- Dias Y, Mata-Sucre Y, Thangavel G, Costa L, Baez M, Houben A, Marques A, Pedrosa-Harand A. 2024. How diverse a monocentric chromosome can be? Repeatome and centromeric organization of *Juncus effusus* (Juncaceae). *Plant J* **118**(6): 1832-1847.
- Herklotz V, Ritz CM. 2017. Multiple and asymmetrical origin of polyploid dog rose hybrids (*Rosa* L. sect. *Caninae* (DC.) Ser.) involving unreduced gametes. *Ann Bot* **120**(2): 209-220.
- Hofstatter PG, Thangavel G, Lux T, Neumann P, Vondrak T, Novak P, Zhang M, Costa L, Castellani M, Scott A, et al. 2022. Repeat-based holocentromeres influence genome architecture and karyotype evolution. *Cell*.
- Kalfusová R, Herklotz V, Kumke K, Houben A, Kovařík A, CM R, Lunerová J. 2024. Epigenetic histone H3 phosphorylation marks discriminate between univalent- and bivalent-forming chromosomes during canina asymmetrical meiosis. *Annals of Botany* **133**(3): 435-446.
- Lunerová L, Herklotz V, Laudien M, Vozárová R, Groth M, Kovařík A, Ritz CM. 2020. Asymmetrical canina meiosis is accompanied by the expansion of a pericentromeric satellite in non-recombining univalent chromosomes. *Annals of Botany* **125**(7): 1025-1038.

- Naish M, Alonge M, Wlodzimierz P, Tock AJ, Abramson BW, Lambing C, Kuo P, Yelina N, Hartwick N, Colt K, et al. 2021.** The genetic and epigenetic landscape of the Arabidopsis centromeres. *BioRxiv*: 2021.2005.2030.446350.
- Nuckolls NL, Bravo Nunez MA, Eickbush MT, Young JM, Lange JJ, Yu JS, Smith GR, Jaspersen SL, Malik HS, Zanders SE. 2017.** wtf genes are prolific dual poison-antidote meiotic drivers. *Elife* **6**.
- Nybom H, Esselink GD, Werlemark G, Leus L, Vosman B. 2006.** Unique genomic configuration revealed by microsatellite DNA in polyploid dogroses, *Rosa* sect. *Caninae*. *Journal of Evolutionary Biology* **19**(2): 635-648.
- Nybom H, Esselink GD, Werlemark G, Vosman B. 2004.** Microsatellite DNA marker inheritance indicates preferential pairing between two highly homologous genomes in polyploid and hemisexual dog-roses, *Rosa* L. sect. *Caninae* DC. *Heredity* **92**(3): 139-150.
- Ritz CM, Köhnen I, Groth M, Theißen. G, Wissemann V. 2011.** To be or not to be the odd one out – allele-specific transcription in pentaploid dogroses (*Rosa* L. sect. *Caninae* (DC.) Ser). *BMC Plant Biology* **11**: 37.
- Täckholm G. 1922.** Zytologische Studien über die Gattung *Rosa*. *Acta Horti Bergiani* **7**: 97-381.
- Wissemann V, Hellwig FH. 1997.** Reproduction and hybridisation in the genus *Rosa*, section *Caninae* (Ser) Rehd. *Botanica Acta* **110**(3): 251-256.

Referees' comments:

Referee #1 (Remarks to the Author):

Thank you – the authors have addressed my concerns well, both by rewriting the text, and adding the new CENH3 ChIP-seq and immunostaining data, as well as the work on synthetic hybrids. Together these new data provide strong support for the proposed model of univalent and bivalent inheritance, and provide a much higher resolution picture of centromere structure and function.

R1.1: We thank the reviewer for the very helpful comments that supported the improvement of our manuscript.

Abstract line 39 – should this be 'role of centromeres'? The use of 'chromosomes' here doesn't seem quite right to me.

R1.1: We thank the reviewer for this suggestion. Indeed, the wording "role of centromeres" better reflects our findings. We have now replaced this in the abstract.

Referee #2 (Remarks to the Author):

The authors have done an excellent job responding to all of my comments. I have no further issues with this manuscript.

R2.1: We thank the reviewer for the very helpful comments that supported the improvement of our manuscript.

Referee #3 (Remarks to the Author):

After working on this version of the manuscript, I really appreciate the changes, and I feel the manuscript is more readable and better explained. I am also very happy about the new results included, particularly the inclusion of new genomes, the CENH3 ChIP-Seq, and especially the cytology in male meiosis. However, I still have important concerns about the main claim that differences in centromere structure between subgenomes explain their different behaviors during meiosis.

R3.1: We thank the reviewer for the very helpful comments that supported the improvement of our manuscript.

I am aware that the text shows some caution regarding this claim in some parts (for instance, in L361-362: "CANR4 enrichment alone does not fully explain univalent drive."), but other statements are, in my opinion, not as cautious. For instance, L50-51: "These structural differences likely drive the asymmetric segregation of univalents during female meiosis" or L352-354: "The dominance of CANR4 repeats in univalent centromeres suggests a potential mechanism for centromere drive – a phenomenon in which certain centromeres are preferentially transmitted during meiosis – leading to biased segregation (Figure 6a)." Also, in L362-366: "the structural divergence of centromeres is also closely linked to their behavior in male meiosis."

R3.2: We thank the reviewer for pointing this out. We have now made sure that we present our wording in a more cautious form. We avoided speculative hypotheses such as "These structural differences likely drive the asymmetric segregation of univalents...", throughout the text.

Future results might prove me wrong, but with the current evidence, in my view, the association between centromeric structure and meiotic behavior is at most correlative. I made a contingency table for meiotic behavior (bivalent vs. univalent) against

centromere composition (satellite-based vs. retrotransposon-based), and if I perform a statistical test (chi-square or Fisher's exact), it is not significant. This suggests that the observed combinations of meiotic behaviors with centromere compositions could occur by chance, assuming independence between the two factors. Additionally, in *R. agrestis*, where the bivalent- and univalent-forming chromosomes change, the compositions of centromeres remain the same, making it difficult to assume that one determines the other. Overall, I think the text does not clearly acknowledge the lack of evidence that differences in centromere structure between subgenomes determine their different behavior during meiosis.

R3.3: We thank the reviewer for this criticism. We have now added a sentence in the discussion to highlight the current limitations of our study. We also removed speculative sentences about the strong centromeres driving premature separation of chromatids during the first division of pollen meiosis. We agree that future studies will be necessary to validate the association of *CANR4*-based centromeres with their drive in female meiosis.

“However, while the observed correlation is intriguing, we acknowledge that future studies will be essential to confirm whether the expansion of *CANR4* in univalent centromeres is directly linked to their drive during female meiosis.”

On the other hand, I am okay with the word “bimodal” — I think it is a great choice, actually — because it implies a trend that is not always fulfilled. Additionally, I would be very curious about differences in the relative coverage of CENH3 ChIP reads, so that centromere function can be compared across chromosomes and subgenomes. Perhaps CENH3 abundance might be better correlated with differences in meiotic behavior.

R3.4: We thank the reviewer for this suggestion. We suppose the “coverage” mentioned here means the CENH3 abundance not the alignment coverage of ChIP-seq data. Regardless, it is worth pointing out that the CENH3 ChIP enrichment shown on new Figure 3d was already normalized by coverage to rule out the impact of sequencing depth. We have explicitly clarified it in the legend of Figure 3d as well. To respond to the reviewer's concern regarding the difference of CENH3 across chromosomes, we added a metaplot of CENH3, DNA methylation, *ATHILA* density, and *CANR4* density for each chromosome in Supplementary Figure 14. It was observed that when *CANR4* is the dominant element at centromeres, CENH3 is more enriched compared to *ATHILA*-dominant centromeres. To quantify this correlation, we plotted the CENH3 abundance against *CANR4* size for every chromosome in new Figure 3e, which clearly shows that CENH3 abundance is positively correlated with *CANR4* length at centromeres, implying that the accumulation of *CANR4* repeats increases the CENH3 occupancy on the centromere. Interestingly, the *CANR4*-free Rca1_R4 centromere, which shows the cenLTR array, did show one of the highest CENH3 abundance signals for *CANR4*-free centromeres (new Figure 3e).

New Figure 3e panel: Linear regression of *CANR4* size and CENH3 abundance on the centromere across all chromosomes. Each dot colored by its subgenome presents a chromosome. The abundance of CENH3 was calculated by the sum of CENH3 ChIP-seq (\log_2 CENH3/H3) signals on centromeres normalized by coverage. The R-square of the linear regression model is 0.84. Since the data were not normally distributed (Shapiro-Wilk normality test p -value $\ll 0.05$), we used Spearman's rank correlation, resulting in 0.93. Note the high CENH3 enrichment for the cenLTR1-based centromere in Rca1_R4, which lacks *CANR4* repeats.

New Supplementary Figure 14. Metaplot of CENH3 enrichment, DNA methylation, and centromeric elements – *ATHILA* and *CANR4* density on each chromosome. CENH3 enrichment marked in dark red was calculated by CENH3 ChIP-seq (\log_2 CENH3/H3) signal normalized by coverage. DNA methylations have 3 contexts with CpG marked in black, CHG marked in dark grey, and CHH marked in light grey. *ATHILA* (yolk yellow) and *CANR4* (magenta) were presented by their density. All signals were calculated in 50 kb adjacent windows. All chromosome coordinates were scaled based on their distance to centromere against the distance of centromere (CEN) to telomere (TEL). Centromere position (CEN) was defined by where maximum CENH3 enrichment was located. All signal values (y-axis) were scaled from 0 to 1 by global minimum to global maximum except for DNA methylations, which retained the original percentage values. The p- and q-arm values were averaged and mirrored.

Additional Comments

Although I am very glad that the authors included CENH3 ChIP-Seq data in this version, I think it is fair to explicitly mention the limitation of using data from leaves to study meiotic behavior.

R3.5: We agree with the reviewer that our ChIP-seq only reflects the centromere composition in somatic cells. However, performing ChIP-seq for plant meiocytes is a rather challenging experiment. In future, more sensitive methods like CUT&Tag could be adapted for low input plant samples. We have now added to the end of fifth paragraph in the centromere results “These results confirm the bimodal architecture of *R. canina* centromeres, which are preferentially *ATHILA*-based in bivalents and *CANR4*-based in univalents with the caveat that our ChIP-seq experiment was performed with leaf tissue.”

L227-246: These crosses demonstrate that the subgenome that can be in double dosage varies depending on the parents used for the cross, but unless I am missing something, this experiment shows that bivalents are formed during meiosis. I assume that to support this claim, cytological observations of meiosis or sequencing from sorted pollen would be needed. Probably, the word “potentially” in L244 is meant to indicate this, but I think it should be clearer that whether bivalents can be formed is not directly tested.

R3.6: The reviewer is totally right about this. Although hybrids have double dosage of different subgenomes depending on each cross, we do not know whether they form bivalents or not, which would be, however, a very interesting point for future research. Our previous population genetic study based on microsatellites revealed considerable number of hybrids between *Caninae* and *Rubigineae* but interestingly, we found very few 5x hybrids but rather a high number of 6x hybrids which originated from unreduced egg cells (Herklotz et al, 2017; Herklotz & Ritz, 2014; 2017). Therein, we argued that bivalent-formation in hybrids might be hindered when they came from different (subsectional) parental origin. Some microsatellite alleles allowed us to track subsection-of-origin specific alleles, and based on these alleles we estimated the abundance of hybrids in natural populations. In principle, if subsection-of-origin specific alleles are lacking in potential hybrids we could have misinterpreted samples as non-hybrids because they contained 4 alleles in 5x samples, because microsatellite data did not allow for estimating allele dosage.

We have now rewritten this to make the text more clear. Thus, we have replaced the wording “bivalent-forming” for “different”. It now reads: „Our findings further suggests that different subgenomes are potentially inter-exchangeable in hybridisation events, however, hybrids in extant populations originated mostly from unreduced eggs

suggesting some subsection-specific differentiation subgenomes which might impact bivalent formation⁴⁵.

I am wondering how exceptional the result found in CAN2 is. Could it be that the bivalent-forming subgenome is highly variable? Perhaps depending on which haploid subgenome came from the pollen donor? This would explain why the univalent-forming subgenomes do not degenerate (e.g., through fractionation and higher K_a/K_s), as all subgenomes have a chance to recombine occasionally. In other words, could it be that the claim that the bivalent-forming subgenomes are interchangeable is true not only for interspecies crosses but also within species? Of course, this is just speculation, but is there any previous data or evidence to rule out this scenario?

R3.7: The CAN2 is indeed a very intriguing case. In the near future we aim to also sequence this sample to further validate its subgenome composition. The reviewer asks several questions that we are planning to address in future studies. Indeed, it will very interesting to understand whether CAN2 is an exception or cases like this can be more commonly found across wild dogroses. As mentioned during the first revision, we tried to use SCO data from dogroses to check for the variability in bivalent-forming subgenomes across dogroses but the low sequence variability combined with only Illumina short reads (150 bp paired-end) let us fail here. We again tried to map the short reads from the target enrichment experiment of each sampled dogrose against the single-copy gene set of the *R. canina* assembly, identified using the OrthoFinder pipeline. We expected that short reads from bivalents should map at twice the quantity to the corresponding subgenome. Despite applying the highest possible stringency and counting only uniquely mapped reads (resulting in only a few hundred mapped reads out of millions), an unambiguous assignment of the bivalent subgenome was not possible, probably also due to between species variations.

Furthermore, It may be significant that the ITS1 markers showed relatively large variation across the *R. canina* populations (populations taken from Herklotz & Ritz (2017)) which may support the hypothesis that bivalent genomes might be somewhat variable across the populations. However, the result also shows that the univalent R4 and R3 genomes from section *Rosa* probably never form the bivalents in *R. canina* since Canina-types ITS are predominating (except perhaps of HK-15 pop where the Canina/Rubiginosa types are balanced). Thus, lack of degeneracy as implied from low K_a/K_s ratios cannot be explained by interchangeable *Synstylae* and *Rosa* genomes in *R. canina*.

Proportions of *nrlITS*-types in 16 accessions of *R. canina* across Central and Eastern Europe obtained by conventional cloning.

Geographic origin of samples: BD: Germany, Bayern; HD: Germany, Hessen; HK: Croatia, Gvozdi; HW: Germany, Hessen; IC: Italy, Trentino; LB: Slovenia SE; MH: Germany, Mecklenburg-Vorpommern; PC: Poland, Lesser Poland; RB: Serbia, South Banat; RT: Rheinland-Pfalz; UC: Hungary, Zala; WC: Germany, Baden-Württemberg; DE: Germany, Sachsen; SW: Sweden S; CZ: Czech Republic S. Individuals of *R. canina* were sampled during a population genetic study by Herklotz & Ritz (2017), see more details of sampling therein.

In short, we hope the reviewer understands this will require a deeper sampling and sequencing of dogrose populations, which extends the scope of our current manuscript.

I also have some difficulty understanding the model of how male gametes only transmit bivalent-forming chromosomes while univalents are lost. L335-336 states: “while single chromatids from univalents lagged behind and were likely eliminated (Figure 5i–j; Supplementary Movies 9–10, n = 31).” It is not clear to me from the figure how univalents are eliminated; I just see a few chromosomes lagging behind. Since in metaphase II there are no more bivalents, only chromosomes, I do not think it is possible to determine which chromosomes originated from a bivalent and which from a univalent. Examining the telophase II stage might be more informative.

R3.8: This is a very insightful observation. Indeed, in dogroses at late stages after telophase II, polyads are formed. While it is theoretically possible for pollen to contain both bivalent-forming chromosomes (these should always end up into gametes due to their regular segregation) and univalents that occasionally end up in gametes by chance, our flow cytometry data (Figure 2a) show that mature pollen are always haploid (1x = 7 chromosomes). This suggests that univalents are absent from viable pollen.

Indeed, we have now checked for polyads and found that they mostly reveal nuclei with correct number of centromeres containing regular structures such as the nucleoli, while several micronuclei with varying number of centromere signals are formed (now added to the new Extended Data Figure 9 and Supplementary Movie 11), the low pollen viability (~20%) implies that only a small fraction of pollen survives, and these are exclusively haploid. This indicates a strong selection process during pollen

development, as synthetic hybrids did not exhibit altered dosages of univalents (Esselink et al, 2004; Nybom et al, 2004; Ritz & Wissemann, 2011). The most plausible explanation is that univalents are actively eliminated or excluded during pollen maturation, ensuring that only bivalent-forming chromosomes are transmitted.

Extended Data Figure 9. CENH3 immunostaining in polyads that are formed at the end of male meiosis in *R. canina* (S27). (a) A single polyad showing at least two nuclei with seven CENH3 signals (arrowheads), while several other nuclei show irregular number of centromeric foci. (b–d) Different Z-stacks of three polyads showing few nuclei with seven centromeric foci, while several other smaller nuclei are formed with irregular number of centromeres. Source data available in Supplementary Dataset 14. Scale bars = 10 μ m.

Actually, I find the observations in metaphase I in Figure 5 quite remarkable. It seems that univalents systematically display bipolar orientation. In *Arabidopsis thaliana*, this only happens when sister chromatid cohesion is defective (D'Erfurth et al., 2009), but it appears to be more common in wheat univalents (Lukaszewski, 2010). This observation contrasts with that of female meiosis by Täckholm (1922), where univalents seem to have monopolar orientation. I wonder if it would be worthwhile to explicitly discuss the possibility that sexually dimorphic adjustments in sister chromatid cohesion regulation could be part of the adaptations enabling canina meiosis.

R3.8: We thank the reviewer for this suggestion. Indeed, this is quite remarkable. We agree that it must be some difference in spindle symmetry between male and female meiosis, as well as a difference in how sister chromatid cohesion is regulated in both sexes. We have now added a sentence in the discussion to highlight this. It reads: “Remarkably, in *Arabidopsis thaliana*, bipolar orientation of univalents only happens when sister chromatid cohesion is defective⁵³, but it appears to be more common in wheat univalents⁵⁴. This observation contrasts with that of female meiosis, where univalents seem to have monopolar orientation in dogroses (Supplementary Figure 17)³. Thus, a potential role for sexual dimorphism in sister chromatid cohesion regulation could be part of the adaptations enabling *Canina* meiosis.”

Minor Comments

L34: “relies on meiotic chromosome pairing” → I would add “forming bivalents” if the word count allows.

R3.9: We have added this now.

L35: “Ploidy asymmetry” is not a well-known term. The same applies to “polyploid hemisexual” in L96.

R3.10: We agree with the reviewer and have now removed the term ploidy asymmetry from the title and the abstract.

Figure S7: Some axis labels (e.g., FC, RC) are not clarified in the legend. Comparisons with DToL genomes would be interesting (e.g., *R. agrestis* R4 vs. *R. canina* R4 or S27 subgenomes vs. DToL subgenomes).

R3.11: We thank the reviewer for this comment. However, we already state at the end of the figure legend the meaning of RC = *R. chinensis*. We did not find FC, but only FvH4, which stands for *F. vesca*. Regarding the comparison with the DToL genomes, we think that this would not add much as despite differences in subgenome dosage, they share the same subgenomes. Furthermore, this would require quite some additional work, as we do not have reliable gene annotation for these two genomes. I hope the reviewer understands that.

L97-99: The introduction more cautiously states the correlation: “accumulates in dogroses, especially in univalents, possibly linking centromere expansion with their drive in female meiosis.”

R3.12: Thank you. We have better rephrased the correlation throughout the text.

L185-200: It would be nice to specify how many individuals from each species were used for DNA extraction. Were different *R. canina* individuals analyzed?

R3.13: Only one *R. canina* individual was used for sequencing, immunostaining and ChIP-seq analysis, while an additional individual was also used for FISH analysis. This is now better mentioned in the Methods.

Figure S10: The meaning of SG1, SG2, and SG3 is not included in the legend.

R3.14: We have now added the meaning in the figure legend.

Figure 3D: The “5C” label looks a bit like “6C.”

R3.15: Thank you. We have improved that.

L265-266: “The total centromere length per subgenome confirmed that the S2 and R3 centromeres were larger compared to those of S1 and R4.” → It would be more accurate to state that the length of the centromeric repeat array is larger, as in many centromeres the repeat array is larger than the CENH3-positive region. Later, once immunostaining results are presented, would be a better time to discuss centromere size.

R3.16: In fact, for *R. canina* S7 what is measured here is the total size of CENH3 binding regions and not CANR4 arrays, while for DToL genomes we used the corresponding regions annotated in *R. canina* S27 (this is mentioned in Methods). Therefore, total centromere length is more appropriate. We have changed the text to make this clearer. Now it reads “The total centromere length, as the measurement of the CENH3 binding regions per subgenome, confirmed that the S2 and R3 centromeres were larger compared to those of S1 and R4”. We have also moved this to when we describe the immunostaining results.

L267-268: “CANR4-based centromeres were also found in six (syntenic groups 2, 4, and 5) out of 14 centromeres of the S1_h1/h2 bivalent-forming chromosomes and in two R4 chromosomes.” → It might be clearer to state “3/7 syntenic groups” rather than “6/14,” as this better conveys the deviation from the expected pattern.

R3.17: Thank you. Indeed, this represents our findings better. We have now changed the text accordingly.

L324-325 and Figure 5e: “At onset of metaphase I, we observed seven bivalents organized at the center of the cell surrounded by 21 univalents.” → This may not be metaphase I but rather pro-metaphase I or late diakinesis.

R3.18: We have now changed Metaphase I for Pro-Metaphase I.

Comments on Responses to Reviewers

Figures S19 and S20: Scale bars are not explained

R3.19: Thank you for noticing this. We have now added to the figure legends.

Comment R2.6) Line 127. Any evidence for recombination between the more divergent chromosomes early following polyploidization?

Answer to R2.6: As far as we know there is no evidence available. Also based on our data, we could not detect any sign of recombination between univalents and the bivalent-forming subgenomes. This greatly supports that hybridisation events most likely happened in an asymmetric genome contribution, with two copies of a subgenome hybridised with an additional copy of a different subgenome, potentially forming an allotriploid.

My comment -> I would emphasize this in the text.

R3.20: We thank the reviewer for this suggestion. Although we have no evidence for recombination between the different subgenomes, we think that more robust analyses using chromosome-specific markers should be performed in future studies to rule out this hypothesis. Therefore, we think that it would be more cautious to avoid mentioning this in the current story. I hope the reviewer understands that.

Answer to R3.7: The reviewer is entirely right in that the R4 genome does not show expansion of centromeric satellite as opposed to R3 and S2 genomes. We now quantified CANR4 repeats in each R. canina chromosome including the two new genomes available from DToL. We used bivariate Bayesian Generalized Linear Mixed Model (GLMM) using Markov chain Monte Carlo techniques to statistically analyse the proportion of ATHILA and CANR4 across subgenomes in R. canina (S27 and DToL) and R. agrestis (Supplementary Figure 14). The model included subgenome as a fixed effect, while pairing type (Univalent/Bivalent) among others were set as random effects. This experiment clearly showed that the bivalent-forming subgenomes (S1 and R4) harbour ATHILA-type or mixed ATHILA/CANR4 satellite-type centromeres irrespective of genetic background. The proportion of CANR4 satellite is however low. In contrast, the S2 and R3 subgenomes, which always form univalents, show high content of CANR4 satellite. The genome space of the satellite is extraordinary in some of these chromosomes. For example, in Rca3_S2, Rca7_S2, Rca3_R3 and Rca4_R3 chromosomes it far exceeds 2 Mb which is equivalent to >10000 copies of CANR4 monomers. Given that the chromatin of CANR4 satellite is more enriched for CENH3 (Figure 4), we believe that the R3 and S2 chromosomes harbour strong centromeres supporting our hypothesis about the evolution of centromeric drive in R. canina.

My comment -> If I understand mixed models properly I thought that pairing type as a fixed effect (and not sub-genome) since pairing type is the explanatory variable whose effect needs to be analyzed. If subgenome is used as a fixed affect, I am not surprised that a correlation is found as there are clear differences between subgenomes. What I am not so sure about if there is a correlation between pairing type and centromere type.

R3.21: We thank the reviewer for this criticism. The reviewer is completely right, it would be much clearer to have pairing type as the fixed factor. We have redone the analyses to check whether CANR4 abundance differed between bivalents and univalents having genome, subgenome and syntenic groups as random factors. CANR4 abundance was significantly higher in univalents. In addition, we also checked if there is a difference between univalents (i.e. S2, R3) against bivalents and univalents which act in other species as bivalents (i.e. S1, R4), which also showed that the latter group had a lower CANR4 abundance compared to univalents. You are right as mentioned above that this is a correlative observation and does not necessarily mean causality but we think it is remarkable that CANR4 abundance is rather related with pairing behaviour than with phylogenetic origin of the subgenome. We adjusted the Supplementary Dataset 12, accordingly.

References

Esselink, G. D., Nybom, H. & Vosman, B. (2004) Assignment of allelic configuration in polyploids using the MAC-PR (microsatellite DNA allele counting-peak ratios) method. *Theor Appl Genet*, 109(2), 402-8.

Herklotz, V., Mieder, N. & Ritz, C. M. (2017) Cytological, genetic and morphological variation in mixed stands of dogroses (Rosa section Caninae; Rosaceae) in Germany with a focus on the hybridogenic *R. micrantha*. *Botanical Journal of the Linnean Society*, 184(2), 254-271.

Herklotz, V. & Ritz, C. M. (2014) Spontane Hybridisierung von Hundsrosen (Rosa L. sect. Caninae (DC). Ser.) an einem natürlichen Vorkommen in der Oberlausitz (Sachsen, Deutschland). *Peckiana*, 9, 1-12.

Herklotz, V. & Ritz, C. M. (2017) Multiple and asymmetrical origin of polyploid dog rose hybrids (Rosa L. sect. Caninae (DC.) Ser.) involving unreduced gametes. *Ann Bot*, 120(2), 209-220.

Nybom, H., Esselink, G. D., Werlemark, G. & Vosman, B. (2004) Microsatellite DNA marker inheritance indicates preferential pairing between two highly homologous genomes in polyploid and hemisexual dog-roses, *Rosa* L. sect. *Caninae* DC. *Heredity*, 92(3), 139-150.

Ritz, C. M. & Wissemann, V. (2011) Microsatellite analyses of artificial and spontaneous dogrose hybrids reveal the hybridogenic origin of *Rosa micrantha* by the contribution of unreduced gametes. *Journal of Heredity*, 102(2), 217-227.

Referees' comments:

Referee #3 (Remarks to the Author):

I am fully satisfied with this latest version as now it expresses more caution in some of the proposed conclusions. From my perspective, this is a great piece of work and the article is now of the highest standard.

I have a few really minor suggestions though, that I would leave up to the authors and the editor if they want to implement or not:

-L162-163: “subgenomes, which were distinctly separated in both species (Supplementary Fig. 8c–d). This pattern suggests that the univalent subgenomes, S2 and R3, are accumulating more divergence over time.” I think it is a bit early to speak about univalent-forming sub-genomes at this point of the manuscript, since this conclusion came later.

A: We have now removed the word univalent from that sentence.

-I would also suggest a bit more of extra caution while interpreting the results in the section entitled “Deciphering bivalent-forming subgenomes and meiotic asymmetry in dogroses”. I feel that this nice piece of data shows unambiguously which genomes are transmitted from the male and the female side. Which genomes are forming bivalents, can only be unambiguously observed cytologically. I totally agree that based on previous literature and on the new results provided by this work. Proposing that the bivalent forming chromosomes are the one that are transmitted by pollen is a very solid model, but I would maybe state the slight difference between the two things.

A: We have now rephrased the first sentence of this section to better state this. It now reads “Only bivalent-forming chromosomes are able to segregate properly and produce viable haploid (1x) pollen in dogroses. Thus, we used flow sorting to isolate pollen nuclei as a proxy to confirm which subgenomes are exclusively pollen-inherited and form bivalents in dogroses.”

-I would add scale bars on figures 2A and 2D.

A: This has been added now.

Only one
(rather than two)?

Bivalent?